# Proteomic analysis reveals key differences between squamous cell carcinomas and adenocarcinomas across multiple tissues

Qi Song [1,5], Ye Yang [1,5], Dongxian Jiang [1,5], Zhaoyu Qin [1,5], Chen Xu[1], Haixing Wang[1], Jie Huang[1], Lingli Chen[1], Rongkui Luo [1], Xiaolei Zhang[1], Yufeng Huang[1], Lei Xu[1], Zixiang Yu [1], Subei Tan [1], Minying Deng[1], Ruqun Xue[1], Jingbo Qie[1], Kai Li [1], Yanan Yin [1], Xuetong Yue [1], Xiaogang Sun [2], Jieakesu Su[1], Fuchu He[3✉], Chen Ding [1,2,4✉] & Yingyong Hou [1✉]

Squamous cell carcinoma (SCC) and adenocarcinoma (AC) are two main histological subtypes of solid cancer; however, SCCs are derived from different organs with similar morphologies, and it is challenging to distinguish the origin of metastatic SCCs. Here we report a deep proteomic analysis of 333 SCCs of 17 organs and 69 ACs of 7 organs. Proteomic comparison between SCCs and ACs identifies distinguishable pivotal pathways and molecules in those pathways play consistent adverse or opposite prognostic roles in ACs and SCCs. A comparison between common and rare SCCs highlights lipid metabolism may reinforce the malignancy of rare SCCs. Proteomic clusters reveal anatomical features, and kinase-transcription factor networks indicate differential SCC characteristics, while immune subtyping reveals diverse tumor microenvironments across and within diagnoses and identified potential druggable targets. Furthermore, tumor-specific proteins provide candidates with differentially diagnostic values. This proteomics architecture represents a public resource for researchers seeking a better understanding of SCCs and ACs.

[1] Department of Pathology, Zhongshan Hospital, State Key Laboratory of Genetic Engineering and Collaborative Innovation Center for Genetics and Development, School of Life Sciences, Institute of Biomedical Sciences, Human Phenome Institute, Fudan University, Shanghai, China. [2] State Key Laboratory Cell Differentiation and Regulation, Overseas Expertise Introduction Center for Discipline Innovation of Pulmonary Fibrosis, (111 Project), College of Life Science, Henan Normal University, Xinxiang, Henan, China. [3] State Key Laboratory of Proteomics, Beijing Proteome Research Center, National Center for Protein Sciences (Beijing), Beijing Institute of Lifeomics, Beijing, China. [4] Academy of Medical Science, Zhengzhou University, Zhengzhou, China. [5] These authors contributed equally: Qi Song, Ye Yang, Dongxian Jiang, Zhaoyu Qin. ✉email: hefc@nic.bmi.ac.cn; chend@fudan.edu.cn; hou.yingyong@zs-hospital.sh.cn

Solid cancers in humans comprise two main histological subtypes, squamous cell carcinomas (SCCs) and adeno-carcinomas (ACs), representing a major morbidity and mortality worldwide[1,2]. Options for the diagnosis and treatment of ACs have emerged through a better understanding of the molecular mechanisms of tumor formation and progression[3–6]; however, the indicators for diagnosis of metastatic SCCs and treatment of advanced SCCs remain largely unknown.

SCC is an aggressive malignancy arising within the stratified epithelium of skin, lung, esophagus, aerodigestive or genitourinary tracts, owing to increased exposure to certain risk factors, such as Human Papilloma Virus (HPV) infection, Epstein–Barr Virus (EBV) infection, smoking or sun exposure[7–10]. SCC can also be found in atypical tissues, such as pancreas[11,12], thyroid[13,14], breast[15–17], and anus[18,19]. Clinically, these rare SCCs (usually called metaplastic SCC) are more metastatic and aggressive than common SCCs[12,14,16]. However, SCCs share histological features, such as the presence of squamous differentiation visible by the formatting of keratin peals or intracellular keratinization, which are of limited value for predicting site of origin when metastasis happens. P63 is a master regulator of the development and maintenance of epithelium, working as the recommended pan-SCC diagnostic marker[20–25], but suffers from low specificity due to its reactivity in a substantial proportion of other tumor types, particularly lymphomas[26,27]. Traditionally, the treatment for SCCs commonly follows the anatomical divisions; for example, head and neck SCCs (HNSCCs) are treated separately from reproductive regions[10,28]. In addition to autonomous tumor features, patterns of infiltrating immune cell types involved in the tumor microenvironment (TME) have been associated with tumor progression and patient prognosis. However, the global immune landscape is still unknown[29].

The Cancer Genome Atlas (TCGA) studies have generated comprehensive molecular profiles including somatic mutations, copy-number alterations, DNA methylation, RNA/micro-RNA, and a panel of protein expression for SCCs from 5 individual sites, including lung[30], head and neck[31], esophagus[32], cervix[33], and bladder[34]. Those studies identified recurrent mutations in genes associated with cell cycle and apoptosis (TP53, CDKN2A, CCND1, and RB1), RTK signaling (EGFR, FGFR1, and PIK3CA), squamous differentiation (TP63, SOX2, and NOTCH1), and chromatin remodeling (KMT2C and KMT2D). Moreover, RNA profiling identified subgroups and highlighted pathways thought to be active in these groups[35], but targeting these pathways has largely been unsuccessful. A potential explanation is that these mechanisms could not reflect the functional effects, as they reside many regulatory layers away from the protein. Thus, the proteome, shaped by these genomic and transcriptomic alterations, representing tumor progression and infiltration of immune cells, has potential vulnerabilities that can be therapeutically exploited. Although valuable, in-depth coverage of functional proteome information for SCCs is still lacking.

In this work, systemic proteomics analysis uncovered the differences between SCCs and ACs, common and rare SCCs, HPV-positive, and negative SCCs. Proteomic clustering of SCC entities and TME elucidated the tumor initiation mechanism and therapeutic strategies, respectively. Further analysis of the proteomics data constructed a diagnostic classifier containing 19 proteins, and the diagnostic value of PRKCE, SLC27A1, and CPXM2 were validated. We seek to demonstrate that the systematic proteomic study leads to functional insights that will help drive translational efforts.

## Results

**Proteomic analyses of 17 SCCs.** The present study assembled a cohort of 333 primary tumor samples from treatment naïve SCC patients, including ten common SCC sites (Fig.1a, Supplementary

Data 1, Supplementary Fig. 1a): nasopharynx (20), oral cavity (22), throat (20), skin (20), esophagus (20), lung (20), cervix (21), penis (22), vagina (21), perineum (20); and seven rare SCC sites: thyroid (13), thymus (21), breast (20), pancreas (21), gallbladder (20), bladder (22), anus (10). A mass spectrometry (MS)-based label-free quantification strategy was adopted in the proteomics study, and a tissue microarray (TMAs) based immunohistochemistry (IHC) strategy was carried out on samples in the validation study. All samples were sourced from Zhongshan Hospital, Fudan University. A schematic of the experiment design is shown in Fig. 1a. Our study, therefore, provides a systemic proteomic characterization of SCCs.

The clinical and pathological characteristics, including gender, age, tumor differentiation, keratinization, cell nest size, cell size, mitotic figures, stroma ratio, stromal inflammation, and cancer inflammation, were evaluated and summarized for all cases (Fig. 1b and Supplementary Fig. 1b–r, Supplementary Data 1). Estimate score, immune score, and stromal score were calculated by ESTIMATE[36]. The average length of follow-up is 32 months (3–160 months). For SCCs with a low incidence, 68 patients with no outcome information. These patients were also included in this work but not included in survival analysis. The overall survival (OS) and disease-free survival (DFS) of pan-SCC cohort showed significant differences across organs (OS: log-rank test, $p < 0.0001$; DFS: log-rank test, $p < 0.0001$; Supplementary Fig. 1s). Upon multivariate analysis, both OS and DFS were associated with age (OS, $p < 0.0001$; DFS, $p < 0.0001$) and stage (OS, $p = 0.0083$; DFS, $p = 0.01$). SCC origin is not significant concerning to OS and DFS (Supplementary Data 1). The level of differentiation and keratinization varied across 17 SCCs ($p < 0.0001$). SCCs located in the thyroid, pancreas, and gallbladder had smaller cell nest size than other organs ($p < 0.0001$). The stromal score by ESTIMATE analysis showed a consistent trend with the stromal ratio by pathological evaluation (Spearman correlation, $R = 0.31$, $p < 0.001$; Supplementary Fig. 1t). The diverse spectrum of immune score and estimate score motivated us to explore the immune microenvironment across SCCs (Fig. 1b).

For proteomic analysis, we dissected SCC regions with >80% tumor cells and followed with high-resolution LC-MS/MS analysis on the timsTOF Pro mass spectrometers (Fig. 1a). A spearman's correlation coefficient was calculated for all quality-control runs of 293T cell and repeated samples (Supplementary Fig. 2a, b). The average correlation coefficients were 0.90 and 0.92 for the quality-control samples and repeated samples respectively, demonstrating the consistent stability of the MS platform. The Spearman's correlation for all 333 samples were between 0.56 and 0.99 (median = 0.74) (Supplementary Fig. 2c), showing a high correlation within cancer types. Processed data tables are available in Supplementary Data 1, raw data are available via the ("Methods"). Overall, 14,840 protein groups were quantified (with 1% false discovery rate (FDR) on the peptide and protein levels, intensity higher than 500) (Supplementary Fig. 2d–f). On average, the SCC proteome had 8120 protein groups per sample, ranging from a minimum of 6261 in thyroid to a maximum of 9296 in thymus and showing coverage differences among 17 SCCs (Kruskal–Wallis test, $p < 0.0001$; Fig. 1c), and 5648 proteins were present in all 17 SCCs (Supplementary Fig. 2g). All downstream statistical analyses were performed upon further data filtration to retain only proteins identified in at least 1/3 of the samples (Supplementary Fig. 2h; Supplementary Data 1). These lists, of 14,598 proteins (Pro 2, Supplementary Fig. 2h) in total, included 229 phosphatases, 318 kinases, 1723 membrane proteins, 489 oncogenes, 520 tumor suppresser genes, 918 transcription factors, and 1605 drug targets, indicating sufficient coverage for analysis of intracellular processes.

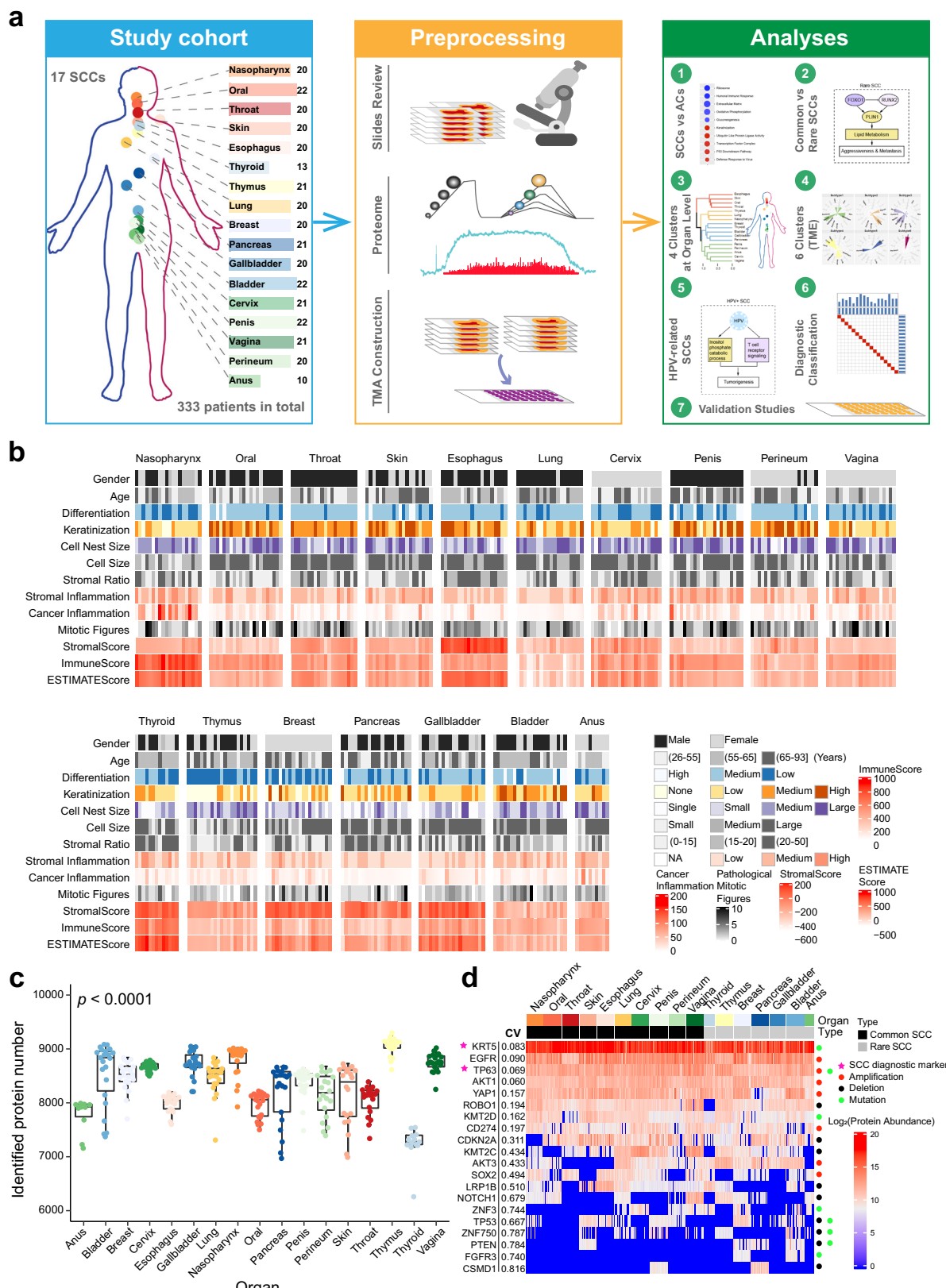

We listed the proteomic data of key molecules in Fig. 1d, which were highlighted in previous genomics researches or served as diagnostic markers[37]. KRT5 (CV of 333 samples: 0.083) and TP63 (CV of 333 samples: 0.069), as known SCC diagnostic markers[25], showed a high and ubiquitous expression level among all SCCs. EGFR and CD274 (PDL1), serve as limited therapeutic targets for SCCs[38–42], presented similar expression level across all SCCs. Frequently copy-number gain gene *AKT1* expressed ubiquitously (CV of 333 samples: 0.060) in all cases, whereas *YAP1*, *AKT3*, and *SOX2* exhibited different extent loss of expression on protein level (AKT1, YAP1, AKT3, and SOX2), indicating differentially activated signaling in 17 SCCs. Similarly, frequently mutated

**Fig. 1 Proteomics of pan-squamous cell carcinoma (SCC) cohort. a** Overview of the proteomics workflow involving pan-SCC cohort, preprocessing, and analyses. In the preprocessing step, Haematoxylin and eosin (H&E) stained slides were examined and evaluated, a mass spectrometry (MS)-based label-free quantification strategy was adopted in the proteomics study, and a tissue microarray (TMAs) was constructed. **b** The study cohort included 333 SCC patients of 17 organs. Clinicopathological parameters were included in the heatmap. See also Supplementary Fig. 1a–1s and Supplementary Data 1. **c** Number of proteins quantified in each SCC patients (Kruskal–Wallis test, $p < 0.0001$). $n_{anus} = 10$, $n_{bladder} = 22$, $n_{breast} = 20$, $n_{cervix} = 21$, $n_{esophagus} = 20$, $n_{gallbladder} = 20$, $n_{lung} = 20$, $n_{nasopharynx} = 20$, $n_{oral} = 22$, $n_{pancreas} = 21$, $n_{penis} = 22$, $n_{perineum} = 20$, $n_{skin} = 20$, $n_{throat} = 20$, $n_{thymus} = 21$, $n_{thyroid} = 13$, and $n_{vagina} = 21$ biologically independent samples examined. Data are expressed as mean values ± SEM. The boxes indicate the interquartile ranges, and no outliers are shown. **d** The protein abundance of SCC diagnostic markers and highly variant genes, the corresponding coefficient of variation (CV) for each marker among 333 SCCs was labeled on the left side. Source data are provided as a Source Data file.

tumor genes, including *TP53*, *PTEN*, and *FGFR3*, showed differential loss of expression on protein level (p53, PTEN, and FGFR3) across 17 SCCs. We also observed the expression of tissue-specific proteins were globally lost by exploring and referring the human protein atlas (https://www.proteinatlas.org/humanproteome/tissue/tissue+specific, Supplementary Fig. 3), indicating high tumor purity of the pan-SCC cohort. These results affirmed the high quality of our proteomic data and our study has so far established a systematic proteomic landscape of Chinese SCCs.

**Proteomic characteristics of SCCs compared with ACs.** SCC and AC are the most common histological cancer subtypes. The initial bioinformatic analysis examined the functional differences between SCCs and ACs by comparing 333 SCCs to an independent AC cohort, consisting of 69 AC patients of 7 sites: breast (8), thyroid (10), lung (11), gastric (12), pancreas (8), gallbladder (12), and colorectum (8) (Fig. 2a). To explore the proteomic coverage of SCCs and ACs, we firstly performed a pairwise comparison of identified protein groups between SCCs and ACs of colorectum (anus in SCC cohort), breast, gallbladder, lung, pancreas, and thyroid after batch effect removal (Supplementary Fig. 4, 5). Venn diagrams showed that overlap of quantified proteins in each comparison accounted for the majority of the total, indicating the comparability of each group (Supplementary Fig. 5a). Furthermore, the principal-component analysis (PCA) (Supplementary Fig. 5b) and correlation analysis (Supplementary Fig. 5c) of these six pairs of SCCs and ACs showed various correlation features of the proteome of SCCs and ACs from different tissues.

Among the 14,598 proteins identified in SCCs, 5130 proteins were commonly expressed in all 17 SCCs (Pro 3, Supplementary Fig. 2h). Meanwhile, 4845 proteins were commonly expressed in all 7 ACs among 10,414 proteins identified in ACs (Supplementary Data 2). With the purpose of comparing the differences between ACs and SCCs, we combined these two protein lists into one dataset based on Uniprot IDs, and batch effects were removed (Fig. 2a and Supplementary Fig. 4). In total, 1538 proteins showed significant differential expression (Wilcoxon rank-sum test, BH-adjusted $p < 0.05$, fold change > 2; Supplementary Data 2), with 643 proteins overexpressed in ACs and 895 proteins overexpressed in SCCs (Fig. 2b, c). Gene set enrichment analysis (GSEA) demonstrated that AC-enriched proteins were significantly enriched in pathways (one-side Fisher's exact test, BH-adjusted $p < 0.05$), including the ribosome (such as RPS27A, RPL4, and RPL6), humoral immune response (such as C3, FGB, and C1R), extracellular matrix (ECM) (such as ANXA2, FN1, and S100A10), oxidative phosphorylation (such as SDHA, HADHB, and NDUFS2), and gluconeogenesis (such as ALDOC, ENO1, and GAPDH), whereas proteins enriched in SCCs were mainly involved in pathways related to keratinization (such as KRT13, DSC3, and TGM1), ubiquitin-like protein ligase activity (such as ANAPC5, MED21, and PPP1R11), transcription factor complex (such as AJUBA, RUNX3, and GTF2H1), p53 downstream pathway (such as TP63, BAK1, and SERPINE1), and defense

response to virus (such as APOBEC3F, CGAS, and EIF2AK4) to be dominant in SCCs (Fig. 2d; Supplementary Data 2). These observations revealed that the significant difference between SCCs and ACs was in line with the property of originating epithelial tissues.

To investigate how these significantly differentially expressed proteins (DEPs) affected prognosis, we then tested the prognostic power (multivariate Cox proportional hazard model for pan-SCC cohort, Kaplan–Meier survival curve with log-rank test for nine TCGA cancer cohorts; BH-adjusted $p < 0.05$; Supplementary Fig. 6a; Supplementary Data 2) of DEPs in these pathways in our dataset and nine TCGA datasets, including 4 SCCs (head and neck, esophagus, lung, and cervix) and 5 ACs (breast, lung, pancreas, colon, and cervix). Under this strict analysis, RPL12 served as a good prognostic marker ($p = 0.036$), and SERPINE1 as an adverse prognostic marker ($p = 0.0135$) in pan-SCC cohort (Supplementary Fig. 6b). Strikingly, DEPs in ECM (such as COL4A1, FN1, and PKM) and glucose metabolism (such as SDHA, LDHA, and ENO1) of AC-enriched pathways showed consistently poor prognostic values in both ACs and SCCs, indicating the tumor metabolic status and microenvironment may promote tumorigenesis and progression regardless of tumor histological types (Fig. 2e). By contrast, DEPs in keratinization (such as KRT13, DSC3, and PKP1) showed opposite roles, displaying favorable prognostic values in SCCs, while unfavorable in ACs (Fig. 2e). Additionally, the prognosis of lung and pancreatic adenocarcinomas, which were two cancers with poor prognosis[43–45], were significantly affected by these DEPs in ECM, glucose metabolism, and keratinization (Fig. 2e).

To further elucidate the underlying mechanism of differences between ACs and SCCs, we constructed a computational model for the pathways shown in Fig. 2e using the GENEMINIA[46] to predict kinases and transcriptional factors (TFs) interactions that may target these DEPs. As shown in Fig. 2f, DEPs enriched in ACs involved in ECM and glucose metabolism were mediated by an 8 kinases-8 TFs network, representatively mediated by PRKAA2/PKM-ENO1 and PKM-HNRNPD/YBX1 axes. Keratinization was mediated by an 11 kinases-5 TFs network, predominantly mediated by CHUK-TP63/IRF6 axis. We also tested the prognosis of these kinases-TFs in ACs and SCCs. Consistently with the findings in these DEPs, we found that PKM-ENO1 showed a consistent poor prognosis value ($p < 0.05$) in pancreatic adenocarcinoma and HNSC/ESCC, whereas VRK2-TP63 showed a poor prognostic value ($p < 0.05$) in pancreatic adenocarcinoma and a good prognostic value in ESCC/LUSC (Supplementary Fig. 6c). Collectively, these pathways, DEPs, and kinase-TF networks over-represented in SCCs or ACs represented the unique or shared biological and clinicopathological features, further highlighting their clinical implications.

**Proteomic features in rare SCCs compared with common SCCs.** To investigate how pathological parameters (common versus rare, differentiation, keratinization, and cell nest size) affect the proteome and signal transduction, we examined DEPs (Wilcoxon

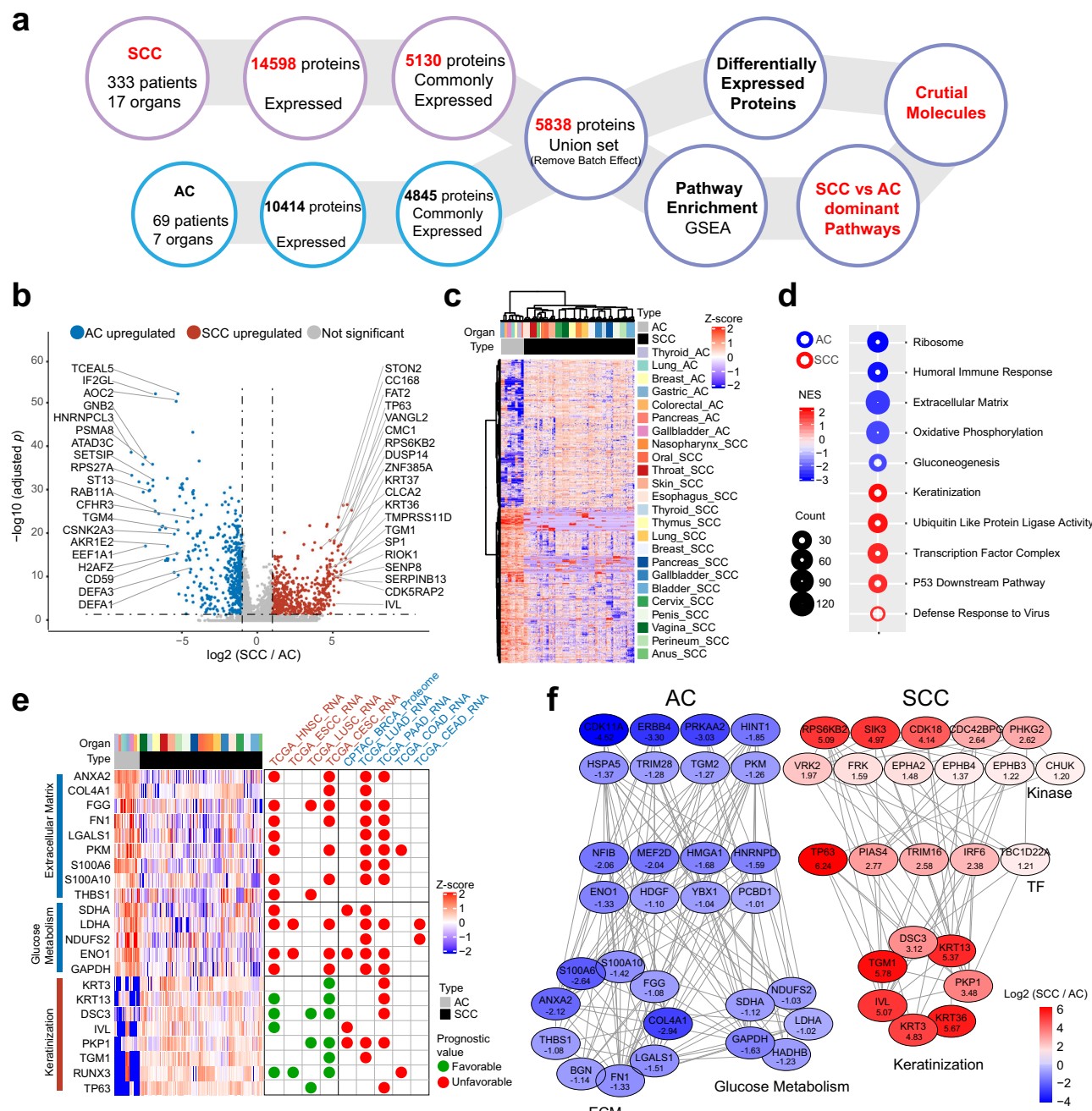

**Fig. 2 Proteomic differences of pan-SCCs and pan-ACs. a** The workflow for differential analysis between pan-SCCs and pan-ACs. A total of 5838 proteins commonly expressing in SCCs and/or ACs was obtained and differentially expressed proteins were calculated (Wilcoxon rank-sum test), and Gene set enrichment analysis (GSEA) was performed. **b, c** A volcano plot and a heatmap showed the results of a two-sided Wilcoxon rank-sum test (BH-adjusted $p < 0.05$, fold change > 2) comparing pan-ACs and pan-SCCs. Significantly differentially expressed proteins were labeled in the volcano plot. **d** GSEA revealed the pathways that were significantly enriched in pan-ACs and pan-SCCs. NES, normalized enrichment score. **e** The prognostic value of differentially expressed proteins in pan-AC-enriched pathways (extracellular matrix and glucose metabolism) and pan-SCC enriched pathway (keratinization) by exploring pan-SCC cohort (this study) and 9 TCGA datasets. *P* values of pan-SCC cohort were from multivariate Cox proportional hazard model (including age, gender, TNM stage, protein expression, organ, and histology), and *p* values for 9 TCGA datasets were from log-rank test. **f** The regulation network of kinases, TFs, and DEPs in pan-ACs and pan-SCCs. All kinases, TFs, and DEPs were colored by log2 (SCC/AC).

rank-sum test, BH-adjusted $p < 0.05$, fold change > 2) in patients between common versus rare, low differentiation versus medium/high differentiation, medium/low keratinization versus high keratinization, and small tumor nest versus medium/large tumor nest (Fig. 3 and Supplementary Fig. 7). High differentiation level and high keratinization showed very similar over-represented pathways (one-side Fisher's exact test, BH-adjusted $p < 0.05$), including

keratinization, matrisome, and intermediate filament cytoskeleton (Supplementary Fig. 7a, b). Patients with low keratinization were elevated in pathways such as mitochondrion, cellular amino acid metabolic process, and mRNA processing (Supplementary Fig. 7b). RNA processing was recently reported to be related to head and neck SCC oncogenesis because copy-number drivers were involved in this pathway[47], indicating that this event may occur more frequently in

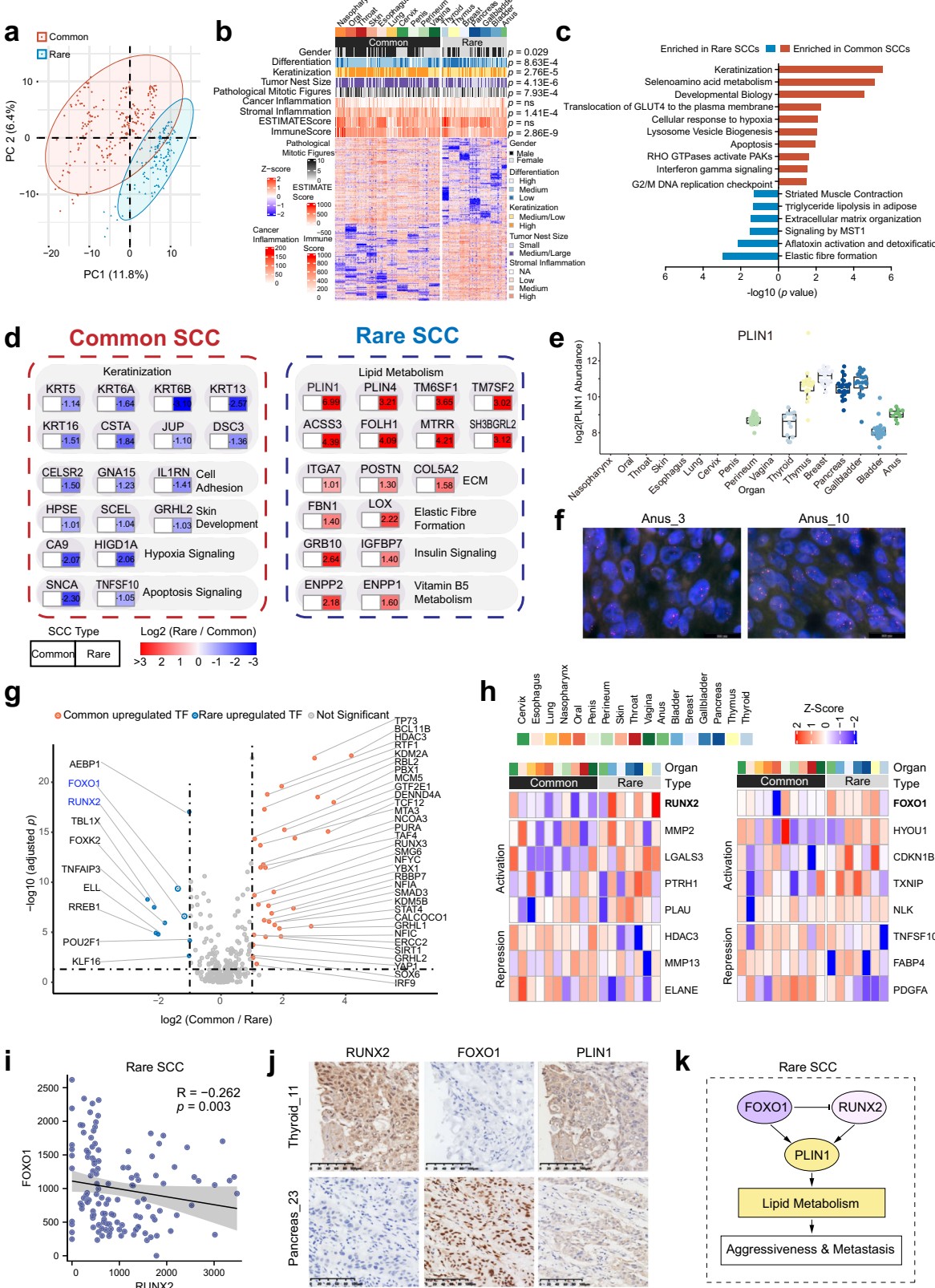

low keratinization samples. Cell nest size was engaged in a novel histopathological grading system for lung, oral, and esophageal SCCs, being recognized as a prognostic morphological parameter that small nests associated with unfavorable prognosis[48]. Large cell nests proteome over-represented in chromatin organization, dephosphorylation, catalytic complex, sequence-specific DNA binding, and phosphorus containing groups transferase activity transferring, whereas small cell nests proteome over-represented in cell surface, peptide metabolic process, endoplasmic reticulum, peptidase activity, and matrisome (Supplementary Fig. 7c). Small cell nests overrepresented with a higher metabolic and protein synthesis phenotype, which was similar to low keratinization.

**Fig. 3 Proteomic differences of common SCCs and rare SCCs. a** Principal-component analysis (PCA) of common SCCs and rare SCCs. **b** The association of common and rare SCCs with 9 variables (two-sided Fisher's exact test was used for categorical variables and two-sided Wilcoxon rank-sum test was used for continuous variables), and the heatmap of significantly DEPs (Wilcoxon rank-sum test, BH-adjusted $p < 0.05$, fold change > 2) in common SCCs and rare SCCs. **c** Enriched pathways of significantly DEPs (two-sided Wilcoxon rank-sum test, BH-adjusted $p < 0.05$, fold change > 2) in common SCCs and rare SCCs. **d** Proteins in pathways that were differentially expressed in common SCCs and rare SCCs, and representative DEPs. **e** The PLIN1 protein expression in 17 SCCs. **f** Representative PLIN1 fluorescence in situ hybridization signal patterns (red signals = PLIN1, green signals = CEP15), left, this case (Anus_3) was scored negative for PLIN1 amplification. PLIN1/nucleus ratio = 2.52; right, this case (Anus_10) was scored as positive for PLIN1 amplification. PLIN1/nucleus ratio = 6.2. The boxes indicate the interquartile ranges, and no outliers are shown. **g** Differential expressed TFs (two-sided Wilcoxon rank-sum test, BH-adjusted $p < 0.05$, fold change > 2) in common SCCs and rare SCCs. The two TFs, RUNX2 and FOXO1, were labeled in blue. **h** The expression heatmap of downstream transcriptional targeted genes (TGs) regulated by RUNX2 and FOXO1 (in bold). The expression level was scaled by row. **i** A scatterplot showed the association between the protein abundance of RUNX2 ($x$-axis) and FOXO1 ($y$-axis). Pairwise Spearman correlation. **j** Immunohistochemistry staining for RUNX2, FOXO1, and PLIN1 expression in rare SCCs (one case of thyroid SCC and one case of pancreatic SCC) was concordant with the mass spectrometry findings. Scale bar, 100 μm. **k** Diagram depicted our hypothesis of lipid metabolism upregulation contributing to rare SCC aggressiveness and metastasis. Source data are provided as a Source Data file.

To obtain a general insight into the distinction between common SCCs and rare SCCs, we compared the proteomic profiles of these two groups. PCA analysis revealed a clear difference between the common and rare SCCs, indicating a significant proteomic difference during the development and progression of SCCs (Fig. 3a). A total of 6213 proteins (Pro 4, Supplementary Fig. 2h) were used to do a comparison, and a total of 938 proteins were differentially expressed between common and rare SCCs (Wilcoxon rank-sum test, BH-adjusted $p < 0.05$, fold change > 2; Supplementary Data 3). Among them, 500 proteins were upregulated in common SCCs, and 438 proteins in rare SCCs.

Associations between SCC types and clinicopathological characteristics (Fisher's exact test was used for categorical variables and Wilcoxon rank-sum test was used for continuous variables) for the 333 patients were shown in Fig. 3b. Common/rare SCC types correlated with gender ($p = 0.029$), differentiation ($p = 8.63e{-}4$), keratinization ($p = 2.76e{-}5$), tumor nest size ($p = 4.13e{-}6$), pathological mitotic figures ($p = 7.39e{-}4$), stromal inflammation ($p = 1.41e{-}4$), and immune score ($p = 2.86e{-}9$). Rare SCCs are positively correlated with worse pathological characteristics, including poor differentiation, lower keratinization, smaller cell nest size, lower stromal inflammation, and lower immune score. However, less mitotic figures were unexpectedly found in rare SCCs (Fig. 3b).

Pathway enrichment analysis by Reactome (https://reactome.org/) demonstrated that common SCCs were significantly enriched in pathways including keratinization (such as KRT5, KRT6B, and CSTA), cell adhesion (such as CELSR2, GNA15, and IL1RN), skin development (such as HPSE, SCEL, and GRHL2), and apoptosis signaling (such as SNCA and TNFSF10). Proteins enriched in rare SCCs were mainly involved in pathways related to lipid metabolism (such as PLIN1, PLIN2, and TM7SF2), ECM organization (such as ITGA7, POSTN, and COL5A2), elastic fiber formation (such as FBN1 and LOX), insulin signaling (such as GRB10 and IGFBP7), and vitamin B5 metabolism (such as ENPP1 and ENPP2) (Fig. 3c, d, Supplementary Data 3). Particularly, PLIN1 was highly expressed in rare SCCs and was not detected in the most of common SCCs (188/288) (Fig. 3e). Then, we tested the PLIN1 copy number in ten cases for each SCC. Interestingly, we detected gene amplification in 3 anal SCCs by fluorescence in situ hybridization analysis (3/10, Fig. 3f, Supplementary Data 3), indicating a potential mechanism for PLIN1 high expression in rare SCCs.

Considering that TFs play essential roles in carcinogenesis and aggressiveness, we focused on the differentially expressed TFs between the common and rare SCCs (Wilcoxon rank-sum test, BH-adjusted $p < 0.05$, fold change > 2). Among them, 10 TFs were overexpressed in rare SCCs (such as AEBP1, FOXO1, RUNX2, TBL1X, and FOXK2) and 33 in common SCCs (such as TP73,

BCL11B, HDAC3, RTF1, and KDM2A) (Fig. 3g). By TF-TG enrichment analysis, two TFs (RUNX2 and FOXO1) were overexpressed in rare SCCs, regulating the majority of TGs, indicating the dominant biological function in rare SCCs (Fig. 3h). RUNX2 regulating skeletal morphogenesis and FOXO1 regulating the cellular response to oxygen levels participated in lipid metabolism. Interestingly, it is reported[49] that FOXO1 could suppress the transcriptional activity of RUNX2. We also found a negative correlation between the protein abundance of RUNX2 and FOXO1 in rare SCCs (Spearman correlation, $R = -0.262$, $p = 0.003$; Fig. 3i). Anti-RUNX2, FOXO1, and PLIN1 IHC validated evidence of negative correlated RUNX2 and FOXO1(only RUNX2 positive in thyroid_11, and only FOXO1 positive in pancreas_23), and both two cases showed positive staining of PLIN1(Fig. 3j). Together, these data indicated that lipid metabolism was possibly more active and played critical roles in squamous cell differentiation in rare SCCs (Fig. 3k). This insight reveals an opportunity by staining RUNX2, FOXO1, and PLIN1 to diagnose rare SCCs.

**Proteome-based hierarchical clustering of 17 SCCs.** Genomic and transcriptomic information has been used to cluster SCCs into subgroups[35]; however, in-depth coverage of proteome-based clustering is still lacking. As expected, the t-SNE analysis showed that samples from the same organ tended to cluster together, indicating more similarity in SCC-originated tissue types (Fig. 4a). To identify a proteome signature-based classification, we conducted a hierarchical clustering of 333 tumor samples of 17 SCCs. Consequently, we identified four proteomic subtypes based on 1500 most variable proteins (Pro 5, Supplementary Fig. 2h) that were significantly associated with anatomical sites and revealed distinguished patterns of protein expression and signal transduction (Fig. 4b and Supplementary Fig. 8a, Supplementary Data 4).

Cluster 1 (EOST, named by initials of each organ and ordered alphabetically), included esophagus, skin, oral, and throat, was associated with cytoskeleton function and immune, such as cytoskeletal protein binding, KRAS signaling down, complement, and innate immune system (Fig. 4b, c). Cluster 2 (LNT) included thymus, lung, and nasopharynx, was characterized by the highest level of aerobic oxidation, including mitochondrion, phospholipid metabolism, oxidative phosphorylation, and TCF-dependent signaling. Cluster 3 (BBGPT) consisted breast, thyroid, bladder, gallbladder, and pancreas, which were all rare SCCs. It was enriched in pathways of ECM glycoproteins, fatty acid metabolism, and inflammatory response thus had a very similar characteristic with ACs (Figs. 2d and 4d). BBGPT showed the worst prognosis among these 4 clusters (Supplementary Fig. 8b, c). Cluster 4 (ACPPV) were all anogenital SCCs, including penis,

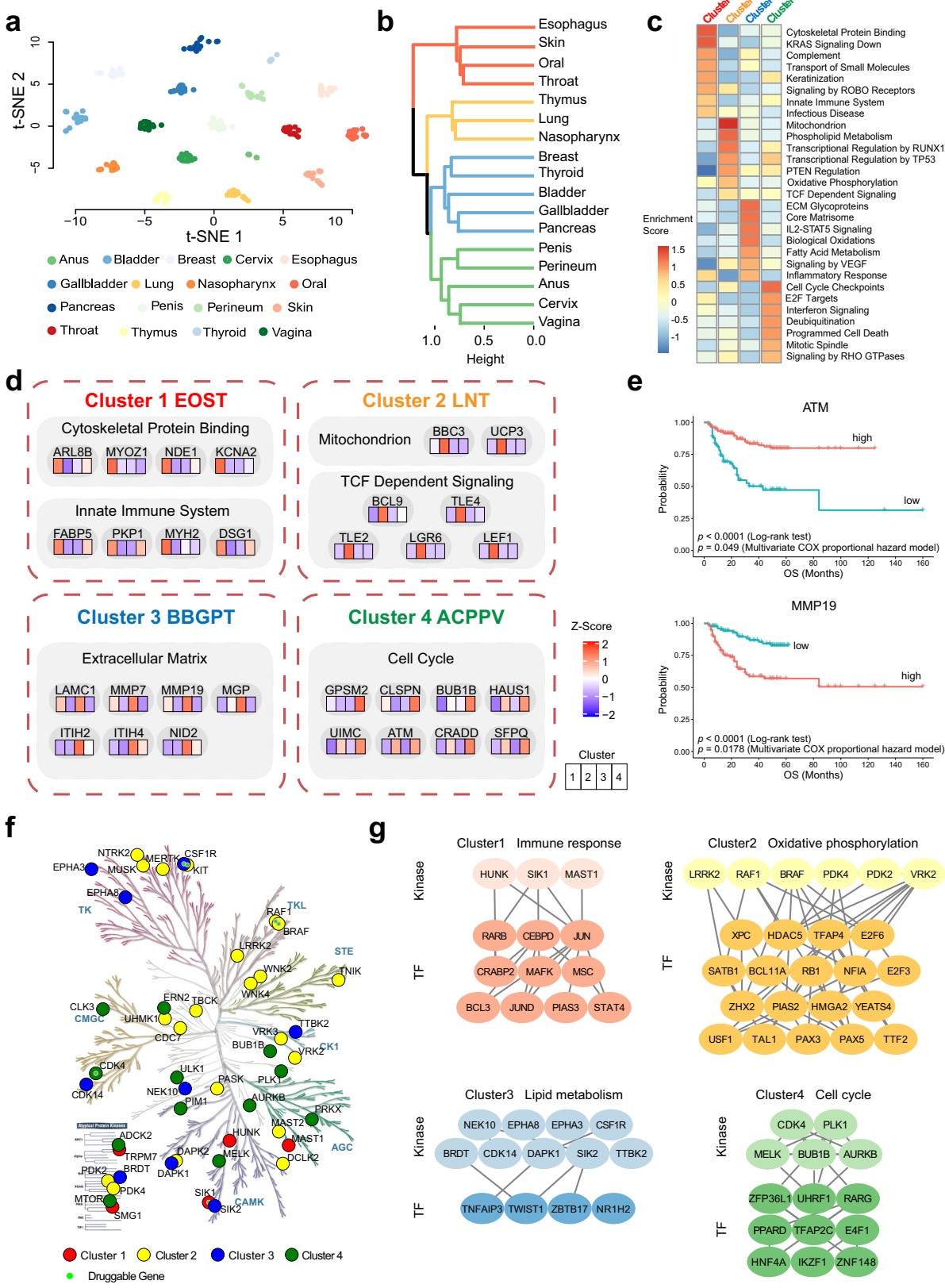

perineum, anus, cervix, and vagina, with the best prognosis (Supplementary Fig. 8b, c). Notably, ACPPV with a high HPV infection frequency was enriched for cell cycle checkpoints, E2F targets, and programmed cell death. Furthermore, we calculated the prognostic values of all DEPs of the 4 clusters (multivariate Cox proportional hazard model), and two proteins were

found with prognostic values among these DEPs. The protein ATM ($p = 0.049$) was associated with a favorable prognosis, and MMP19 ($p < 0.0001$) was associated with an unfavorable prognosis (Fig. 4e).

To elucidate the underlying differential SCCs initiation mechanism, we matched the kinases from all DEPs of these 4

**Fig. 4 Proteomic clusters of pan-SCCs and associations with SCC initiation. a** T-SNE plot of 17 SCC samples, color coded for SCC originating organs. **b** Hierarchical clustering analysis of proteomics for 333 SCC samples. Color differences in the dendrogram indicated 4 clusters that were resolved by multiscale bootstrapping. **c** GSEA revealed the pathways that were significantly enriched in the four proteomic clusters respectively. **d** Representative pathways and corresponding significant DEPs (Kruskal–Wallis test, BH-adjusted $p < 0.05$, fold change > 2) in the four proteomic clusters, cluster names were named by initials of each organ and ordered alphabetically. **e** Kaplan–Meier survival curves for ATM and MMP19 with $p$ value from multivariate Cox proportional hazard model (including protein expression, age, gender, histology, organ, and stage) labeled. **f** Kinmap (http://www.kinhub.org/kinmap/index.html) of differentially expressed kinases with different colors for the four proteomic clusters. Illustration reproduced courtesy of Cell Signaling Technology, Inc. (www.cellsignal.com). **g** The regulation networks of kinases and TFs in the four proteomic clusters.

clusters and mapped them to the KinMap[50] (http://www.kinhub.org/kinmap/index.html; Fig. 4f, Supplementary Fig. 8d). Specific kinases of Cluster 1 mainly belonged to CAMK and AGC, such as HUNK, SIK1, and MAST1, participating in the immune response. kinases of Cluster 2 were widely distributed in all kinase categories, mainly participating in oxidative phosphorylation (such as LRRK2, RAF1, and BRAF). Specific kinases of Cluster 3 were distributed in TK, CAMK, and CMGC, mainly involved in lipid metabolism (such as NEK10, DAPK1, and SIK2). Specific kinases of Cluster 4 mainly belonged to AGC, CAMK, and CMGC, participating in cell cycle (such as CDK4, PLK1, and AURKB; Fig. 4f, g). We further conducted four Kinase-TF networks using the GENEMINIA for each cluster, and multiple TFs may be regulated by those kinases participating in dominant pathways of each cluster (Fig. 4g and Supplementary Fig. 8e). In summary, these four proteomic clusters revealed anatomical features and potentially indicated differential SCC initiation.

**Immune landscape of SCCs and their potential druggable insights**. To gain insight into immune features in SCCs, we next analyzed the TME composition of all 333 tumors using xCell[51], including gene signatures of deconvoluted immune, stroma, and 64 different microenvironment cell types. Consensus clustering based on inferred cell proportion identified 6 SCC subtypes with distinct TME characteristics and prognosis, were discriminated by the dominant presence of specific cell types and pathways (Fig. 5a, b, and Supplementary Fig. 9; Supplementary Data 5). We defined six subtypes in this Pan-SCC cohort: (1) Classical squamous (ClSq), (2) Fatty acid metabolic (FaSq), (3) Basophils inflamed (BaSq), (4) Neutrophils inflamed (NeSq), (5) Eosinophils inflamed (EoSq), and (6) Immune hot (IhSq), characterized by unique TME signatures and discriminating signaling pathways (Fig. 5a). These molecularly based cell-type classifications were supported by histopathological assessment (Fig. 5c, d). PLIN1 positive immunobiological staining was detected in the FaSq subtype (Fig. 5d).

The ClSq subgroup, containing a mixture of bladder (21), esophageal (13), skin (13), oral (11), anus (10), pancreas (4), breast (2), gallbladder (1), and perineum (1) SCCs. Not only with enrichment of cluster EOST, anus and the majority of the bladder SCCs fell into ClSq. The possible reason is that bladder (such as chronic S hematobium infection) and anus [such as human papilloma virus (HPV) infection] are affected by external stimulus, which are similar as aerodigestive SCCs[52,53]. ClSq was characterized by a high degree enrichment of epithelial cells, sebocytes, keratinocytes, and fibroblasts. Notably, ClSq tumors also revealed the presence of immune inhibitory cells such as regulatory T cells. Proteomic analysis showed upregulated pathways involved in signaling by FGFR3, leukocyte transendothelial migration, and complement (Fig. 5b). In addition, ClSq tumors showed upregulation of cell–cell tight junction that provide barrier functions for epithelium, suggesting a mechanical barrier against immune cell infiltration[54,55].

The FaSq tumors uniquely and mainly contained thymus (21) and thyroid (13) SCCs, and a small number of breast (4), vaginal

(3), skin (3), and pancreatic (2) SCCs, were characterized by upregulation of multiple fatty acid-related pathways, including metabolism of water-soluble vitamins, mitochondrial protein import, mitochondrial fatty acid beta oxidation, and fatty acid metabolism. As a subset of rare SCCs, FaSq tumors may rely more on fatty acid as cellular building blocks for membrane formation, energy storage, and the production of signaling molecules, and targeting fatty acid metabolism might be more selective for this subtype of SCCs (Fig. 5b).

Granulocyte infiltration reflects a state of host inflammation[56]. Three subgroups of SCCs were characterized by basophils, neutrophils, and eosinophils inflamed individually. The BaSq tumors, containing penis (18), gallbladder (18), pancreatic (15), lung (6), perineum (4), skin (4), oral (4), bladder (1), and breast (1) SCCs, were characterized by a high degree enrichment of basophils, and upregulated pathways involved in KRAS signaling up, insulin processing, signaling by PDGF, and hypoxia. The NeSq tumors, containing gynecological (18 vaginal and 21 cervical cases), breast (13), lung (13), esophageal (7), oral (7), penis (4), and gallbladder (1) SCCs, were in a high degree enrichment of mv endothelial cells, ly endothelial cells, and neutrophils. Proteomic analysis showed upregulated pathways involved in DNA double-strand break repair, NOTCH signaling pathway, cell cycle, and apoptosis. The EoSq tumors, containing perineum (14) and throat (20) SCCs, enriching in eosinophils, MSC, and NKT cells, involving in HSP90 chaperone cycle, estrogen response late, regulation of autophagy, and UV response up (Fig. 5b). Multiple studies have shown an improved prognosis with tumor-associated tissue eosinophilia in various types of SCCs, including oral SCCs[57,58], esophageal SCCs[59], nasopharyngeal SCCs[60], and penile cancer[61].

The IhSq tumors, containing 20 nasopharyngeal SCCs and one perineum SCC, were distinguished from other 5 subtypes by their stronger signatures for a high degree of B cells, CD4+ T cells, and M1 macrophages. The proteome was characterized by upregulation of multiple oncogenic, immune-related, and signaling pathways including regulation of KIT signaling, B cell receptor signaling, interferon gamma response, and oxidative phosphorylation, and thus may benefit from immunotherapy (Fig. 5b).

The TME not only plays an important role in tumor progression, but it is also a potential treasury for finding therapeutic options by targeting non-tumor stromal cells or the interaction between tumor cells and stromal cells. To examine the utility of these subtypes in guiding treatment selection, we considered druggability (drugbank version 5.1.5) and expression advantage in each subgroup and found considerable drug targets for each subgroup (Fig. 5e, Supplementary Data 5). Furthermore, as the microenvironment-targeted strategies mainly include inhibition of the extracellular interactions, we summarized membrane proteins on tumor cells that can be targeted (Fig. 5e, genes in bold). Ion channels comprise an attractive tool for targeted therapy for cancer[62], we found ion channel drug targets in 5 subtypes, including sodium channel (SCN4B in ClSq and SCN5A in IhSq), potassium voltage-gated channel (KCNQ4 in BaSq, KCNA5 in NeSq, and KCND2 in IhSq), and voltage-

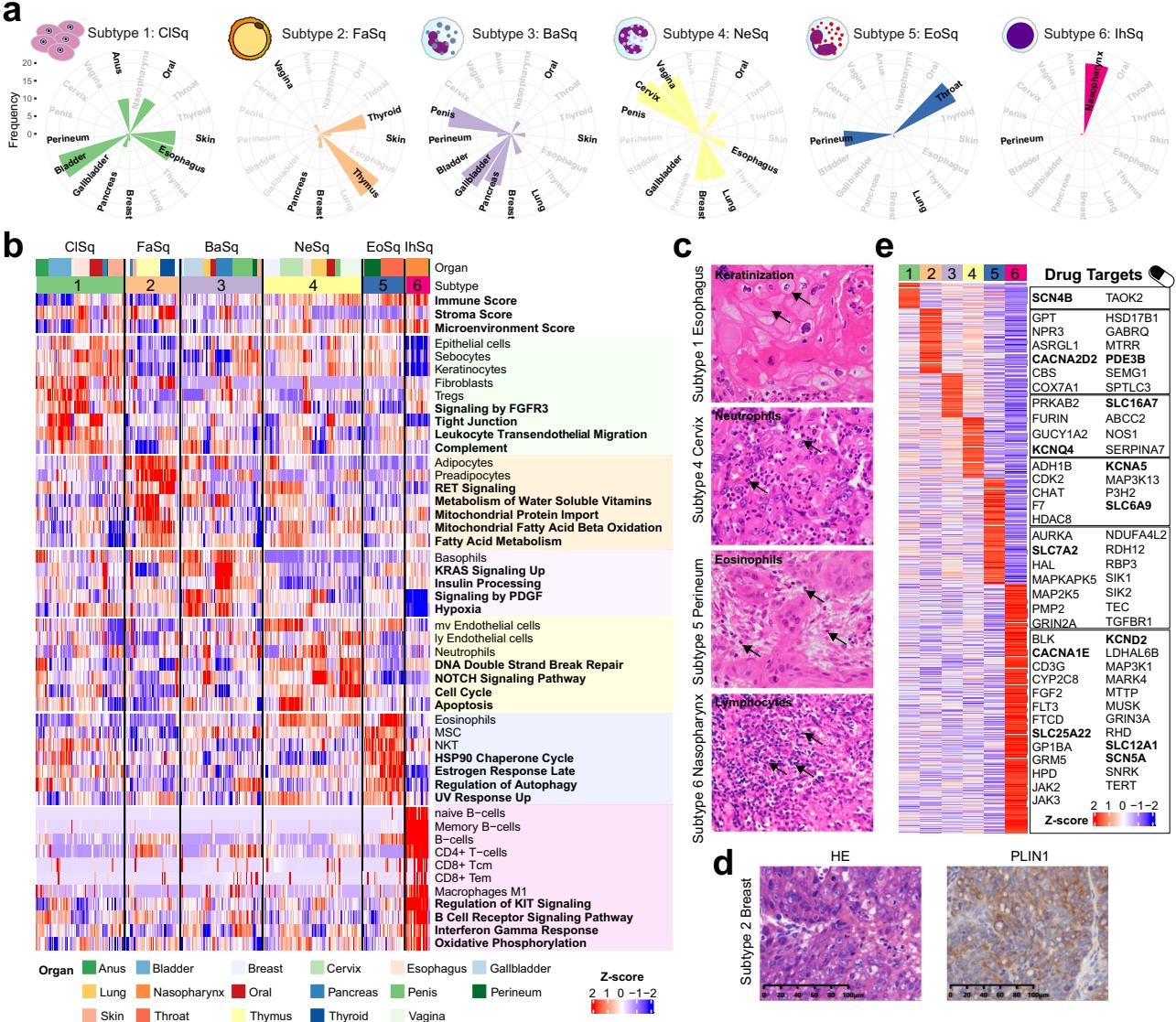

**Fig. 5 Immune-based subtyping of pan-SCCs. a** Coxcomb diagrams showing the distribution of 17 SCCs in 6 subtypes, including (1) Classical squamous (ClSq), (2) Fatty acid metabolic (FaSq), (3) Basophils inflamed (BaSq), (4) Neutrophils inflamed (NeSq), (5) Eosinophils inflamed (EoSq), and (6) Immune hot (IhSq). **b** Proteome-based microenvironmental cell signatures and over-represented pathways in 6 subtypes. **c** Represented morphologies of SCCs with specific tumor microenvironment cell infiltrating in 4 subtypes (subtype 1, 4–6). Arrows depict the specific cell types. Basophils were not shown because they cannot be recognized by HE staining. Scale bar, 100 μm. **d** Haematoxylin and eosin (H&E) stained and PLIN1 immunohistochemistry (IHC) images showing one example of subtype 2 samples with suspected lipid droplets. **e** Drug targets in 6 subtypes (drug targets discussed in the text were in bold).

dependent calcium channel (CACNA2D2 in FaSq and CAC1E in IhSq). Solute carrier (SLC) family transporters utilize an electrochemical potential difference or an ion gradient for transporting their substrates across biological membranes, can be therapeutically targeted[63]. We found SLC16A7 (BaSq), SLC6A9 (NeSq), SLC7A2 (EoSq), and SLC12A1 (IhSq) were specifically expressed in four subtypes and could serve as treatment targets for certain subtypes. PDE3B, involved in fatty acid metabolism[64], can be a drug target in FaSq (Fig. 5e). Based on these observations, an approach that assesses the TME properties for treatment selection seems to be warranted.

**Characterization of HPV-related SCCs.** In order to better characterize the influence of HPV as a contributor to part of the SCCs, we tested 15 high-risk HPV types, including 16, 18, 31, 33, 35, 39, 45, 51, 52, 56, 58, 59, 66, 68, and 82 for all cases. Of the

333 core-set tumors, only anogenital SCCs had HPV-positive patients, including anus (infection rate 100%), penis (50%), perineum (70%), cervix (95%), and vagina (95%) (Fig. 6a, Supplementary Data 6).

HPV16, as the prominent type for anogenital SCC patients, is with a positive rate above 54% (Fig. 6b), whereas HPV18 (6.67%) was not as popular as in western countries (~25%)[33,65]. It was reported[33] that HPV18 was enriched in the low keratinization SCCs and adenocarcinomas, which is the possible reason that our cohort had a low infection rate. Cervix and vagina were generally infected by more than one HPV clade, probably due to their anatomic location with high infection affinity. However, multiple HPV infections may affect the protection of effective HPV vaccines[66]. As reported, some persist HPV infections and viral oncogene E6- and E7- expression inactivate *TP53* and *RB*[67]. We found that p53 completely lost expression in the anus, cervix, and vagina SCCs (Supplementary Data 6). Compared to non-

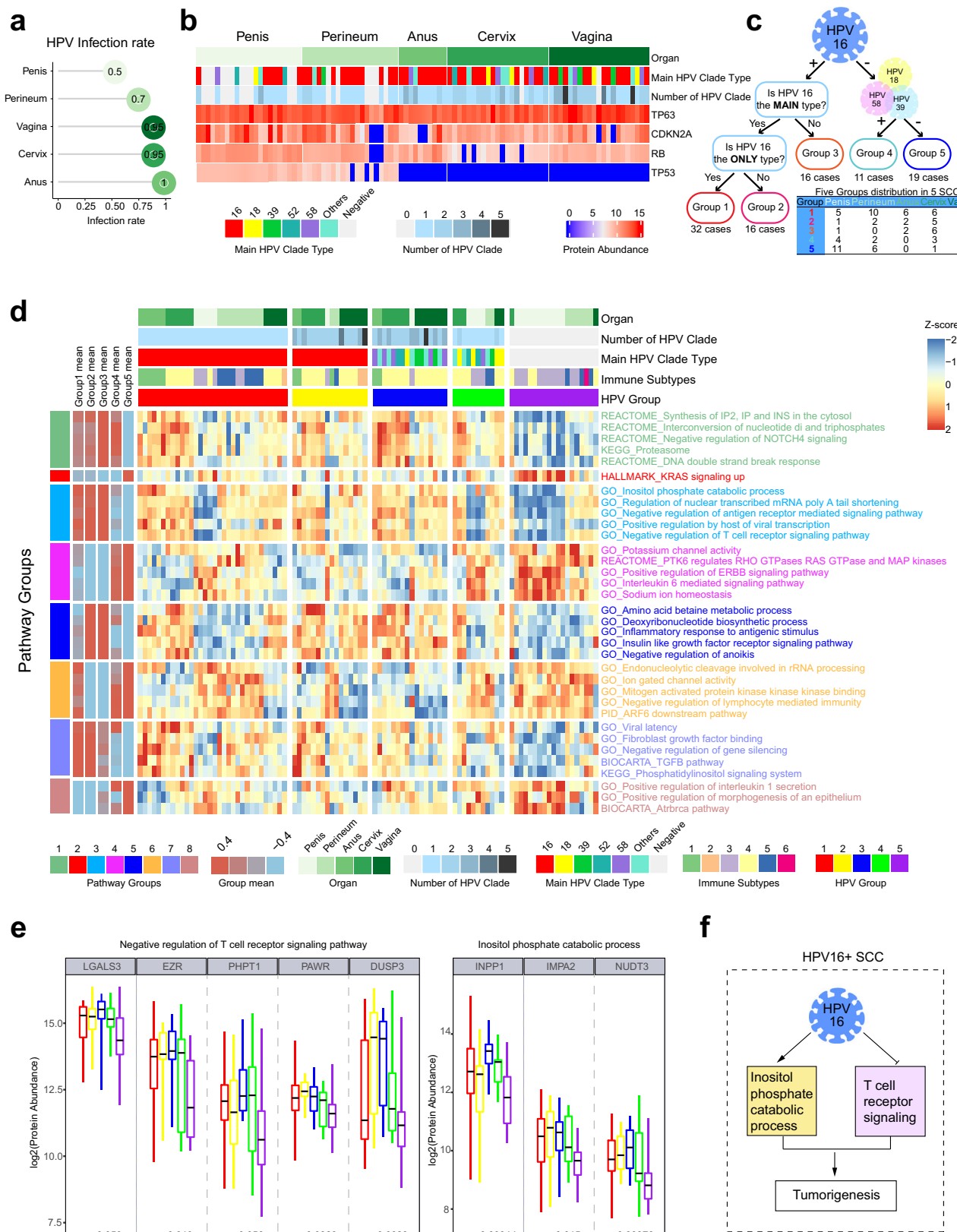

anogenital SCCs, anogenital SCCs showed a lower expression of RB level (Wilcoxon rank-sum test, $p < 0.0001$; Supplementary Fig. 10a). RB showed expression differences among these five anogenital SCCs, and cervical SCC showed the lowest expression among five anogenital SCCs (Kruskal–Wallis test, $p < 0.001$;

Supplementary Fig. 10b). Interestingly, a negative correlation trend was found in cervical SCC (Spearman correlation; $R = -0.255$, $p = 0.265$; Supplementary Fig. 10c). CDKN2A expression was higher in HPV-positive perineum SCC than HPV negative perineum SCC (Fig. 6b and Supplementary Fig. 10d).

**Fig. 6 Characterization of HPV-related SCCs. a** HPV infection rate of 5 anogenital SCCs. **b** Details of HPV infection and protein abundance of known molecules related to HPV infection of 5 anogenital SCCs. **c** Grouping of 5 anogenital SCCs according to HPV16 infection status and 5 groups were obtained. **d** Impact of HPV status on pathways in 5 anogenital SCCs. The heatmaps showed protein-expression derived, differentially regulated pathways associated with differential HPV infection. Pathway groups were defined according to the patterns of differential HPV infection. See also "Methods" and Table S6B. **e** Boxplots showing log2 protein abundance of differentially expressed molecules in two pathways that were over-represented in HPVT16 infected SCCs (Kruskal–Wallis test, BH-adjusted $p < 0.05$). $n1 = 32$, $n2 = 16$, $n3 = 16$, $n4 = 11$, and $n5 = 19$ biologically independent samples examined. Data are expressed as mean values ± SEM. The boxes indicate the interquartile ranges, and no outliers are shown. **f** Diagram depicted our hypothesis of inositol phosphate catabolic process upregulation and negative regulation of T cell receptor signaling contributing to HPV16-related SCC tumorigenesis. Source data are provided as a Source Data file.

To further elucidate the molecular pathways differentially activated or inactivated in HPV16 infected SCCs compared with other types or negative SCCs, we grouped these anogenital SCCs according to the HPV infection patterns (Fig. 6c, Supplementary Data 6). Group 1, HPV16 infection only; Group 2, multiple infection and HPV16 is the main type; Group 3, multiple infection and HPV16 is not the main type; Group 4, HPV infection and not HPV16; Group 5, negative (HPVneg). We identified 8 patterns of differential pathway regulation between these 5 groups (Fig. 6d). Pathways including synthesis of IP2, IP and INS, interconversion of nucleotide di and triphosphates, proteasome, and DNA double-strand break response were higher in HPV-positive SCCs, indicating the high metabolic circumstance. Notably, HPVneg SCCs had a considerable overlap with Basophil subtype, was prominently characterized by KRAS signaling elevation (Fig. 6d). HPV16 positive SCCs, covering Group1-3, showed higher levels of inositol phosphate catabolic process and positive regulation by host of viral transcription, meaning that HPV16 infected cells was more dependent on inositol phosphate metabolism. Meanwhile, HPV16 positive SCCs had negative regulation of both antigen receptor-mediated signaling pathway and T cell receptor signaling pathway, revealing the immunosuppression by HPV16 infection. Contrariwise, interleukin 6 mediated signaling pathway was lower in HPV16 positive tumors, together with PTK6 regulates RHO GTPase RAS GTPase and MAP kinases, positive regulation of ERBB signaling pathway, and ion channel activity (Fig. 6d). Next, we focused on how multiple HPV infection (including HPV16, Group2-3) affected on cell signaling. Similar as HPV-positive SCCs (Group1-3), some metabolic pathways were higher than the other 3 groups. Furthermore, we found that insulin-like growth factor receptor signaling pathway, negative regulation of anoikis and inflammatory response to antigenic stimulus were higher in multiple HPV-infected SCCs. Ion gated channel activity was higher in the other three groups, which was similar as non-HPV16 infected tumors (Fig. 6d). Finally, pathways including the viral latency, fibroblast growth factor binding, TGFb pathway were higher in HPV16 infection as the main type SCCs, where positive regulation of interleukin 1 secretion, positive regulation of morphogenesis of an epithelium, and atrbrca pathway were lower (Fig. 6d).

Proteins in negative regulation of T cell receptor signaling pathway, LGALS3, EZR, PHPT1, and PAWR were elevated in Group1-4, DUSP3 elevated only in Group2-3 (Kruskal–Wallis test, BH-adjusted $p < 0.05$; Fig. 6e). INPP1 was elevated in Group1-4, and NUDT3 was elevated in Group1-3 involving in inositol phosphate catabolic process (Kruskal–Wallis test, BH-adjusted $p < 0.05$; Fig. 6e). The protein abundances of molecules in inositol phosphate catabolic process for the non-anogenital SCCs were compared to anogenital SCCs in Supplementary Fig. 10e, INPP1 and IMPA2 showed a lower expression in non-anogenital SCCs compared to Group 1-3 of anogenital SCCs (Wilcoxon rank-sum test, $p < 0.0001$ for both; Supplementary Fig. 10e). NUDT3 showed an opposite expression pattern as it mediates phosphate degradation[68], expressing higher in non-anogenital SCCs (Wilcoxon rank-sum test, $p < 0.0001$;

Supplementary Fig. 10e). INPP1 showed a high expression level in Group 1 of anal, penis SCCs (Supplementary Fig. 10f). Thus, these analyses indicating that HPV16 infection may lead to active inositol phosphate catabolic process and immune evasion (Fig. 6f), participating in HPV16 + SCC carcinogenesis.

**Performance of tumor type classifier.** Effective management of SCC should include reliable biomarkers for detection and rationally designed drugs for its prevention and treatment. We hypothesized that the information content from deep proteomic profiling would be sufficiently rich to predict the tumor site of origin with high accuracy. We developed a machine learning-based classifier to determine the ability of proteomic expression to inform the diagnosis in patients with SCCs (Fig. 7a). In our training set of 249 patients, the diagnostic SCC type was accurately predicted (both sensitivity and specificity were 100%) in all patients based on 10-fold cross-validation (Supplementary Fig. 11a). When applied to the validation set of 84 samples, the model achieved 100% for both sensitivity and specificity (Fig. 7b).

Given that these proteomic markers might have averaged signals from different cell populations, we examined the spatial expression of 3 markers, PRKCE, SL27A1, and CPXM2, on the tissue level by immunohistochemistry of consecutive slides using patients in the pan-SCC cohort (Fig. 7c, d; same tissue from one patient in a row). In addition, we stained for P63 as a classical pan-SCC marker, P16 as a marker associated with HPV infection, and did EBER in situ hybridization (ISH) to mark the EB virus infected cells. Staining of P63 showed an overall positive in these SCC tissues. EBER was only positive in the nasopharyngeal SCC. P16 expression was strong positive in the cases of cervical and vagina SCCs with positive HPV infection, whereas it was positive in the HPV negative ESCC and thymus SCCs shown in Fig. 7d, limiting its diagnostic value for HPV-positive SCCs.

PRKCE plays essential roles in the regulation multiple cellular processes linked to cytoskeletal proteins, functions in ion channel regulation, and is involved in cancer cell invasion and regulation of apoptosis[69]. Immunostaining of PRKCE was significantly different among 17 SCCs, and showed an overall high expression in cervical and vagina SCCs (Kruskal–Wallis test, $p < 0.0001$, Fig. 7d and Supplementary Fig. 11b). SLC27A1, mediates the ATP-dependent import of long-chain fatty acids into the cell by mediating their translocation at the plasma membrane and serves as an acyl-CoA ligase activity for long-chain and very-long-chain fatty acids[70]. From the proteomic data, SLC27A1 was highly expressed in nasopharyngeal, gallbladder, and pancreatic SCCs. In agreement, we noted a high proportion of tumor-specific positive SLC27A1 staining in gallbladder and pancreatic SCCs (Kruskal–Wallis test, $p < 0.0001$, Fig. 7d and Supplementary Fig. 11c). Immunostaining of SLC27A1 identified that normal epithelial cells of nasopharynx and lung showed positive staining other than SCC cells. CPXM2 has been associated with developmental diseases and are reported as an unfavorable prognostic marker involving in gastric cancer[71], osteosarcoma progression[72], and hepatocellular carcinoma[73]. In this study, we

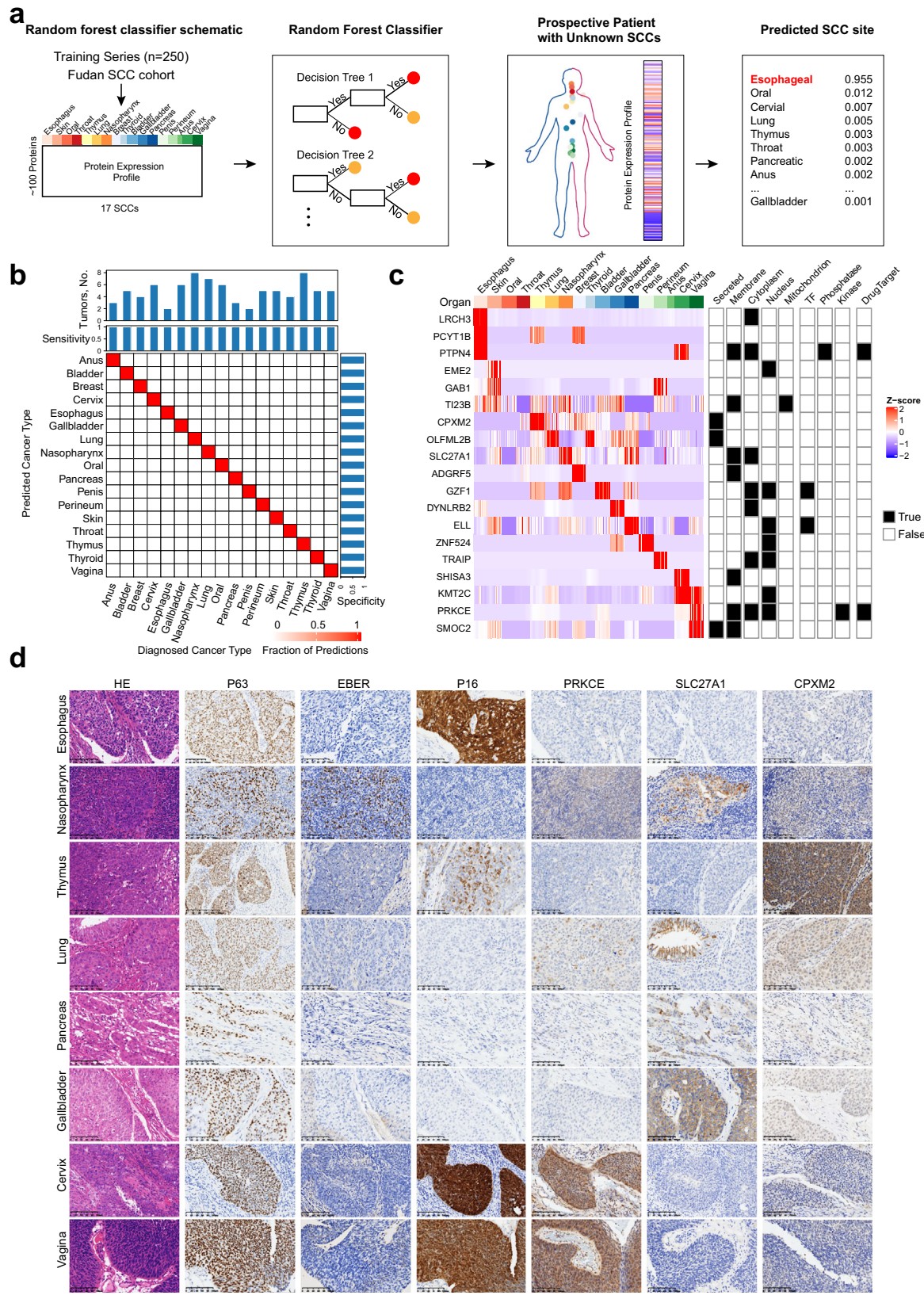

found CPXM2 IHC staining, in agreement with the proteomics data, was highly expressed in thymus SCCs (Spearman correlation, $p = 0.015$; Fig. 7d and Supplementary Fig. 11d, e). These results highlight the power of incorporating proteomics and pathology to explore cancer-type-specific biomarkers.

## Discussion

Comprehensive genomic and transcriptomic analysis of pan-SCCs has broadened our knowledge of the molecular events relevant to this malignancy[35,74]. Herein, we report that the systematic proteome analysis would unravel insights of SCCs shared

**Fig. 7 Performance of tumor type classifier and validation for diagnostic markers. a** Schematic of random-forest classifier. Seventy five percent patients of pan-SCC cohort were used to train the classifier. **b** Performance of the classifier across 17 SCCs in validation set. True (established) cancer types are displayed horizontally and predicted cancer types are displayed vertically. The number of tumors of each cancer type in the cohort is shown at the top, and sensitivity and specificity of the predictions are indicated at the top and right. **c** The heatmap of 19 proteins used in the classifier and annotation of cellular component, TF, kinase, phosphatase, and drug target are provided. **d** Haematoxylin and eosin (H&E) staining, immunohistochemistry (IHC) staining of P63, P16, PRKCE, SLC27A1, and CPXM2, and in situ hybridization (ISH) of EBER in representative samples of partial SCC types (originating organ includes: esophagus, nasopharynx, thymus, lung, pancreas, gallbladder, cervix, and vagina). Scale bar, 100 μm.

and specific features into the clinical, biological, and therapeutic understanding of SCCs. The proteomic landscape of pan-SCC samples and pan-AC samples revealed the differentially activated key signaling pathways, rare SCC-specific metabolic characteristics, clinically and therapeutically relevant subgroups, HPV16-specific features in anogenital SCCs, and differentially diagnostic markers for specific SCC. Primarily, our dataset might serve the scientific community as a resource of clinical proteomic data, which is still sparse in some rare SCCs.

The inclusion of deep-scale proteomic of 17 SCCs and 7 ACs allowed us to compare these two main histological cancer types and identified pivotal distinguishable pathways, including keratinization, glucose metabolism, and ECM. The glandular epithelium is characterized by particular exocrine and endocrine functions that have resulted in AC as characterized by the ribosome and metabolic phenotypes[75]. We revealed that most of DEPs in glucose metabolic alterations and ECM are essential factors that are associated with the poor clinical outcome both in SCCs and ACs using the TCGA database, indicating metabolism reprogramming and TME may participate in tumorigenesis and progression. Keratinization as the major property of squamous epithelium was enriched in SCCs. Keratinization as a good differentiation indicator of SCCs led to good prognosis for SCCs, but poor prognosis for ACs. Epithelial tumors maintain specific keratin expression patterns of the respective originating cell type, and ACs are characterized by the predominance of simple-epithelial keratins, notably KRT7, KRT8, KRT17, KRT18, KRT19, and KRT20[76–78]. Those keratins were also frequently found to be associated with adverse prognosis for adenocarcinomas, such as KRT17 in endometrial cancer[79] and KRT18 in colorectal cancer[80]. Moreover, SCCs were characterized by increased p53 downstream pathway, consistent with the *TP53* mutation was much higher in SCCs compared to ACs, especially in lung cancer[30,43], indicating cancer type-specific effects of DNA repair. While the overall number of the pan-SCC and pan-AC cohort is substantial, the numbers of samples available per anatomic site ranged from 10–22 for SCC and 8–12 for AC. Therefore, the relatively small sample size may limit our findings for specific tumor types. As the survival analysis was mainly explored using RNA-seq datasets, further proteomic studies will be needed to validate the prognostic values of these proteins.

Dysregulation in lipid metabolism is among the most prominent metabolic alterations in cancer[81]. We identified that rare SCCs showed a high level of lipid metabolism, indicating that metabolic reprogramming was associated with rare SCC aggressiveness. Notably, two TFs, RUNX2 and FOXO1, were identified as vital transcriptional regulators involved in rare SCC. It has been reported that RUNX2 is upregulated in various cancer types and may drive breast cancer cell growth and metastasis[82,83]. A previous study also revealed that loss of RUNX2 could sensitize osteosarcoma to chemotherapy[84]. FOXO1 affects adipocyte differentiation by regulating lipogenesis and cell cycle[85]. However, FOXO1 may inhibit RUNX2 transcriptional activity in prostate cancer[49], and we also revealed a negative correlation between these two proteins. Thus, upregulation of RUNX2 or FOXO1 may alternatively favor fatty acid metabolic pathways to promote

cancer proliferation. Of note, PLIN1, as a lipid droplet associated protein, showed a high expression in rare SCCs that may be regulated by FOXO1 and RUNX2. Three anal SCCs showed a PLIN1 amplification, which is needed to be further studied.

Based on pan-SCC proteome, hierarchical clustering stratified patients into four distinct subgroups, revealing anatomical features. Emerging evidence showed that how tumor cells respond to therapy is not solely dependent on the tumor entity but also is regulated by the TME[86]. Furthermore, a consensus clustering based on multiple cell-type signatures was performed to gain an overview of pan-SCC TME. Also, we advocated the importance of characterizing TME subtypes that cross SCC organ boundaries and unite SCCs of disparate organs. Thus, such insights will lead to extensions of treatments shown to be effective in one type of tumor to other tumors sharing the same TME features. Ion channels contribute to most of the fundamental biological processes including cell proliferation, secretion of hormones, as well as immune response[87,88]. We found distinguishing ion channel drug targets in 5 TME subtypes, including sodium channel (SCN4B in ClSq and SCN5A in IhSq), potassium voltage-gated channel (KCNQ4 in BaSq, KCNA5 in NeSq, and KCND2 in IhSq), and voltage-dependent calcium channel (CACNA2D2 in FaSq and CAC1E in IhSq). Solute carrier (SLC) transporters facilitate the transport of a wide array of substrates across biological membranes[89], human proteogenomic studies have provided powerful insight into the therapeutic roles of SLC transporters in variety cancer types[90–93]. Moreover, we found SLC16A7 (BaSq), SLC6A9 (NeSq), SLC7A2 (EoSq), and SLC12A1 (IhSq) were specifically expressed in four subtypes and could serve as treatment targets for certain subtypes. Although targeting TME compartments proves to be successful in the preclinical phase, an enormous challenge still lies ahead for translating these strategies to clinical practice.

HPV16 infection as the main type for Chinese anogenital SCCs, over-representing with the high metabolic characteristics especially in inositol phosphate catabolic process. Interestingly, Inositol phosphates were reported promoting HIV-1 assembly and maturation to facilitate viral spread in human CD4 + T cells[94]. Multiple isomers of inositol phosphate were found in Epstein-Barr-virus- transformed (T5-1) B-lymphocytes and may be related to cell transformation or proliferation[95]. So, we think the Inositol phosphate catabolic process participates in HPV-related tumorigenesis. However, due to the small sample size, a large-scale study will be needed to explore this further.

Tissue enriched proteins provided insights into differentially diagnostic values. We observed the diagnostic values that PRKCE for vaginal and cervical SCCs, CPXM2 for thymus SCCs, and SLC27A1 for pancreatic and gallbladder SCCs. Our results highlighted the power of incorporating proteomics and pathology to explore cancer-type-specific biomarkers. There are limitations to the validation study. Firstly, only three markers were validated, as no specific antibodies were available for the other 16 markers. We will validate the other markers in future research when antibodies are available. Secondly, the number of samples for validation is relatively small. Most importantly, how these markers perform in telling the origin of metastatic SCCs remain

unknown. Large-scale validation analyses, including metastatic SCCs, are urgently needed.

Taken together, our current work provides a systematic and comprehensive analysis of pan-SCCs using proteomic platform. These results add an additional layer to the complex, which might have important implication for better understanding SCCs.

## Methods

### Patient samples of SCCs

*Clinical sample acquisition.* Archival formalin-fixed paraffin-embedded (FFPE) tissues of 17 different organs from 333 SCC participants were randomly selected from May 2001 to December 2019 with SCC who were undergoing surgical resection. Most of the cases had squamous cell carcinoma histology, and a small part of cases had adenosquamous carcinoma or urothelial carcinoma with squamous differentiation belonging to pancreas, gallbladder, lung, or bladder (Table S1A and S1B), but collected regardless of histologic grade or surgical stage. Patients were categorized into common SCCs (SCCs originated from nasopharynx[96], oral cavity[96], throat[96], skin[97], esophagus[98], lung[99], cervix[100], penis[101], perineum[102], and vagina[103]) and rare SCCs (SCCs originated from thyroid[104], thymus[105], breast[106], pancreas[107], gallbladder[108], bladder[109], and anus[110], as evaluated by 4th WHO Classification of Tumors[111–118]. Patients were excluded if they had other advanced disease, active second malignancy, or any condition that may influence the outcome evaluation (68 rare SCC patients with no outcome information), such as neoadjuvant treatment with chemotherapy, radiotherapy, or targeted therapy. Clinical information of 333 patients including gender, age, TNM staging (AJCC cancer staging system 8th edition), year of surgery, status of cancer recurrence or relapse, and status of survival is listed in Supplementary Data 1.

As comparison group of pan-SCC cohort, we randomly collected 69 adenocarcinomas, including pancreatic (8), colorectal (8), gallbladder (12), thyroid (10), breast (8), gastric (12), and lung (11) adenocarcinomas, who underwent surgery were collected at Zhongshan Hospital, Fudan University between July 2017 and August 2019. Patients were excluded if they have been treated with preoperative radiotherapy, chemotherapy, targeted therapy, or suffering from other cancers.

The samples used in this study were obtained from the Zhongshan Hospital, Fudan University. The Research Ethics Committees of Zhongshan Hospital approved the study and all patients provided written informed consent for sample collection, analysis, and publishing basic and clinicopathological information.

*Histological evaluation and immune cell infiltration.* Haematoxylin and eosin (H&E) stained slides were examined and evaluated independently by two expert pathologists and information regarding tumor histological subtype, degree of differentiation, degree of keratinization, size of cell nest, cell size, cancer inflammation, stromal ratio, stromal inflammation, tumor necrosis, pathological mitotic figures, and tumor purity were provided. In addition, the non-tumor cell populations, including stroma ratio and immune cell infiltration were evaluated. Acceptable SCC and AC tissue segments were determined by pathologists based on the percent viable tumor nuclei (>80%), total cellularity (>50%), and necrosis (<20%).

*Sample preparation.* FFPE specimens were prepared and provided by Zhongshan hospital. One 4 µm thick slide sliced from FFPE blocks were used for H&E staining. For proteomics sample preparation, 10 slides (10 µm thick) were sectioned, deparaffinized with xylene and washed with gradient ethanol and water. Selected specimens according to H&E staining were scraped, and materials were aliquoted and placed in storage at −80°C until further processing. Each sample was assigned a new research ID and the patient's name or medical record number used during hospitalization was de-identified.

*DNA extraction and HPV infection status detection.* DNA extraction was performed using GenElute™ FFPE DNA Purification Kit (Sigma-Aldrich) according to the manufacturer's protocol. DNA was quantified by the NanoDrop 2000 (Thermo Scientific). HPV infection status detection was performed using High-risk Human Papilloma virus (HPV) Genotyping Real Time PCR Kit (Life River) according to the manufacturer's protocol. A total of 15 high risk HPV types, including 16, 18, 31, 33, 35, 39, 45, 51, 52, 56, 58, 59, 66, 68, and 82 for all cases were tested.

### Proteomic workflow

*Protein extraction and tryptic digestion.* For SCC sample preparation, slides (10 µm thick) from FFPE blocks were dissected according to HE staining (adjacent non-tissue was discarded), deparaffinized with xylene and washed with gradient ethanol. Approximately one milligram of human SCC samples was homogenized in lysis buffer at a ratio of about 200 µl lysis buffer for every 1 mg wet weight tissue. The lysis buffer consisted of 0.1 M Tris-HCL (pH 8.0), 0.1 M DTT, and 1 mM PMSF. The samples were grinded by grinding rods for 3 min and 50 µl of 20% SDS was added to reach a maximum SDS concentration of 4%. Lysates were boiled at 99 °C, 1,500 g for 0.5, 1, and 1.5 h and then centrifuged at 12,000 g for 5 min at room

temperature. Lysate supernatant was transferred into acetone at a maximum ratio of 1:3 and kept in −20 °C for at least 4 h. Precipitated proteins were washed by cooled acetone for three times, redissolved by 8 M Urea. Prior to digestion, samples were loaded into an ultrafiltration filter column (10 kD, 500 µl) and centrifuged at 12000 g for 15 min. The reduction and alkylation progresses were then carried out, 100 µl of reduction buffer (10 mM DTT, 25 mM NH$_4$HCO$_3$) were loaded on ultrafiltration filter column, incubated for 1 hour at 56°C, and centrifuged at 12000 g for 15 min. Then 100 µl of alkylation buffer (55 mM IAA, 25 mM NH$_4$HCO$_3$) were loaded on ultrafiltration filter column, incubated for 45 min in dark at room temperature. The filter was then washed three times by adding 100 µl NH$_4$HCO$_3$ (50 mM) to the column, followed by centrifugation. The protein concentration of the solution was measured using spectrophotometer (NanoDrop, Thermo, USA). The amounts of protein were adjusted to 400 µg. Digestion was performed with ultrafiltration filter column with Sequencing Grade Modified Trypsin (Promega V5111) in 50 µl NH$_4$HCO$_3$ (50 mM) for 16 h at a 1:50 enzyme-to-protein ratio at 37 °C. Digested samples were acidified in 0.1% formic acid (FA) solution buffer to stop digestion and were collected by centrifugation at 12,000 g for 15 min and followed by twice washing with 200 µl of LC-MS grade water. Samples were desalted on C18 columns and dried down using centrifugation.

*Nano-LC-MS/MS.* Dried peptides were reconstituted in 0.1% FA and injected onto a reverse phase C18 homemade 150 µm × 30 cm silica microcolumn (particle size, 1.9 µm; pore size, 120 Å; SunChrom, USA) using a nanoElute (Bruker Daltonics). Target on-column load was 200 ng (measured by NanoDrop before loading) total peptide per injection with a pressure of 280 bar. The flow rate was 600 µl/min. Mobile phase A was 0.1% FA, 99.9% water; mobile phase B was 0.1% FA and 99.9% acetonitrile. The gradient was linear from 2% B to 35% B over 110 min. A blank wash run followed each sample run to ensure no cross contamination. The mass spectrometer was a timsTOF Pro (Bruker Daltonics) set to acquire data in Parallel Accumulation Serial Fragmentation (PASEF) mode[119]. The TIMS accumulation time was set to 100 ms and precursor masses for 0.4 min where charge states of 2-4 were allowed. The resolution parameter was set to 50,000 for MS1 and MS2. Mass spectra for MS1 and MS2 scans were recorded between 100 and 1700 m/z. Ion mobility resolution was set to 0.60–1.60 V·s/cm over a ramp time of 100 ms. Data-dependent acquisition (DDA) was performed using 10 PASEF MS/MS scans per cycle with a near 100% duty cycle. An active exclusion time of 0.4 min was applied to precursors that reached 20,000 intensity units.

*Database searching of MS data.* Proteins were identified by PEAKS Online (version 8.5; Bioinformatics Solution Inc., Waterloo, Canada) with searching the library of Uniprot homo sapiens (version 2019_07, SwissProt, 20,431 sequences), from the untargeted proteome data. ID transfer was applied. The enzyme was trypsin and FDR cutoff on both the peptide and protein level was 1%. Proteins were identified based on at least one unique peptide. Carbamidomethylation was set as the fixed modification. Oxidation (M) and N-acetylation were set as the variable modifications. The precursor mass tolerance was set to 15 ppm and the fragment ion tolerance at 0.05 Da. The retention time shift tolerance was 4.0 min. The retention time range is 0.0000 ≤ Retention Time ≤ 10000.0000. Contaminants were regularly searched, which is mainly containing keratin (Human, Mouse, Bovin), and lab-derived contaminants (BSA, trypsin, etc.), as the mass spectrometry maintenance routine.

### Proteome data preprocess

*Mass spectrometry platform QC and SCC proteome QA.* The 293T cell (National Infrastructure Cell Line Resource) lysate was measured every three days as the quality-control standard for the quality control of the performance of mass spectrometry. The quality-control standard was digested and analyzed using the same method and conditions as the SCC samples. A pairwise Spearman's correlation coefficient was calculated for all quality-control runs in the statistical analysis environment R v3.6.3, the results are shown in Supplementary Fig. 2a. The average correlation coefficient among the standards was 0.90, and the minimum and maximum correlation coefficient were 0.85 and 0.96, respectively. In addition to the 293T controls, seven pairs of SCC samples were performed in the middle and at the end of the project. The average, minimum, and maximum correlation coefficient of replicate samples were 0.92, 0.86, and 0.97, respectively. These quality-control samples demonstrating the consistent stability of the mass spectrometry platform. The log2-transformed protein abundances for each SCC sample (Supplementary Fig. 2b) were plotted to show good consistency of proteome quantification. Spearman correlation coefficients for all 333 MS runs were presented in Supplementary Fig. 2c. The median correlation coefficient among these samples was 0.74, and the maximum and minimum values were 0.99 and 0.56, respectively.

*Data normalization and missing value imputation.* Protein quantification used precursor ions MS (MS1) signal intensities with total ion current (TIC) normalization. For a peptide "i" from a sample "n," its quantitative information was its peak area (Pi) calibrated by the relative TIC of the sample: Pi * (TICa/TICn), where "a" represents the sample that was randomly chosen as the benchmark. The quantitative information of the top three peptides of a protein was averaged to get the protein-level relative quantitative information. The normalized TIC intensities of 333 samples were extracted from the PEAKS Online result files to represent the

final expression of a particular protein across samples, resulting in a 14,840 × 333 protein-expression matrix. The expression matrix was then log2-transformed and used in all quantitative analysis. The pan-SCC data was not a normal distribution (Kolmogorov-Smirnov test, p < 0.05 for all 333 samples distribution). Missing values were imputed with the one-tenth of the minimum across our proteome data.

*Batch effect analysis.* We used the hierarchical clustering function hclust and principal-component analyses function prcomp in the R language to assess the batch effects between SCC and AC samples. There was a clear separation between the SCCs and ACs samples (Supplementary Fig. 4a) before correcting batch effects, mostly due to their different batches. Then, we corrected their batch effects on a list of proteins expressed in SCC and AC samples using ComBat[120]. The protein list was identified as follows: (1) The commonly expressed proteins of SCC were required to be identified in at least 50% of the samples of each SCC organ, which contains 5130 proteins; (2) The commonly expressed proteins of ACs were required to be identified in at least 50% the samples of each AC organ, 4845 proteins were identified; (3) The union set of commonly expressed proteins in SCC (5130 proteins) and AC (4845 proteins) contained 5914 proteins, and the intersection contained 4061 proteins. A total of 1069 proteins were only commonly expressed in SCCs, and 46 proteins were only detected in SCCs (not detected in ACs). A total of 784 proteins were only commonly expressed in ACs, and 30 proteins were only detected in ACs (not detected in SCCs); (4) We removed a total of 76 proteins only expressed in SCCs or ACs, and a total of 5838 proteins were obtained. Hierarchical clustering of Combat corrected proteomic profiles showed no significant batch effects, PCA analysis showed that SCC and AC has a clear proteomic difference (Supplementary Fig. 4).

*Dataset filtering.* Pan-SCC protein-expression matrix from the raw PEAKS Online result file (14,840 × 333) was filtered to 14,598 × 333, which is the union set of proteins identified in at least 1/3 samples of each organ. Similarly, Pan-AC protein-expression matrix from the raw PEAKS Online result file (10,501 × 69) was filtered to 10,414 × 69. Some of the filtering steps were specified for different analyses in the study. In the comparison between SCC and AC, at least half of samples of each organ were required to have non-missing values, resulting 5130 common proteins of SCC, 4845 common proteins of AC. These two datasets were combined and unique proteins identified only in SCC or AC were removed, resulting in a 5838 × 402 protein-expression matrix. For the comparison between Common SCCs and Rare SCCs, 6,123 proteins were selected by combining the common proteins of Common SCCs and Rare SCCs, the common proteins of Common SCCs and Rare SCCs were defined as the intersection of proteins expressed in at least half samples of each organ within Common or Rare SCCs, respectively. Alternate filtering has been noted in descriptions of the relevant methods.

## Tissue microarray (TMA) experiment

*TMA construction.* TMAs were constructed using 333 tumor tissues (the same cohort as proteomics) and paired normal epithelial/non-tumor tissues (partially) from the Fudan Pan-SCC cohort using the method as we previously described[121]. In brief, all cases were histologically inspected by H&E staining and representative areas were pre-marked on the paraffin blocks, away from necrotic and hemorrhagic regions. Duplicates of 2 mm side length square from two different areas, tumor center and normal epithelium/non-tumor tissue, were included, along with a series of different controls, to ensure reproducibility and homogeneous staining of the slides.

*Immunohistochemistry staining.* FFPE tumor tissues were sliced into 4 μm slides for immunohistochemistry (IHC) staining. Slides were stained with TP63 antibody (1:100, Thermo, catalog No. 703809), P16 antibody (1:100, Sigma-Aldrich, catalogue No. MAB4133), PLIN1 antibody (1:100, Cell signaling technology, catalogue No. 9349), RUNX2 antibody (1:100, R&D Systems, catalogue No. MAB2006), FOXO1 antibody (1:100, Cell signaling technology, catalogue No. 2880), PRKCE antibody (1:50, Absin, catalogue No. abs136354), SLC27A1 antibody (1:50, Thermo, catalogue No. PA5-50574), and CPXM2 antibody (1:100, Absin, catalogue No. abs138862) was performed on the Leica Automated Quantitative Pathology System. TMA slides were first deparaffinized and rehydrated followed by microwave antigen retrieval. After blocking endogenous peroxidase and nonspecific binding sites, primary antibodies and second HRP-conjugated antibody (Bond Polymer Refine Detection, Leica Biosystems, Catalogue No. DS9800) were applied. Immunoreactivity was evaluated independently by two investigators who were blinded to clinical data according to the intensity and extent of staining. Staining intensity was scored as: 0 (negative), 1 (weak), 2 (moderate), and 3 (strong). The staining extent was scored as 0 (0%), 1 (<10%), 2 (10-50%), and 3 (>50%), on the basis of the percentage of positively stained cells. The product of the intensity and extent scores was used as the final staining score.

*PLIN1 fluorescence in situ hybridization.* Fluorescence in situ hybridization (FISH) assay was performed on the tissue microarrays using PLIN1 probe that hybridizes to the band 15q26 with Spectrum Orange and CEP 15 with Spectrum Green (Empire Genomics Corp, PLIN1-20-OR) following routine methods. Two experienced evaluators blinded to the clinical data interpreted FISH analyses. At least 50

nuclei per patient were evaluated. The threshold for assigning a sample to amplification was an average number of PLIN1 signals/tumor cell nucleus ≥ 6.0, and copy-number gain was an average number of PLIN1 signals/tumor cell nucleus < 6 and ≥ 4.0. PLIN1 signals/ tumor cell nucleus < 4 was scored negative for PLIN1 amplification.

**Differential expression analysis.** The Wilcoxon rank-sum test (as implemented in R software) was used to examine proteins that were differentially expressed between SCCs and ACs, common SCC samples and rare SCC, followed by multiple testing correction using the Benjamini–Hochberg (BH) Algorithm. Upregulated or downregulated proteins in SCCs were defined as proteins differentially expressed in SCCs compared with ACs (fold change (expressed as log2(ratio of average protein abundance between SCCs and ACs)) ≥ 1 or ≤ −1, Wilcoxon rank-sum test, BH-adjusted p < 0.05). Upregulated or downregulated proteins in common SCCs were defined as proteins differentially expressed in common SCCs compared with rare SCCs (fold change (expressed as log2(ratio of average protein abundance between common SCCs and rare SCCs)) ≥ 1 or ≤ −1, Wilcoxon rank-sum test, BH-adjusted p < 0.05). Kruskal–Wallis test (BH-adjusted p < 0.05) was used to identify proteins that were differentially expressed between the four clusters. A simple linear model and moderated t-statistics, implemented with the R/Bioconductor package limma v.3.40.6, were used to identify differentially expressed proteins between the 6 immune subtypes, and the following cutoff criteria were used: (1) all BH-adjusted P values should be less than 0.05 compared to the other subtypes; (2) fold change (expressed as log2(ratio of average protein abundance between immune subtypes) ≥ 1.5 or ≤ −1.5); and (3) at least 50% expression in one subtype. Because the differentially proteins of subtype 6 are overly abundant, we adjusted the filter criteria (fold change (expressed as log2(ratio of average protein abundance between immune subtypes) ≥ 3 or ≤ −3)). Proteins significantly regulated were visualized by ggplot2 (version 3.2.1) and ComplexHeatmap (version 2.5.3).

**Pathway enrichment analysis.** Gene set enrichment analysis (GSEA) performed by clusterProfiler (version 3.12.0) was used for pathway enrichment analysis of the comparison between SCCs and ACs. An FDR value of 0.05 was used as a cutoff. Pathway enrichment analysis by Reactome (https://reactome.org/) was used for the comparison between common SCCs and rare SCCs. For further insight into biological implications according to the four clusters identified based on the proteomic profiles of SCCs, single-sample gene set enrichment analysis (ssGSEA) was performed using the GSVA algorithm (R/Bioconductor package GSVA, version 1.32.0) to calculate the ssGSEA scores for each gene set with at least ten overlapping genes. The gene sets used both in the GSEA and the ssGSEA analysis were downloaded from the MsigDB database (https://www.gsea-msigdb.org/gsea/msigdb/index.jsp, version 7.2), which contains 1986 concepts (including 186 Kyoto Encyclopaedia of Genes and Genomes gene sets, 196 PID gene sets, 1554 Reactome gene sets and 50 hallmark gene sets).

**Correlation between SCC types and clinical features.** In the measurements of correlations between SCC types and clinical features, Fisher's exact test was used on categorical variables, including gender, differentiation, keratinization, tumor nest size, and stromal inflammation. Wilcoxon rank-sum test was used on continuous variables including pathological mitotic figures, cancer inflammation, immune score and ESTIMATE score.

**Hierarchical clustering.** Hclust function implemented in the R language was used to perform unsupervised clustering of Pan-SCC samples to identify proteomic features of each cluster. Before clustering, we selected proteins as follows: (1) Proteins were required to express in at least 1/3 samples in certain SCC type. (2) The filtered proteins were sorted in descending order by mad (Median Absolute Deviation). The 20% bottom mad proteins of each certain organ were selected as the proteins with consistent and ubiquitous expression in each SCC. (3) A total of 17 protein sets were combined and duplicates were removed, resulting a protein set of 10,259 proteins. Those 10,259 proteins were averaged in each organ, resulting in a 10,259 × 17 protein-expression matrix, and sorted in descending order by mad. Four clear clusters were found when using the Pearson algorithm based on the 1500 top MAD proteins.

**Immune-based consensus clustering.** The abundance of 64 different cell types of Pan-SCC samples were computed via xCell[51] using log2 (TIC normalized) protein expression values. Table S5A contains the final score computed by xCell for different cell types for the 333 samples. Consensus clustering was performed using the R packages ConsensusClusterPlus (version 1.48.0)[122]. Only cells that were detected in at least 20 patients were utilized. Specifically, 80% of the original samples were randomly subsampled without replacement and partitioned into 6 major clusters using the Partitioning Around Medoids (PAM) algorithm, which was repeated 1000 times. In addition to xCell, ESTIMATE[123] was used to estimate Tumor Purity, Stromal and Immune Scores. ssGSEA[124] was performed to obtain pathway scores and identify the pathway alterations that underlie our Pan-SCC clusters, using the R package GSVA[125]. The protein expression matrix was subjected to the GSVA algorithm to calculate the ssGSEA scores for each gene set with at least 10 overlapping genes. Wilcoxon test was performed subsequently to find pathways

differentially expressed between every cluster and all other clusters. The resulting *p* values were adjusted via Benjamini–Hochberg procedure. Table S5B and S5C show pathways and proteins altered of each cluster.

**Potential diagnostic biomarkers**. A random-forest classifier was developed to predict tumor type based on proteomic profiles of our Pan-SCC cohort, using the R package randomForest (version 4.6.14)[126]. Before analysis, we selected proteins as follows:1) the most consistently expressed proteins were selected within each organ as 1% bottom mad proteins identified in at least 75% cases in each organ. A total of 17 protein lists were then combined, and one protein list containing 1220 proteins after removing repetitive proteins was obtained (a total of 1220 proteins were obtained from 333 cases). 2) all 333 SCC cases were randomly divided into a training set and a validation set, containing 75% and 25% cases, respectively. 3) the RandomForest function was used on the training set to calculate the importance (Indicated by "Mean Decrease Accuracy") of each protein (top importance was obtained only from training set). 10-fold cross-validation was used to select the suitable number of top important proteins.

**Kinase-TF network**. We extracted pathway-specific TFs by filtering their GO terms. The hierarchical network of kinases, TFs, and target genes were constructed based on multiple layers. The protein interaction among kinases, TFs, and target genes was annotated using information accessed from GeneMANIA (http://genemania.org/). The network was visualized using Cytoscape v 3.8.1.

**HPV-related SCCs analysis**. In the set of anogenital SCCs (94 cases in total), including anus (10/10 infected, HPV infection rate 100%), penis (11/22, 50%), perineum (14/20, 70%), cervix (20/21, 95%), and vagina (20/21, 95%) (Fig. 6a, Supplementary Data 6).

We used gene sets of molecular pathways from the MsigDB database version 7.2 (https://www.gsea-msigdb.org/gsea/msigdb/index.jsp), which contains 1,986 concepts (including 186 Kyoto Encyclopaedia of Genes and Genomes gene sets, 196 PID gene sets, 1,554 Reactome gene sets and 50 hallmark gene sets), to compute single-sample gene set enrichment scores for each sample.

To elucidate the molecular pathways differentially activated or inactivated in HPV16 infected SCCs compared with other types or negative SCCs, we grouped these anogenital SCCs according to the HPV infection patterns (Fig. 6c). Group 1, HPV16 infection only; Group 2, multiple infection and HPV16 is the main type (minimum cycling threshold was used in PCR process); Group 3, multiple infection and HPV16 is not the main type; Group 4, HPV infection and not HPV16; Group 5, negative (HPV_neg). A simple linear model and moderated t-statistics, implemented with the R/Bioconductor package limma (version 3.40.6), were used to identify differentially pathways between the 4 Groups with HPV infection and Group 5 (tissue of origin was adjusted).

To select the eight groups of pathways with different characteristics of HPV infection status, we used BH-adjusted *p* value < 0.05 (between 4 Groups with HPV infection and Group 5) for differential behavior, and fold change (expressed as log2(ratio of enrichment score between Group1-4 and Group 5) > 0) for over-represented pathways in Group 1-4 and fold change (expressed as log2(ratio of enrichment score between Group1-4 and Group 5) < 0) for over-represented pathways in Group 5. For specific pathway groups, this amounted to the following conditions: pathway group 1: BH *p* < 0.05, log2 (FC) > 0 of Group 1-4; pathway group 2: BH *p* < 0.05, log2 (FC) < 0 of Group 1-4; pathway group 3: BH *p* < 0.05, log2 (FC) > 0 of Group 1-3; pathway group 4: BH *p* < 0.05, log2 (FC) < 0 of Group 1-3; pathway group 5: BH *p* < 0.05, log2 (FC) > 0 of Group 2-3; pathway group 6: BH *p* < 0.05, log2 (FC) < 0 of Group 2-3; pathway group 7: BH *p* < 0.05, log2 (FC) > 0 of Group 1-2; pathway group 8: BH *p* < 0.05, log2 (FC) < 0 of Group 1-2. Pathways were visualized by ggplot2 (version 3.2.1) and ComplexHeatmap (version 2.5.3).

**Survival analysis**. We constructed multivariate Cox proportional hazard model using 'age', 'gender', 'stage', 'histology', 'organ', 'protein expression' as the fitting variables, followed by multiple testing correction using the BH Algorithm for pan-SCC cohort. The following 3 criteria were used to identify potential prognostic biomarkers: (1) log-rank test *p* < 0.05; (2) the multivariate Cox proportional hazard model *p* < 0.05; (3) all survival analysis has the consistent prognostic value. For the TCGA datasets survival analysis, prior to the log-rank test of a given protein, survminer (version 0.2.4) R package with maxstat[127,128] (maximally selected rank statistics) was used to determine the optimal cutoff value for the samples. OS curves were then calculated (Kaplan–Meier analysis, log-rank test) based on the optimal cutoff value, followed by multiple testing correction using BH Algorithm.

Kaplan–Meier survival curves (log-rank test) were used for overall survival (OS) or disease-free survival (DFS) of the pan-SCC cohort, 4 hierarchical clusterings, 6 proteomic subtypes. Multivariate Cox proportional hazard model were also conducted using 'age', 'gender', 'stage', 'histology', 'organ' (only for 17 SCCs, Supplementary Data 1), 'proteomic clusters' (only for Supplementary Fig. 8), 'microenvironment subgroups' (only for Supplementary Fig. 9) as the fitting variables.

**Screening potential drug targets of SCCs**. Drug targets either approved by FDA or under clinical trials were retrieved from Drugbank database (version 5.1.5)

(http://www.drugbank.ca/). Target proteins that were unregulated in SCCs with potential curative drugs were chosen.

**Statistics and reproducibility**. The statistical significance of differences between two groups was calculated with the Wilcoxon rank-sum test; for three or more groups comparison, Kruskal–Wallis test was used. Fisher's exact test was used for categorical variables and two-sided Wilcoxon rank-sum test was used for continuous variables, when testing association of different groups with clinical variables. For the correlation analysis between two variables, Spearman correlation was used. Kaplan–Meier plots with log-rank test and Cox proportional hazard model were used to describe OS and DFS. All statistical tests were two-sided except special explanation, p-values were adjusted using the BH procedure. For validation experiments, each was repeated at least three times independently, representative photos were shown. All the analyses were performed in R (version 3.6.3) and GraphPad Prism (Version 8).

**Reporting summary**. Further information on research design is available in the Nature Research Reporting Summary linked to this article.

## Data availability

The mass spectrometry proteomics data have been deposited to the ProteomeXchange Consortium (http://proteomecentral.proteomexchange.org)via the iProX partner repository (https://www.iprox.cn/, Project ID: IPX0002831000)[129] with the dataset identifier PXD033794. TCGA data used in this work were downloaded from (https://portal.gdc.cancer.gov/). All relevant data are included in the manuscript and the Supplementary Information. Source data are provided with this paper.

## Code availability

No special code was used in this study. Code for specific figures (Figs. 4b, 7b) during the study are available for research purposes from the corresponding authors on request.

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

## Acknowledgements

This work was supported by Quality control and management system for whole procedure of precision medicine (No. 2017YFC0910003), National Key R&D Program of China (Nos. 2017YFA0505102, 2016YFA0502500, 2018YFA0507501, 2017YFC0908404), National Natural Science Foundation of China (No. 81702372, 31770886, 31972933, 81200355, 31700682), Shanghai Natural Science Foundation (Nos. 18ZR1406800, 18ZR1446300), Xiamen Science and Technology Project of Fujian Province, China (No. 3502Z20184003), Shanghai Municipal Commission of Science and Technology (No. 19441904000), Shanghai Municipal Key Clinical Specialty (No. shslczdzk01302), Shanghai Science and Technology Development Fund (No. 19MC1911000), Science and Technology Commission of Shanghai Municipality (No. 2017SHZDZX01), and Major Project of Special Development Funds of Zhangjiang National Independent Innovation Demonstration Zone (No. ZJ2019-ZD-004).

## Author contributions

Conception and Design, Y.Y.H., C.D., F.C.H., Q.S., and Y.Y.; Methodology, Q.S., Y.Y., D.X.J., and Z.Y.Q.; Performed Experiment and Data Collection, Q.S., Y.Y., D.X.J., Z.Y.Q., C.X., H.X.W., S.B.T., X.L.Z., J.H., M.Y.D., Z.X.Y., X.G.S., R.Q.X., K.L., Y.N.Y. X.T.Y.; Data Analysis, Y.Y., Q.S., Z.Y.Q., D.X.J., Y.F.H., J.B.Q., and L.X.; Writing, Q.S., Y.Y., Z.Y.Q., D.X.J., and J.B.Q.; Patient Sample Management and QC, Q.S., Y.Y., Z.Y.Q., D.X.J.,

X.L.Z., L.L.C., R.K.L., and J.S.; Supervision, Y.Y.H., C.D., F.C.H., and J.S. All authors contributed to data interpretation, manuscript editing, and revision.

## Competing interests

The authors declare no completing interests.
