## [Peer review file · Nature Communications]

Editorial Note: The following figures within the Peer Review File have been redacted as indicated to remove potential third-party material:

Response figure RL15 contains a figure from Gao, Q. et al. Integrated Proteogenomic Characterization of HBV-Related Hepatocellular Carcinoma. *Cell*, Volume 179, Issue 5, Pages 1240 (2019).

Response figure RL19 contains a figure from Kurman, R. J., Carcangiu, M. L., Herrington, C. S., Young, R. H. WHO Classification of Tumours of Female Reproductive Organs. WHO Classification of Tumours, 4th Edition, Volume 6, International Agency for Research on Cancer (2014).

Response figure RL21 contains a screenshot from the MSFragger tool. References:

Kong, A. T., Leprevost, F. V., Avtonomov, D. M., Mellacheruvu, D., & Nesvizhskii, A. I. MSFragger: ultrafast and comprehensive peptide identification in mass spectrometry-based proteomics. *Nature Methods*, 14(5), 513-520. (2017)

Yu, F., Teo, G. C., Kong, A. T., Haynes, S. E., Avtonomov, D. M., Geiszler, D. J., & Nesvizhskii, A. I. Identification of modified peptides using localization-aware open search. *Nature Communications*, 11(1), 1-9. (2020)

Polasky, D. A., Yu, F., Teo, G. C., & Nesvizhskii, A. I. Fast and Comprehensive N-and O-glycoproteomics analysis with MSFragger-Glyco. *Nature Methods*, 17, 1125-1132. (2020)

REVIEWER COMMENTS

Reviewer #1 (Remarks to the Author): Expert in bioinformatics

This paper is an important work that highlights the generation of a very large dataset consisting of 333 patients with squamous cell tumors from 17 different organs. The authors use label-free, mass spectrometry-based proteomics to characterize this cohort and compare with a similarly generated cohort of 69 patients with adenocarcinomas from 7 different organs. The generation of this dataset is a noteworthy and significant accomplishment as a mechanism for comparing the proteomics of squamous tumors from multiple organs consistently (same methodology, same instrument). In addition, many of the tissue types included are rare diseases. While many diseases have been profiled in one form or another, this is a comprehensive look across squamous cell carcinomas using proteomics that is similar to the PanCancer analysis of squamous cell from DNA/Methylation/RNA.

Overall this work reports many interesting findings related to the similarities among squamous cell cancers, differences between squamous cell and adenocarcinoma, and differences between common and "rare" squamous cell carcinomas in the context of proteomics. Since this work spans many different squamous tumor types, it is difficult to compare with the many existing findings from the literature. However, Figure 1d provides a nice summary of known targets within the proteomics data.

The paper is primarily a discovery and descriptive paper of many characteristics of squamous cell carcinomas. As such, the work provides both a survey of the proteomics landscape and a resource for other researchers to utilize. The paper provides many complex analyses of the data, providing both the detailed results as supplemental data and figures summarizing the results. Overall the methods are appropriate although there are a few areas in which there is not enough detail to fully understand the approach taken (see below for specific comments that are likely to be resolved through additional details).

Suggestions for improvement:

It might be helpful to include p values when describing differences that are qualitatively described within the manuscript. This would allow readers to assess the significance of the observation. For instance (line 100: significant differences across organs) could include a statistical test and associated p value to strengthen the conclusion there are statistically significant differences. Another example: Stromal vs ESTIMATE "showed a consistent trend" - can this be quantified?

In the case of the overall survival/disease free survival there are cohorts collected at different times having different time to followup. Therefore, comparing outcomes may be difficult, as censoring may be primarily a late or early event, and may be influenced by date of collection (due to therapy). There does not appear to be multiple testing correction methods applied to survival p-values. Given the large number of exploratory tests for outcomes, multiple testing correction may be appropriate to consider.

Specific Items:

Lines 125-126: It would be helpful in showing KRT5 and TP63 to again find a quantification of the assertion that they are "high and ubiquitous". There is some variation in KRT5, at least. It appears that many thyroid cancers have lower KRT5, so it might be helpful to summarize observations with quantification (perhaps by tissue of origin). For instance, how does the variability in AKT1 or KRT5 compare (for instance, what quantile of variability using MAD as done elsewhere)?

On the basis of the methods and the color gradient in the figure 1d, blue is "missing" data. If this is the case, it may be worth pointing this fact out. FGFR3, PTEN and TP53 may be mutated and thus either not sufficiently translated or translated with alternate forms. Given the (relatively) low expression of those proteins (FGFR3/PTEN) in samples, it may be worth exploring this further.

Identification of proteins may relate back to tissue specificity so it would be helpful to know if the total numbers were significantly different across tissues. Specifically on line 115 "no major differences in the coverage between the 17 SCCs". I would encourage a test of this assertion. Figure 1c suggests there are significant differences. Were samples run by tissue of origin? If they were randomized, then the differences in total protein could be biologically meaningful and worth further investigation. This is particularly true with copy number changes that have been seen in squamous cell cancers.

Batch correction:

When batch correction was done, there was missing data present. Was the missing data imputed in the batch correction or left missing? That is, the missing data is imputed as a low expression however it is possible that batch correction could increase the level of expression of these missing values? This would presumably introduce an unintended batch effect in that case?

Commonly expressed proteins in SCC or AC: It is not clear to me, when 5130 commonly expressed SCC proteins and 4845 commonly expressed AC proteins yield 5838 common proteins of SCC or AC. Is this perhaps the union of the two sets? It would seem more common to assume that the intersection of the two is used as the basis for common proteins. Overall, there are two potential effects: 1) presence of

proteins in one of AC/SCC and 2) expression differences among proteins in both AC/SCC. It is not clear which (or both) of these is being addressed. It is expected that AC and SCC have unique proteins (of interest in and of itself) but it is not clear how batch correction for missing proteins can be successfully done unless it is implicitly a form of imputation.

Survival Analysis: Gene expression was dichotomized using the maxstat approach. There is no multiple testing corrections performed when testing targets in multiple diseases. For example, Fig 2g. Given the large numbers of tests and calling out only specific targets in specific diseases, it is difficult to assess the statistical significance of these findings. Further, there are references for the use of maxstat that should be considered, vs a URL: e.g.,

Torsten Hothorn and Berthold Lausen. On the exact distribution of maximally selected rank statistics. *Computational Statistics & Data Analysis*, 43(2):121–137, June 2003

or

Berthold Lausen, Torsten Hothorn, Frank Bretz, and Martin Schumacher. Assessment of optimal selected prognostic factors. *Biometrical Journal*, 46(3):364–374, 2004.

Figure 3e: This applies to the PLIN1 finding, but likely applies to a number of other findings. In the case of PLIN1, it appears that the large log₂ ratio for rare SCC comes from the fact that most (188/228) common samples did not identify PLIN1. In these cases, it would be helpful to provide any information on presence of peptides associated with PLIN1 in the common cancers. It would be extremely interesting if this protein was only (or substantially) present in rare squamous tumors, potentially due to a deletion in "common" squamous tumors. However due to label-free characteristics it is possible that technical artifacts caused this difference (perhaps fewer proteins overall, run order, peptide interference).

Line 148: I believe it should refer to Supplementary Fig 5 only, unless Figure 4 is referring to batch correction.

Supplemental Figure 2a: It would be helpful to have a better contrast in color, since the numbers are difficult to read.

Supplemental Figure 2b: It is not clear if this is the best representation, as only tumor pairs are expected to have high correlation.

Supplemental Figure 2c: I do not know what GP Number stands for here, but "cumulative identified proteins" may be more informative.

Supplemental Figure 2g: Is this nomenclature used in the paper?

Supplemental Figure 3: This is difficult to understand. For instance, it appears that SPRR1A is high in normal esophagus from the human protein analysis, yet missing (it appears) in the pan-squamous cohort. Please clarify this finding.

Supplemental Figure 5c: There are several tissue types labeled with color (blue, red). Does this color indicate a particular relationship? It does in the case of Thyroid and Gallbladder (similar) but it's not clear with breast scc.

Figure 2 legend: "Top 20 variant proteins were labeled in the volcano plot." - I believe this should be top 20 most significant differences or something similar, not variant proteins.

Line 160,161: It is not clear if the term overrepresented is accurate. It appears to be the results of a wilcoxon test, therefore it is overexpressed or overabundant.

Line 196-197: It is not clear what data is used to assert that prognosis "were significantly affected by these DEPs".

Line 194-195: This appears to be a statement about other work, not specifically results from this cohort although it is not clear.

- Characterization of HPV-related SCCs

Lines 402-403: it seems likely that the positive rate above 54% is in the HPV+ patient cohort (as is HPV18 6.67), but it is not clear.

Lines 410-411: p53 loss occurred in HPV- cases as well. This would be an example where a statistic may provide more information (perhaps the frequency of p53 loss).

Line 411: "CDKN2A expression was highly correlated" - please include p value or correlation coefficient. Figure 6b does not include these numbers either.

Line 418: "We identified 8 patterns of differential pathway ..." - was tissue of origin included in the limma analysis for this? Was this driven by tissue of origin?

Fig 7d and line 465-466: Can you clarify if "P16 expression was strong positive in two cases" refers to the specific tissues selected for staining or in the proteomics cohort?

Line 481-482: It would be helpful if there is a test for agreement with the proteomics data and/or the highly expressed in thymus SCC.

Figure 7 (caption): Can you clarify whether or not the the same tissue was assessed for H&E, P63, EBER, P16, etc or representative tissues from each organ type depending on the stain?

Methods:

Clinical Sample Acquisition (582). Patients were excluded if they had advanced disease (perhaps this could include clinical stage for clarity). Also, "any condition that may influence the outcome evaluation". It would be helpful to indicate that some patients are missing outcomes. Presumably these were dropped from survival analysis, but may have been censored at time 0. It would be helpful to clarify how this was addressed.

Additionally, samples were collected from 2001 to present. The manuscript states "all patients provided written consent". This suggests that a general banking protocol was used to collect tissues, distinct from this specific study. It would be helpful to clarify this. In the case of the adenocarcinomas, it is explicitly stated that patients were consented prior to surgery.

Patients with adenocarcinomas were excluded if radiation was used (unclear if this was preoperative). Presumably this was not the case in the squamous cohort, was this evaluated?

Histological evaluation (594): Were ADC using the same pathology assessment/acceptance criteria (>80% viable tumor nuclei, >50% cellularity)?

Database Searching (640): What version of Uniprot was used? It may be present in a figure, but not stated in the methods. Also, were missed cleavages considered for trypsin identification? Modifications? More details here would be needed to understand how the searching was done.

MAD-based protein selection:

line 759-767: What does "the 20% bottom mad proteins of each organ were combined and duplicates removed" mean? It would seem reasonable to remove the bottom 20% of mad proteins due to lack of variability, but it's not clear what the term "combined" means here (averaged?).

line 785-794.: Why would the 1% bottom mad proteins be selected? These are the least variable (assuming smallest mad values are meant). It is not clear why the 1220 proteins were used to compute an average expression value (for what purpose). In particular, ordering by mad does not impact the random forest. Perhaps something different was meant, but it was not clear from the description. In general, it would also be good to indicate if all filtering and protein selection was done only on the training set (independent evaluation) or on both the training and testing sets.

line 721: "multiple testing correction using ..."

Line 811/812: Please define HPV16 as the "main type".

Discussion comments:

line 511: "involved in rare SCC initiation". It would be more accurate to describe them as regulators involved in rare SCC since it is a descriptive finding.

Reviewer #2 (Remarks to the Author): Expert in SCC proteomics

In the manuscript entitled "Proteomic Landscape of Pan-squamous Cell Carcinomas", Song et al. provided an interesting panorama of SCC proteomes from 333 patients and 17 sites using mass spectrometry label-free quantitation. By performing several group comparisons (SCC vs AC; common SCC vs rare SCC; HPV+ anogenital SCC vs HPV- anogenital SCC), the authors indicated signatures of differentially abundant proteins that have a prognostic significance and are able to modulate specific pathways. Four groups of SCC types were defined by clustering tumor proteomes and six subtypes of SCCs were determined based on immune enrichment of cell composition. Molecules from these groups may potentially modify biological pathways and are promising as druggable targets. Finally, a machine learning model was built to predict the tumor site of origin based on SCC proteomes and 3 proteins were selected for validation using IHC.

Overall, that is a detailed study that provided insights into the biology of a common cancer type (SCCs) by profiling the tumor proteomes followed by an extensive bioinformatic analysis. The significant number of patients evaluated is certainly a highlight of the study. Since large-scale analysis of tumors available in the literature have focused on DNA/RNA levels, describing the proteome and the potential functional implication is definitely novel and opens up new perspectives for the study of cancer. The description of a protein signature that can possibly define the origin of SCCs is also remarkable and revealed targets with potential to translate to the clinics. For instance, the authors provided a valuable resource to the scientific communities for further exploration. Considering the methodology, a noteworthy point is that the manuscript presents a deep characterization of SCCs achieved by implementing a strong and robust bioinformatic analysis. The proteomic results indicate a successful workflow of sample preparation, MS run and data analysis (~ 15,000 protein groups quantified for SCCs and ~10,000 for ACs), even though some additional information should be provided to assure the reproducibility and quality of the data.

In summary, the study is of major interest for the oncology and proteomics fields and is in-line with most articles published in Nature Communications. There are therefore some shortcomings that should be addressed.

Results

1. Line 109: The paper relies on proteomics data from a large group of SCC samples and the quality control of MS runs is certainly one of the main concerns to assure the reliability of bioinformatic analysis. However, I am not sure about QC analysis. How was the set of repeated samples selected for correlation (SFig. 2b)? Additionally, the criteria for QC of individual samples were not described in the text and it would be good to see some QC results for all MS runs. Plotting Spearman's correlation, counting valid values per sample or, if suitable, describing retention times or m/z for trypsin autolysis peaks across samples are suggested approaches.
2. Normalized protein intensities from pan-SCC and pan-AC were log₂ transformed. Even that is not always true, log transformation reduces the skewness of large-scale data and make it more closely to a normal distribution. If that is the case, it would be appropriate to use parametric tests for group comparison instead of the non-parametric analysis described in the manuscript (Wilcoxon, Kruskal-Wallis). Did the authors test the normality assumption of the data? Please comment.
3. SFig. 3: Not clear what is represented on this image.
4. Fig. 2d; SFig. 7: Not sure what NES means.
5. Lines 179 and 325: Besides using their own proteomics data for Cox analysis of DEPs between SCCs and ACs, 9 RNA databases from cancer tissues were used to assume the role of proteins in prognosis. Did the authors evaluate if transcripts are somehow correlated with proteomics data? Otherwise, the assumptions may not be true.
6. Fig. 5a: The authors should amend Subtype 2 name to FaSq.

7. Fig. 5d: Not mentioned in the main text.

8. Fig. 6b: How can the authors explain the high levels of protein pRB in HPV-infected tumors? Since the silencing of pRb by E7 viral protein produces a rise in p16, the abundance of pRB is not in agreement with what is reported in the literature or in this study. Also, SOX2 is not frequently associated with HPV infection and it would be appropriate to describe the rationale for including this protein in the analysis.

9. Fig. 6e: The error bars are too large and it is difficult to believe that there is a real difference between HPV+ and HPV- cases for these proteins, even with an adjusted p value. Maybe including the protein abundances for the non-anogenital HPV-negative SCC types would make the hypothesis of Fig.6f stronger.

10. Line 459: I didn't get the point of why the authors evaluated the 3 markers by IHC. If a signature of 19 proteins was accurately able to discriminate SCC tumor based on their origin (but the way, this information is not stated in the main text), why the 3 proteins were selected for validation? I understand the validation phase is an important step in large-scale experiments, but analyzing the 3 proteins alone does not make sense in the context of the classifier and did not make the proteomic data stronger.

Discussion

11. It is not necessary to extensively re-state the key findings in the discussion.

Methods

12. Line 580: References should be provided for WHO classification and TNM system.

13. Fundamental information is missing in the proteomic workflow and it is difficult to judge the reproducibility and quality of the methods employed. Please provide additional information.

Sample preparation

a. Why did the authors add an acetone precipitation step in the FASP protocol? FASP did not take care of contaminant removal? The authors should provide the appropriate references.

b. How was alkylation and quenching performed?

c. The authors declare that a trypsin-to-protein ratio of 1:50 was used for digestion, but there is no information on protein quantification. How were protein levels determined? The authors also state that "Target on-column load was 200ng total peptide per injection", indicating that peptides were also quantified. Please clarify.

d. It is appropriate to present specifications of the trypsin used for digestion.

e. What is MS water? How were peptides acidified to stop digestion?

LC-MS/MS

f. Some fundamental aspects of the MS runs are missing, like m/z range, mode of data acquisition (DDA? DIA?), resolution, etc.

g. Although the identifier is provided in the text, the repository where raw files are deposited is not informed and data could not be accessed.

MS search

h. Details for the Uniprot library should be provided, including download date, number of residues considered. Were SwissProt and TrEMBL entries considered?

i. A fragment ion tolerance of 0.05Da was used. Why did the authors use such a restrictive cut-off? I am afraid that important protein identification was lost.

j. How did the authors handle contaminants?

k. I could not find any information about variable and fixed modifications, or the retention time window considered.

Data analysis

l. Replacing missing values engenders intense debate in the scientific community. Are there any reasons why missing values were replaced? Maybe keep the original data would be appropriate and statistics would take care. Have other approaches been tested for data imputation to assure that replacing by one-tenth of the minimum intensity is the most suitable strategy?

14. I was wondering whether the separation of SCCs and ACs in PCA before batch effect correction just reflects their distinct biological characteristics. How can the authors be sure that the separation was a batch effect?

15. Not clear if IHC was performed in the same cohort as proteomics.

16. For HPV grouping, how the authors defined if HPV16 is the main type (group 2) or not (group 3) in multiple infections?

Reviewer #3 (Remarks to the Author): Expert in SCCs

Understanding the molecular pathways driving histologically similar cancers across anatomic sites may provide new treatment paradigms that historically have been site-specific. Differences in mutational patterns and gene expression profiles between squamous cell carcinomas (SCC) and adenocarcinomas (AC) arising across anatomic sites have been well described using common resources such as TCGA. However, the proteomics landscape of SCC across anatomic sites has not been previously investigated in large numbers of tumors. The current manuscript describes proteomic patterns in 333 treatment-naïve squamous cell carcinoma (SCC) tissues obtained from 17 organ sites and 69 treatment-naïve adenocarcinoma (AC) tissues obtained from 7 organ sites from a single university-based hospital in Shanghai, China. Proteomic characterization of tissues was conducted using a mass spectrometry-based approach validated using tissue microarray (TMA) immunohistochemistry (IHC).

Major findings include the elucidation of pathways differentiating SCC from AC (keratinization, glucose metabolism and extracellular matrix), molecules within those pathways associated with disease prognosis, and proteomic clusters/immune subtypes that may represent potential druggable targets. The resulting data repository will serve as a unique and valuable shared resource for investigators to use in the future.

Methods and results are described in great detail, yet there are some key points that should be clarified and/or expanded upon.

Case selection and classification:

- The methods state that the cases were randomly selected. How was this accomplished, and what percentage of the total SCC cases treated in the 18-year range do the cases included in the current study represent? Exclusion criteria are presented, yet it's unclear how many patients were excluded for the reasons listed.
- While the overall number of tumors (n=333) is substantial, the numbers of samples available per anatomic site ranged from 10-22 for SCC and 8-12 for AC. Therefore, inferences drawn for specific sites are limited by small sample size. This point should be added to the discussion.
- The classification of rare versus common tumors is unclear. The authors state that the WHO Classification of Tumors was used, yet there is no reference, and some cancers seem to be misclassified. For example, SCC of the vagina is very rare (i.e. incidence is less than 6 per 100,000), yet it is included here as a common cancer.

- As the authors point out in the Background, metastatic SCC's (or primaries with elevated metastatic potential) are an important clinical challenge. However, only primary SCC's were included in this case series, and no information was provided on whether or not patients developed metastases during follow-up. This is an important design limitation that should be discussed.

Patient follow-up and survival analysis:

- No information is provided on the average length of follow-up (and range), as well as whether or not patients were lost to follow-up, and if so, how they were handled in the analysis. It is also not clear that the proportional hazards assumption was assessed.
- Why were patient age and gender (as well as other patient characteristics such as stage at diagnosis) not considered as potential covariates in the multivariable modeling, along with the three covariates stated (protein expression, organ and histology)?

Results:

- In general, it was difficult to follow the results section given the sheer number of figures, figure panels and supplementary materials. The figures were very detailed, as were the supplementary data (often patient-level data files). In many cases, it would have been helpful to create summary tables that allow the reader to directly compare percentages between groups and better ascertain the statistical significance of the observed results. For example, for Figure 1b- it would be helpful to show the information in tabular form so that percentages of samples across anatomic sites could be more directly compared with respect to tissue characteristics such as stromal score and keratinization; statistical significant testing could be used to determine which differences are most likely to be real and not due to chance. The raw data are included in supplementary Table 1, but a table showing the percentages across groups would be most helpful to view.
- Regarding the HPV results, it is difficult to glean from Figure 6e whether the protein patterns depicted are specific to HPV 16. It would be helpful to present HPV type-specific prevalence by tumor type in tabular form, and then present the percentages of each of the five HPV groups defined in Fig 6c that express proteins corresponding to the different pathway groups of interest defined in 6d. Were HPV16 E6 and E7 proteins detected in any of the SCC samples?
- The survival analysis described in Fig 2e is intriguing, given that these proteomic features may be useful prognostic markers. It appears as if fewer proteins were predictive of survival in the current PanSCC dataset compared to a majority of the TCGA datasets included in Fig 2e. Could this be a function

of sample size? It would be helpful if the Hazard Ratios and 95% confidence intervals were provided to better interpret the magnitude and precision of these estimates.

REVIEWER COMMENTS

Reviewer #1 (Remarks to the Author): Expert in bioinformatics

This paper is an important work that highlights the generation of a very large dataset consisting of 333 patients with squamous cell tumors from 17 different organs. The authors use label-free, mass spectrometry-based proteomics to characterize this cohort and compare with a similarly generated cohort of 69 patients with adenocarcinomas from 7 different organs. The generation of this dataset is a noteworthy and significant accomplishment as a mechanism for comparing the proteomics of squamous tumors from multiple organs consistently (same methodology, same instrument). In addition, many of the tissue types included are rare diseases. While many diseases have been profiled in one form or another, this is a comprehensive look across squamous cell carcinomas using proteomics that is similar to the PanCancer analysis of squamous cell from DNA/Methylation/RNA.

Overall this work reports many interesting findings related to the similarities among squamous cell cancers, differences between squamous cell and adenocarcinoma, and differences between common and "rare" squamous cell carcinomas in the context of proteomics. Since this work spans many different squamous tumor types, it is difficult to compare with the many existing findings from the literature. However, Figure 1d provides a nice summary of known targets within the proteomics data.

The paper is primarily a discovery and descriptive paper of many characteristics of squamous cell carcinomas. As such, the work provides both a survey of the proteomics landscape and a resource for other researchers to utilize. The paper provides many complex analyses of the data, providing both the detailed results as supplemental data and figures summarizing the results. Overall, the methods are appropriate although there are a few areas in which there is not enough detail to fully understand the approach taken (see below for specific comments that are likely to be resolved through additional details).

Response: We appreciate the reviewers for the positive review and valuable comments. We have revised the manuscript according to the comments. The point-to-point responses were as follows.

Suggestions for improvement:

Q1.

It might be helpful to include p values when describing differences that are qualitatively described within the manuscript. This would allow readers to assess the significance of the observation. For instance (line 100: significant differences across organs) could include a statistical test and associated p value to strengthen the conclusion there are statistically significant differences. Another example: Stromal vs ESTIMATE "showed a consistent trend" - can this be quantified?

Response: Thank the reviewer for bringing this to our attention. We are sorry for not showing p values when describing differences in the manuscript. We systematically searched all these kinds of inaccurate descriptions and added statistical tests and p values in the corresponding part of the revised manuscript. We listed four representative changes in the following, and the others were answered in related questions.

- a) Line 99 (revised manuscript): As the statistical analysis was shown in **Supplementary Fig. 1s**, we performed Kaplan-Meier survival analyses with log-rank tests. We have added p values in corresponding results in the revised manuscript to show the survival differences among 17 SCCs.
- b) Line 104 (revised manuscript): We calculated the correlation between Stromal Score by ESTIMATE analysis and Stromal Ratio by pathological evaluation (Spearman correlation, $R = 0.31$, $p < 0.001$; **Figure RL 1**), suggesting a consistent trend between Stromal Score and Stromal Ratio.

Figure RL 1 A scatterplot showed the correlation between Stromal Score (x axis) and Stromal Ratio (y axis). Spearman correlation.

- c) Line 267 (revised manuscript): As the statistical analysis was shown in **Fig. 3i**, we performed Spearman's correlation between FOXO1 and RUNX2 expression. In the revised manuscript, we labeled the R and *p* value in corresponding results.
- d) Line 470 and 476 (revised manuscript): For these two sentences “Immunostaining of PRKCE was significantly different among 17 SCCs, and showed an overall high expression in cervical and vagina SCCs” and “In agreement, we noted a high proportion of tumor specific positive SLC27A1 staining in gallbladder and pancreatic SCCs”, we showed the immunohistochemistry score for these two markers in pan-SCC cohort, and did Kruskal-Wallis tests (both *p* < 0.0001) to show the differential expression among 17 SCCs (**Figure RL 2**).

Figure RL 2 Boxplots showing the immunohistochemistry score of PRKCE (**a**) and SLC27A1 (**b**) in 17 SCCs. Kruskal-Wallis test.

Q2.

In the case of the overall survival/disease free survival there are cohorts collected at different times having different time to follow up. Therefore, comparing outcomes may be difficult, as censoring may be primarily a late or early event, and may be influenced by date of collection (due to therapy). There does not appear to be multiple testing correction methods applied to survival *p*-values. Given the large number of exploratory tests for outcomes, multiple testing correction may be appropriate to consider.

Response: Thank the reviewer very much for giving this critical comment. Due to the complexity of the pan-SCC cohort, we agree that the multiple testing correction is necessary for the survival analysis of the pan-SCC cohort.

The pan-SCC cohort has its characteristics, as it includes 333 patients originating from 17 different organs and ~20 cases per organ. Therefore, we explored the prognostic value of differentially expressed proteins in the pan-SCC cohort. We included age, gender, stage, histology, organ, and

protein expression as covariates in the multivariate Cox proportion hazard model and did multiple testing correction (BH adjusted) according to your comments in the revised manuscript. After this strict calculation, we determined molecules with prognostic statistical significance, including RPL12 (Ribosome; (95% CI: 0.45-0.82); $p = 0.036$, BH adjusted), ATM (Cell cycle; (95% CI: 0.41-0.84); $p = 0.049$, BH adjusted) with good prognostic value, and SERPINE1 (P53 downstream pathway; (95% CI: 3.6-210); $p = 0.0135$, BH adjusted) and MMP19 (Extracellular matrix; (95% CI:1.1-1.2); $p = 0.0178$, BH adjusted) with poor prognostic value. Please see **Supplementary Fig.6b** and **Fig.4e** in the revised manuscript.

Specific Items:

Q3.

Lines 125-126: It would be helpful in showing KRT5 and TP63 to again find a quantification of the assertion that they are "high and ubiquitous". There is some variation in KRT5, at least. It appears that many thyroid cancers have lower KRT5, so it might be helpful to summarize observations with quantification (perhaps by tissue of origin). For instance, how does the variability in AKT1 or KRT5 compare (for instance, what quantile of variability using MAD as done elsewhere)?

Response: Thank the reviewer for these comments. We have calculated the coefficient of variation (CV) and median absolute deviation (MAD) for 333 SCCs separately according to your suggestions (**Table RL1**). As the CV values presented the data variation nicely, we chose the CV value and labeled the CV on the left side of **Fig. 1d (Figure RL 3)**. AKT1, TP63, and KRT5 are the top three least variable proteins.

Table RL 1 CV and MAD values of known targets among 333 patients.

Symbol ^a	MAD	CV
AKT1	0.68647484	0.05975266
TP63	0.82665095	0.06853299
KRT5	0.82128638	0.08304692
EGFR	1.06819925	0.08955909
YAP1	1.09841542	0.15734491
KMT2D	1.06123182	0.16241598
ROBO1	0.69494423	0.1942276
CD274	0.99106533	0.19731426
CDKN2A	1.64104803	0.3113152
AKT3	2.02498803	0.43342094

KMT2C	2.16721225	0.43479109
SOX2	2.16062882	0.49392111
LRP1B	2.70962462	0.50980926
TP53	0	0.6673242
NOTCH1	0	0.67916089
FGFR3	0	0.73940562
ZNF3	0	0.74413864
PTEN	0	0.78458331
ZNF750	0	0.78708465
CSMD1	0	0.81595286

^a Proteins were in ascending order by CV value.

Figure RL 3 The protein abundance of SCC diagnostic markers and highly variant genes, the corresponding CV for each marker among 333 SCCs was labeled on the left side.

Q4.

On the basis of the methods and the color gradient in the figure 1d, blue is "missing" data. If this is the case, it may be worth pointing this fact out. FGFR3, PTEN and TP53 may be mutated and thus either not sufficiently translated or translated with alternate forms. Given the (relatively) low expression of those proteins (FGFR3/PTEN) in samples, it may be worth exploring this further.

Response: Thank the reviewer for the comments. To answer your question, we performed targeted sequencing of *TP53*, *PTEN*, and *FGFR3* (including SNV and InDel) for lung SCC, esophageal SCC, and cervical SCC (time of surgery were all in 2015). The summarized mutation information is shown in **Figure RL 4a**.

- ***TP53* mutation and expression in lung SCC:** The mutation rate of *TP53* is 90% (18/20) and p53 expression rate is 55% (11/20) in lung SCC (**Figure RL 4a and Table RL 2**). Then, we

calculated the protein expression level between mutated and wild type in lung SCCs, no statistical difference was found between mutated and wild type SCCs (Wilcoxon rank-sum test, $p = 0.55$ respectively; **Figure RL 4b**).

- ***TP53* mutation and expression in esophageal SCC:** The mutation rate of *TP53* is 83.3% (15/18) and p53 expression rate is 88.9% (16/18) in esophageal SCC (**Figure RL 4a and Table RL 3**). The protein expression level between mutated and wild type in esophageal SCCs showed no statistical difference (Wilcoxon rank-sum test, $p = 0.41$ respectively; **Figure RL 4c**).
- ***TP53* mutation and expression in cervical SCC:** The mutation rate of *TP53* is 15% (3/20) and no p53 expression was detected in cervical SCC (**Figure RL 4a and Table RL 4**).
- ***PTEN* and *FGFR3* mutation and expression:** The mutation rate of *PTEN* and *FGFR3* was 35% (7/20) and 10% (2/20) in lung SCC. No mutation was found in esophageal and cervical SCCs for both *PTEN* and *FGFR3*. *PTEN* and *FGFR3* showed no expression in lung SCC, esophageal SCC, and cervical SCC.

From the above analysis, we cannot conclude how mutated *FGFR3*, *PTEN*, and *TP53* affect the protein expression. One head and neck squamous cell carcinoma study (**PMID:33417831**) showed that missense mutations in *TP53* were associated with increased p53 mRNA and protein abundance, suggesting that specific *TP53* mutations might endow oncogenic gain of function to this protein. Due to the relatively small sample size of a certain type of SCC we tested, a large-scale genome-proteome wide study is needed to better illustrate how gene mutation status affect the protein expression in SCCs.

Figure RL 4 Summarized mutation information associated with protein expression. **a** *TP53*, *PTEN*, and *FGFR3* mutation status in SCC originating from esophagus, lung, and cervix. Comparisons of p53 protein abundance between *TP53* mutated and *TP53* wild type samples in lung SCC (**b**) and esophageal SCC (**c**).

Table RL 2 *TP53* mutation and expression in Lung SCC

TP53	Mutation	Wild type
positive	9 (45%)	2 (10%)
negative	9 (45%)	0 (0%)

Table RL 3 *TP53* mutation and expression in Esophageal SCC

TP53	Mutation	Wild type
positive	13 (72.2%)	3 (16.7%)
negative	2 (11.1%)	0 (0%)

Table RL 4 *TP53* mutation and expression in Cervical SCC

TP53	Mutation	Wild type
positive	0 (0%)	0 (0%)
negative	3 (15%)	17 (85%)

Q5.

Identification of proteins may relate back to tissue specificity so it would be helpful to know if the total numbers were significantly different across tissues. Specifically, on line 115 "no major differences in the coverage between the 17 SCCs". I would encourage a test of this assertion. Figure 1c suggests there are significant differences. Were samples run by tissue of origin? If they were randomized, then the differences in total protein could be biologically meaningful and worth further investigation. This is particularly true with copy number changes that have been seen in squamous cell cancers.

Response: Thank the reviewer for the comments. It is our fault to state "no major differences in the coverage between 17 SCCs" with no statistics. Our answer is as follows.

- a) We performed mass spectrometry profiling randomly, not in the order of organs.
- b) On the one hand, a Kruskal-Wallis test on samples of 17 organs was conducted, with a $p < 0.0001$, suggesting a significant difference among 17 SCCs. On the other hand, we conducted a pairwise analysis (Bonferroni) on the samples of 17 organs. A total of 136 comparisons were made between 17 organs, of which 88 (64.7%) had no difference and 48 (35.3%) had

differences. Based on the above analysis, we concluded that there are differences in the protein identification number of 17 squamous cell carcinomas in general, consistent with your comments. Our previous statement is inaccurate, and the corresponding results have been modified (line 118-119).

- c) The thymic SCC and nasopharyngeal SCC are the squamous cell carcinoma with the maximum identification number. As you mentioned, this is probably due to copy number changes in SCCs. We agree that the copy number changes affect protein identification, as tumor samples were identified with more proteins than tumor adjacent normal tissues (**PMID: 33417831, 32649877**). Copy number changes were frequently detected in SCCs (**PMID: 24686850, 31395880**). Moreover, the cell density of the thymus and nasopharyngeal SCC is high (as evaluated by HE staining), which may be another reason for the high number of protein identification in these organs.

Q6.

Batch correction: When batch correction was done, there was missing data present. Was the missing data imputed in the batch correction or left missing? That is, the missing data is imputed as a low expression however it is possible that batch correction could increase the level of expression of these missing values? This would presumably introduce an unintended batch effect in that case?

Response: Thank the reviewer for the comments.

- In the input matrix of batch correction, the protein expression value was the original value, the expression value of missing expressed protein was 0, and there was no blank value.
- After batch effect correction, negative values appeared in the matrix. We reset the negative value to 0 (**PMID:22257669**).

As **Figure RL 5a** showed, the 293T samples of the SCC and AC cohort were separately clustered before batch effect correction. Batch effect correction was done following the steps described above, and the PCA showed a remarkable similarity between these two batches after batch correction (**Figure RL 5b**). Therefore, we think this method reduced the introduction of new batch effects.

Figure RL 5 The PCA analysis showing the 293T samples of the SCC and AC cohort. **a** before batch correction. **b** after batch correction.

Q7.

Commonly expressed proteins in SCC or AC: It is not clear to me, when 5130 commonly expressed SCC proteins and 4845 commonly expressed AC proteins yield 5838 common proteins of SCC or AC. Is this perhaps the union of the two sets? It would seem more common to assume that the intersection of the two is used as the basis for common proteins. Overall, there are two potential effects: 1) presence of proteins in one of AC/SCC and 2) expression differences among proteins in both AC/SCC. It is not clear which (or both) of these is being addressed. It is expected that AC and SCC have unique proteins (of interest in and of itself) but it is not clear how batch correction for missing proteins can be successfully done unless it is implicitly a form of imputation.

Response: Thank the reviewer for the comments. With the purpose of comparing the differences between ACs and SCCs in **Fig. 2**, we firstly selected commonly expressed proteins in SCCs or ACs using a strict criterion. Then, we combined these two commonly expression protein lists together (union set), and excluded proteins only expressed in ACs or SCCs. The detailed process are as follows (**Figure RL 6**):

- a) We chose the proteins expressed in more than 50% cases of one certain SCC as the commonly expressed proteins in the certain SCC. Then, the interaction of 17 groups of commonly expressed proteins was defined as the commonly expressed proteins in SCC (5130 proteins). We did the same thing for 7 ACs (4845 proteins).
- b) The union set of commonly expressed proteins in SCC (5,130 proteins) and AC (4,845 proteins)

contained 5,914 proteins, and the interaction contained 4,061 proteins. A total of 1,069 proteins were only commonly expressed in SCCs, and 46 proteins were only detected in SCCs (not detected in ACs). A total of 784 proteins were only commonly expressed in ACs, and 30 proteins were only detected in ACs (not detected in SCCs).

- c) To compare the differences between SCCs and ACs, we removed a total of 76 proteins only expressed in SCCs or ACs, and a total of 5838 proteins were obtained.

In this case, no missing values were in the data matrix, and the batch effect correction was successfully done.

Figure RL 6 The Venn diagram showing the comparison of commonly expressed proteins in SCCs versus ACs.

Q8.

Survival Analysis: Gene expression was dichotomized using the maxstat approach. There are no multiple testing corrections performed when testing targets in multiple diseases. For example, Fig 2g. Given the large numbers of tests and calling out only specific targets in specific diseases, it is difficult to assess the statistical significance of these findings. Further, there are references for the use of maxstat that should be considered, vs a URL: e.g.,

Torsten Hothorn and Berthold Lausen. On the exact distribution of maximally selected rank statistics. *Computational Statistics & Data Analysis*, 43(2):121–137, June 2003 or Berthold Lausen, Torsten Hothorn, Frank Bretz, and Martin Schumacher. Assessment of optimal selected prognostic factors. *Biometrical Journal*, 46(3):364–374, 2004.

Response: We thank the reviewer for the suggestions. Nine TCGA cohorts were originally analyzed by Kaplan-Meier analysis. In the revised version, we did the multiple testing correction (BH adjusted) and modified **Fig. 2e**, **Fig. 2g**, **supplementary Fig. 6**, and **supplementary Table 2d** with adjusted p values. **Fig. 2g** shows that kinase and transcription factors played a consistent prognostic role with their downstream targets in SCC or AC, giving an insight that the function of

keratinization, glucose metabolism, and extracellular matrix pathway could be consistent or opposite in SCCs or ACs. We want to present this phenomenon, though only in specific tumor types. Now we moved original **Fig. 2g** to supplementary **Fig. 6c**. This finding should be explored in future studies.

References were added to the revised paper. Thank the reviewer again for the recommendation.

Q9.

Figure 3e: This applies to the PLIN1 finding, but likely applies to a number of other findings. In the case of PLIN1, it appears that the large log₂ ratio for rare SCC comes from the fact that most (188/228) common samples did not identify PLIN1. In these cases, it would be helpful to provide any information on presence of peptides associated with PLIN1 in the common cancers. It would be extremely interesting if this protein was only (or substantially) present in rare squamous tumors, potentially due to a deletion in "common" squamous tumors. However due to label-free characteristics it is possible that technical artifacts caused this difference (perhaps fewer proteins overall, run order, peptide interference).

Response: Thank the reviewer for the critical comments. From our result, PLIN1 was highly expressed in rare SCCs and was not detected in the most of common SCCs (188/288) as you mentioned (**Figure RL 7a**). This is really interesting.

- We firstly checked the database searching result and no PLIN1 peptide were found in common SCC cases with no PLIN1 expression. As you mentioned, we cannot tell the gene status and potentially a deletion in common SCCs.
- Then, we ordered the FISH probe for PLIN1 (Empire Genomics Corp, PLIN1-20-OR) and tested the PLIN1 copy number in ten cases for each SCC. No deletion was found in common SCCs. Interestingly, we detected gene amplification in 3 anal SCCs (3/10, **Figure RL 7b**). These results were updated in the revised manuscript.

In this case, we think that the PLIN1 amplification is probably the reason for high expression in rare SCCs. A large-scale study will be needed to explore further the significance of PLIN1 in SCC initiation and progression.

Figure RL 7 The protein expression and gene copy number status of PLIN1. **a** the PLIN1 protein expression in 17 SCCs, **b** Representative PLIN1 fluorescence in situ hybridization signal patterns (red signals = PLIN1, green signals = CEP15), left, this case was scored negative for PLIN1 amplification. PLIN1/nucleus ratio = 2.52; right, this case was scored as positive for PLIN1 amplification. PLIN1/nucleus ratio = 6.2.

Q10.

Line 148: I believe it should refer to Supplementary Fig 5 only, unless Figure 4 is referring to batch correction.

Response: We appreciate this comment. Supplementary Fig 4 is also referring to batch correction. Supplementary Fig 4 presents the principal component analysis of quality control samples and patient samples of pan-SCC and pan-AC cohorts before and after batch correction. So, we put both Supplementary Fig 4 and 5 in line 149 (revised manuscript).

Q11.

Supplemental Figure 2a: It would be helpful to have a better contrast in color, since the numbers are difficult to read.

Response: Thank the reviewer for bringing this to our attention. As these quality control runs had a high correlation (the average correlation coefficients were 0.90), the previous figure had a poor contrast (color bar: 0-1, **Figure RL 8a**). According to your suggestions, we have changed this supplementary Fig 2a to a new one with good contrast in color (color bar: 0.8-1, **Figure RL 8b**).

Figure RL 8 Longitudinal quality control of mass spectrometry using tryptic digest of HEK293T cells by representing the pairwise Spearman's correlation coefficients of the samples. **a** color bar: 0-1, **b** color bar: 0.8-1.

Q12.

Supplemental Figure 2b: It is not clear if this is the best representation, as only tumor pairs are expected to have high correlation.

Response: Thank the reviewer for this comment. To confirm the stability of our LC-MS/MS platform, we tested the commercial 293T cell as quality control runs, showing a high correlation (average, 0.9) and thus demonstrating the consistent stability of the platform. Moreover, **Supplementary Fig. 2b** shows the correlation of 7 replicate samples of 2 bladder, 2 gallbladder, 2 breast, 2 thymus, 2 pancreatic, 2 penis, and 2 perineum SCCs. The repetitive analysis of 7 replicate samples was performed in the middle and at the end of the project, showing good reproducibility, with a high level of correlation (average, 0.92). The replicate samples further confirmed the stability of LC-MS/MS platform.

Repeated runs with the same samples have high correlation as we expected. Also, this result is consistent with the t-SNE analysis (**Fig. 4a**), showing that samples from the same organ tended to cluster together. We also included a new Spearman's correlation for all 333 samples in the **Supplementary Fig.2c**, showing a high correlation within cancer types.

Q13.

Supplemental Figure 2c: I do not know what GP Number stands for here, but "cumulative identified proteins" may be more informative.

Response: Thank the reviewer, we agree that "cumulative identified proteins" is more informative than "GP Number" in **Supplementary Fig. 2c**. In the revised **Supplementary Fig. 2e (original Supplementary Fig. 2c)**, we have changed the "GP Number" to "cumulative identified proteins".

Q14.

Supplemental Figure 2g: Is this nomenclature used in the paper?

Response: Thank the reviewer for bringing this to our attention and we only mentioned "**Supplementary Fig. 2g**" in the submitted manuscript. In the revised version, we have added this nomenclature when referring these protein sets in line 122, 156, 230, and 281.

Q15.

Supplemental Figure 3: This is difficult to understand. For instance, it appears that SPRR1A is high in normal esophagus from the human protein analysis, yet missing (it appears) in the pan-squamous cohort. Please clarify this finding.

Response: Thank the reviewer for this comment. Cancer owns its specific proteomics, such as overexpressing of sustaining proliferative signaling, evading growth suppressors, resisting cell death, inducing angiogenesis, and other new critical characteristics (**PMID: 10647931, 21376230, 35022204**). Thus, cancer samples will lose the expression of original tissue markers as reported (**PMID: 32649877**). Therefore, to prove the high tumor purity of this pan-SCC cohort, we downloaded normal tissue signature protein (tissue enriched proteins) lists from Human Protein Atlas (<https://www.proteinatlas.org/humanproteome/tissue/tissue+specific>) and presented the proteins of specific organs that had lost expression in corresponding SCCs in **supplementary Fig. 3**. For instance, SPRR1A, a cross-linked envelope protein of keratinocytes, is an esophageal signature protein (**Figure RL 9a**). It was reported that it had lost expression in esophageal SCC compared to the matched normal tissue samples (**Figure RL 9b**). As is shown in **Figure RL 9c**, SPRR1A had lost expression in esophageal SCC. In this case, this proved the tumor purity of the pan-SCC cohort to some degree. We have modified our manuscript to make this clearer.

Figure RL 9 SPRR1A expression in normal tissues, esophageal cancer vs matched normal tissues, and in pan-SCC cohort. **a** SPRR1A expression in six normal tissues, including esophagus, liver, kidney, testis, cerebral cortex, and lymph node. **b** RT-PCR analysis showed the differential expression of SPRR1A and other proteins in eight pairs of esophageal cancer tissue samples and the matched normal tissue samples. (PMID: 14647409 DOI: 10.1038/sj.onc.1207218.) **c** SPRR1A expression in pan-SCC cohort.

Q16.

Supplemental Figure 5c: There are several tissue types labeled with color (blue, red). Does this color indicate a particular relationship? It does in the case of Thyroid and Gallbladder (similar) but it's not clear with breast SCC.

Response: Thank the reviewer for the comments, and it is indeed a little confusing. We labeled Gallbladder SCC, Gallbladder AC, and Breast SCC with red in **Supplementary Fig. 5C**, as these three cancers grouped in the PCA (**Supplementary Fig. 5B**). In the revised **Supplementary Fig. 5c**, we changed the color to black.

Q17.

Figure 2 legend: "Top 20 variant proteins were labeled in the volcano plot." - I believe this should be top 20 most significant differences or something similar, not variant proteins.

Response: Thank the reviewer for the comments. We have changed the figure legend to "significantly differentially expressed proteins".

Q18.

Line 160,161: It is not clear if the term overrepresented is accurate. It appears to be the results of a Wilcoxon test; therefore, it is overexpressed or overabundant.

Response: Yes, we agree and we have changed the “overrepresented” to “overexpressed”. Thank the reviewer for the recommendation.

Q19.

Line 196-197: It is not clear what data is used to assert that prognosis "were significantly affected by these DEPs".

Response: Thank the reviewer for bringing this to our attention. We have modified this sentence and added "DEPs in ECM, glucose metabolism, and keratinization (**Fig. 2e**)" to make it clear.

Q20.

Line 194-195: This appears to be a statement about other work, not specifically results from this cohort although it is not clear.

Response: Yes, it is a statement of other works and we have moved these findings to Discussion part.

Characterization of HPV-related SCCs:

Q21.

Lines 402-403: it seems likely that the positive rate above 54% is in the HPV+ patient cohort (as is HPV18 6.67), but it is not clear.

Response: Thank the reviewer for the comments. The total HPV (not type specific) infection rate is ~80%, and HPV16 infection rate is ~54%. This sentence is ambiguous in the submitted manuscript. We have modified this sentence to make it clearer.

Q22.

Lines 410-411: p53 loss occurred in HPV- cases as well. This would be an example where a statistic may provide more information (perhaps the frequency of p53 loss).

Response: Thank the reviewer for the suggestion. We have summarized the p53 loss in HPV positive and negative patients (**Table RL 5-7**), and labeled the frequency of p53 loss in the revised manuscript. As you mentioned, p53 loss occurred in HPV negative cases as well.

In anogenital SCCs (5 sites, **Table RL 5**):

- a) For anal, cervical, and vaginal SCCs, p53 loss occurred in all samples.
- b) For penis SCC, p53 was expressed in all samples.
- c) For perineum SCC, we did a Fisher's exact test and did not get a significant difference between HPV positive and HPV negative concerning p53 expression status ($p = 0.2018$, **Table RL 6**).

In non-anogenital SCCs (12 sites, **Table RL 7**):

- a) Four SCCs, including oral, throat, thymus, and thyroid SCCs, are lost p53 expression completely.
- b) Pancreatic SCC showed an 80.95% (17 cases) loss of p53 expression.
- c) For gallbladder, lung, nasopharyngeal, bladder, and esophageal SCCs, they had a 15% to 45% p53 loss rate.
- d) Two SCCs, including breast and skin SCCs, were expressed in all samples.

Table RL 5 p53 expression in HPV positive and negative anogenital SCC patients.

	HPV positive		HPV negative	
	p53 loss	p53 positive	p53 loss	p53 positive
Anus (10)	10 (100%)	0 (0%)	0 (0%)	0 (0%)
Cervix (21)	20 (95.24%)	0 (0%)	1 (4.76%)	0 (0%)
Penis (22)	0 (0%)	11 (50%)	0 (0%)	11 (50%)
Perineum (20)	1 (5%)	13 (65%)	2 (10%)	4 (20%)
Vagina (21)	20 (95.24%)	0 (0%)	1 (4.76%)	0 (0%)

Table RL 6 p53 expression in HPV positive and negative Perineum SCCs.

	p53 loss	p53 positive	Fisher's exact test p value
HPV positive	1 (5%)	13 (65%)	0.2018
HPV negative	2 (10%)	4 (20%)	

Table RL 7 p53 expression in non-anogenital SCC patients.

	HPV negative	
	p53 loss	p53 positive
Oral (22)	22 (100%)	0 (0%)
Throat (20)	20 (100%)	0 (0%)
Thymus (21)	21 (100%)	0 (0%)

Thyroid (13)	13 (100%)	0 (0%)
Pancreas (21)	17 (80.95%)	4 (19.05%)
Gallbladder (20)	9 (45%)	11 (55%)
Lung (20)	9 (45%)	11 (55%)
Nasopharynx (20)	8 (40%)	12 (60%)
Bladder (22)	7 (31.82%)	15 (68.18%)
Esophagus (20)	3 (15%)	17 (85%)
Breast (20)	0 (0%)	20 (100%)
Skin (20)	0 (0%)	20 (100%)

Q23.

Line 411: "CDKN2A expression was highly correlated" - please include p value or correlation coefficient. Figure 6b does not include these numbers either.

Response: Thank the reviewer for the comments. We have calculated the CDKN2A expression differences between HPV negative and HPV positive patients in all five anogenital SCCs; no statistical significance was found (Wilcoxon rank-sum test, $p = 0.9330$; **Figure RL 10 left**). Then we calculated by organ individually. As anus, cervix, and vagina had a high HPV infection rate (100%, 95.2%, and 95.2%), we only calculated the CDKN2A expression differences between HPV negative and HPV positive patients in the penis (HPV infection rate: 50%) and perineum (HPV infection rate: 70%). While no difference was found in the penis (Wilcoxon rank-sum test, $p = 0.5619$; **Figure RL 10 middle**), HPV positive perineum SCC showed a higher CDKN2A expression than HPV negative perineum SCCs (Wilcoxon rank-sum test, $p = 0.0153$; **Figure RL 10 right**).

After calculation, we think the statement "CDKN2A expression was highly correlated with HPV infection" is not accurate. We have revised the sentence to "CDKN2A expression was higher in HPV positive perineum SCC than HPV negative perineum SCC".

Figure RL 10 Scatter plots showed the CDKN2A expression comparison between HPV negative and HPV positive patients in all five anogenital SCCs (**left**), SCC of penis (**middle**), and SCC of perineum (**right**). Wilcoxon rank-sum test.

Q24.

Line 418: "We identified 8 patterns of differential pathway ..." - was tissue of origin included in the limma analysis for this? Was this driven by tissue of origin?

Response: Thank the reviewer for the comments. We included the tissue of origin in the limma analysis for eight patterns of the differential pathway, and we think it is more HPV driven patterns. In this part, we combined all anogenital SCCs, as these organs belonged to the same system and were HPV infection-related. These analyses indicated that HPV16 infection may lead to active inositol phosphate catabolic process and immune evasion, participating in HPV16+ SCC carcinogenesis.

Interestingly, Inositol phosphates were reported promoting HIV-1 assembly and maturation to facilitate viral spread in human CD4+ T cells (**PMID: 33476323**). Multiple isomers of inositol phosphate were found in Epstein-Barr-virus- transformed (T5-1) B-lymphocytes and may be related with cell transformation or proliferation (**PMID: 1660712**). We hypothesize Inositol phosphate catabolic process probably related to HPV16 related tumorigenesis.

Q25.

Fig 7d and line 465-466: Can you clarify if "P16 expression was strong positive in two cases" refers to the specific tissues selected for staining or in the proteomics cohort?

Response: Thank the reviewer for the comments. Samples in **Fig. 7d** are all in the proteomics cohort. P16 exhibited positive expression in all HPV+ cervix SCCs and vagina SCCs, and we only selected one representative case to present in **Fig. 7d**. We also modified the sentence to make it clear.

Q26.

Line 481-482: It would be helpful if there is a test for agreement with the proteomics data and/or the highly expressed in thymus SCC.

Response: Thank the reviewer for the comments. We have added a correlation analysis (Spearman correlation, $p = 0.015$; **Figure RL 11**) for the immunohistochemistry data and proteomics data.

Figure RL 11 Significant Spearman correlation between the immunohistochemistry data and proteomics data ($p = 0.015$) of CPXM2 in thymus SCC.

Q27.

Figure 7 (caption): Can you clarify whether or not the same tissue was assessed for H&E, P63, EBER, P16, etc. or representative tissues from each organ type depending on the stain?

Response: Thank the reviewer for the comments. It is the same tissue was assessed in rows for H&E, P63, EBER, P16 and three candidates, and we revised the result (line: 459) to make it clear.

Methods:

Q28.

Clinical Sample Acquisition (582). Patients were excluded if they had advanced disease (perhaps this could include clinical stage for clarity). Also, "any condition that may influence the outcome

evaluation". It would be helpful to indicate that some patients are missing outcomes. Presumably these were dropped from survival analysis, but may have been censored at time 0. It would be helpful to clarify how this was addressed.

Response: Thank the reviewer for the comments. We apologize for not explaining it clearly and have revised the sample collection part in the updated manuscript. All patients were excluded if they had 'other' advanced diseases.

We screened documented SCC patients of 17 organs. For 5 SCCs with high incidence, including throat, nasopharynx, esophagus, lung, and cervix, we screened patients who underwent surgery at Zhongshan Hospital, Fudan University in 2015. All these patients were with complete clinical information and follow-up. However, for the other 12 SCCs with a lower incidence, we screened patients from the 2019 to 2001 flashback. Unfortunately, 68 patients lost follow-up and not included in survival analysis. As this work pays more attention to the differences and similarities of the morphology on 17 SCCs, we included these patients. **Supplementary Table 1a** shows the surgery year distribution of all 333 SCC patients.

Q29.

Additionally, samples were collected from 2001 to present. The manuscript states "all patients provided written consent". This suggests that a general banking protocol was used to collect tissues, distinct from this specific study. It would be helpful to clarify this. In the case of the adenocarcinomas, it is explicitly stated that patients were consented prior to surgery.

Response: Thank the reviewer for bringing this to our attention. A general banking protocol was used to collect tissues and patients were consented prior to surgery. We have moved this statement to the last to make the SCC and AC cohort consistent.

Q30.

Patients with adenocarcinomas were excluded if radiation was used (unclear if this was preoperative). Presumably this was not the case in the squamous cohort, was this evaluated?

Response: Thank the reviewer for bringing this to our attention. The radiation mentioned in the adenocarcinoma part is preoperative. The pan-SCC cohort was also excluded if preoperative radiation was used. We have modified this in the revised manuscript.

Q31.

Histological evaluation (594): Were ADC using the same pathology assessment/acceptance criteria (>80% viable tumor nuclei, >50% cellularity)?

Response: Thank the reviewer for the comments. We used the same strict pathological assessment criteria in AC as the SCC. In the revised manuscript, we have added this criterion.

Q32.

Database Searching (640): What version of Uniprot was used? It may be present in a figure, but not stated in the methods. Also, were missed cleavages considered for trypsin identification? Modifications? More details here would be needed to understand how the searching was done.

Response: Thank the reviewer for the comments and apologize for our negligence. The Uniprot library was downloaded on 2019_07. SwissProt was chosen, including 20,431 human entries. We have modified this in the revised methods. It is true that missed cleavages were considered for trypsin identification, we selected 2 for the missed cleavage. Carbamidomethylation was selected as fixed modification and oxidation was selected as variable modification.

MAD-based protein selection:

Q33.

line 759-767: What does "the 20% bottom mad proteins of each organ were combined and duplicates removed" mean? It would seem reasonable to remove the bottom 20% of mad proteins due to lack of variability, but it's not clear what the term "combined" means here (averaged?).

Response: Thank the reviewer for the comments. In this part, we explored the similarities and differences between 17 SCCs.

- Firstly, we'd like to explain why we used "the 20% bottom mad proteins of each organ". To explore the proteomic clustering of 17 SCCs, we screened proteins with consistent and ubiquitous expression in each SCC, meeting the following two criteria. 1, the proteins expressed in at least 1/3 of the samples in specific SCC type; 2, the proteins were sorted according to MAD value. The 20% proteins with the lowest MAD values were selected as the proteins with consistent and ubiquitous expression in each certain SCC. In other words, we think these proteins could represent the molecular features of one certain SCC.
- Next, we "combined" the 17 protein sets, removed the duplicate proteins, and obtained a

protein set containing 10,259 proteins. So, "combined" here means that we get a union set of 17 protein sets. In the following steps, we calculated the average expression value of each protein in each SCC, and 1,500 proteins with top MAD values were used to do the hierarchical clustering.

Q34.

line 785-794.: Why would the 1% bottom mad proteins be selected? These are the least variable (assuming smallest mad values are meant). It is not clear why the 1220 proteins were used to compute an average expression value (for what purpose). In particular, ordering by mad does not impact the random forest. Perhaps something different was meant, but it was not clear from the description. In general, it would also be good to indicate if all filtering and protein selection was done only on the training set (independent evaluation) or on both the training and testing sets.

Response: Thank the reviewer for the comments, and we apologize for the unclear description. For this part, we'd like to establish a random forest to distinguish these 17 SCCs.

- **Firstly**, the most consistently expressed proteins were selected within each organ as 1% bottom mad proteins identified in at least 75% cases in **each organ**. A total of 17 protein lists were then combined, and one protein list containing 1,220 proteins after removing repetitive proteins was obtained (**a total of 1,220 proteins were obtained from 333 cases**).
- **Secondly**, all 333 SCC cases were randomly divided into a training set and a validation set, containing 75% and 25% cases, respectively.
- **Thirdly**, the RandomForest function was used on the training set to calculate the importance (Indicated by "Mean Decrease Accuracy") of each protein (**top importance was obtained only from training set**). 10-fold cross-validation was used to select the suitable number of top important proteins.

As you mentioned, ordering by mad does not impact the random forest and we have moved this filtering.

Q35.

line 721: "multiple testing correction using ..."

Response: We appreciate your correction and, we have modified this in the revised manuscript.

Q36.

Line 811/812: Please define HPV16 as the "main type".

Response: Thank the reviewer for the suggestion, and this is our negligence. The main type was defined using the minimum cycling threshold in the PCR process when multiple infections happened. We have annotated the main type in the revised manuscript.

Discussion comments:

Q37.

line 511: "involved in rare SCC initiation". It would be more accurate to describe them as regulators involved in rare SCC since it is a descriptive finding.

Response: Thank the reviewer for the recommendation. We agree that the 'initiation' is appropriate here, and we have deleted the word.

Reviewer #2 (Remarks to the Author): Expert in SCC proteomics

In the manuscript entitled “Proteomic Landscape of Pan-squamous Cell Carcinomas”, Song et al. provided an interesting panorama of SCC proteomes from 333 patients and 17 sites using mass spectrometry label-free quantitation. By performing several group comparisons (SCC vs AC; common SCC vs rare SCC; HPV+ anogenital SCC vs HPV- anogenital SCC), the authors indicated signatures of differentially abundant proteins that have a prognostic significance and are able to modulate specific pathways. Four groups of SCC types were defined by clustering tumor proteomes and six subtypes of SCCs were determined based on immune enrichment of cell composition. Molecules from these groups may potentially modify biological pathways and are promising as druggable targets. Finally, a machine learning model was built to predict the tumor site of origin based on SCC proteomes and 3 proteins were selected for validation using IHC.

Overall, that is a detailed study that provided insights into the biology of a common cancer type (SCCs) by profiling the tumor proteomes followed by an extensive bioinformatic analysis. The significant number of patients evaluated is certainly a highlight of the study. Since large-scale analysis of tumors available in the literature have focused on DNA/RNA levels, describing the proteome and the potential functional implication is definitely novel and opens up new perspectives for the study of cancer. The description of a protein signature that can possibly define the origin of SCCs is also remarkable and revealed targets with potential to translate to the clinics. For instance, the authors provided a valuable resource to the scientific communities for further exploration. Considering the methodology, a noteworthy point is that the manuscript presents a deep characterization of SCCs achieved by implementing a strong and robust bioinformatic analysis. The proteomic results indicate a successful workflow of sample preparation, MS run and data analysis (~ 15,000 protein groups quantified for SCCs and ~10,000 for ACs), even though some additional information should be provided to assure the reproducibility and quality of the data.

In summary, the study is of major interest for the oncology and proteomics fields and is in-line with most articles published in Nature Communications. There are therefore some shortcomings that should be addressed.

Response: We appreciate the reviewers for the positive review and valuable comments. We have revised the manuscript according to the comments. The point-to-point responses were as follows.

Results

Q1.

Line 109: The paper relies on proteomics data from a large group of SCC samples and the quality control of MS runs is certainly one of the main concerns to assure the reliability of bioinformatic analysis. However, I am not sure about QC analysis. How was the set of repeated samples selected for correlation (SFig. 2b)? Additionally, the criteria for QC of individual samples were not described in the text and it would be good to see some QC results for all MS runs. Plotting Spearman's correlation, counting valid values per sample or, if suitable, describing retention times or m/z for trypsin autolysis peaks across samples are suggested approaches.

Response: Thank the reviewer for the comments, and we apologize for not describing the quality control clear.

- To confirm the stability of our LC-MS/MS platform, we tested the commercial 293T cell as quality control, showing a high correlation (average, 0.9) and thus demonstrating the consistent stability of the platform. Furthermore, the repetitive analysis of 7 randomly selected replicate samples (2 bladder, 2 gallbladder, 2 breast, 2 thymus, 2 pancreatic, 2 penis, and 2 perineum SCCs) was performed in the middle and at the end of the project, showing good reproducibility, with a high level of correlation (median, 0.92; range, 0.86-0.97; **Supplementary Fig. 2b**).
- We plotted the Spearman correlation coefficients for all 333 MS runs as you suggested (**Figure RL 12a**). The median correlation coefficient among these samples was 0.74, and the maximum and minimum values were 0.99 and 0.56, respectively.
- Identified protein number per sample was shown in **Figure RL 12b** by organ. On average, the SCC proteome had 8,120 protein groups per sample, ranging from a minimum of 6,261 in the thyroid to a maximum of 9,296 in the thymus and 5,648 proteins were present in all 17 SCCs.
- You commented, "if suitable, describing retention times or m/z for trypsin autolysis peaks across samples are suggested approaches." Sorry that the trypsin autolysis peaks were not searched in this work, and we chose the conserved peptides instead. Firstly, we chose to describe the retention times (RTs) of consistently identified peptides for stably expressed

proteins, including TP63, EGFR, KRT5, Actin, ELMO2. A total of 65 peptides were identified in all 17 SCCs (not 333 cases), belonging to EGFR, ELMO2, KRT5, and TP63. Secondly, a scatterplot showed the identified peptide frequency and coefficient of variation (CV) of peptide abundance (**Figure RL 12c: left**). Peptides with a frequency $>97\%$ and $CV < 0.7$ were chosen for display in line chart (**Figure RL 12c: right**). As the database search results of Peaks online were only provided RT mean of each SCC, the RT mean of 17 SCCs for four peptides was shown in **Figure RL 12c (right)**.

From these results, we think the works' MS data is of high quality.

Figure RL 12 Quality assessments for MS data. **a** Spearman's correlation coefficients for all 333 MS runs. The median correlation coefficient among these samples was 0.74, and the maximum and minimum values were 0.99 and 0.56, respectively. **b** Number of proteins quantified in each SCC patient. **c** A scatterplot showed the identified peptide frequency and coefficient of variation (CV) of peptide abundance (left), and a line chart showed RT mean of 17 SCCs for four peptides (right).

Q2.

Normalized protein intensities from pan-SCC and pan-AC were log₂ transformed. Even that is not always true, log transformation reduces the skewness of large-scale data and make it more closely to a normal distribution. If that is the case, it would be appropriate to use parametric tests for group comparison instead of the non-parametric analysis described in the manuscript (Wilcoxon rank sum test, Kruskal-Wallis test). Did the authors test the normality assumption of the data? Please comment.

Response: Thank the reviewer for the comments.

Figure RL 13 shows the distribution of log₂ transformed data (left: Pan-SCC cohort, right: Pan-AC cohort). Kolmogorov-Smirnov test was performed to examine the normality of pan-SCC and pan-AC cohorts. As a result, the pan-SCC and pan-AC data are not normal distribution ($p < 0.05$ for all data distribution). In this case, we think the non-parametric analysis is appropriate in this work. Sorry for not showing the data distribution at the beginning.

Figure RL 13 Data distribution of log₂ transformed protein intensities (left: Pan-SCC cohort, right: Pan-AC cohort).

Q3.

SFig. 3: Not clear what is represented on this image.

Response: Thank the reviewer for the comments. Cancer owns its specific proteomics, such as overexpressing of sustaining proliferative signaling, evading growth suppressors, resisting cell death, inducing angiogenesis, and other new critical characteristics (PMID: 10647931, 21376230, 35022204). Thus, cancer samples will lose the expression of original tissue markers as reported (PMID: 32649877). Therefore, to prove the high tumor purity of this pan-SCC cohort, we downloaded normal tissue signature protein (tissue enriched proteins) lists from Human Protein Atlas (<https://www.proteinatlas.org/humanproteome/tissue/tissue+specific>) and presented the

proteins of specific organs that had lost expression in corresponding SCCs in **supplementary Fig. 3**. For instance, SPRR1A, a cross-linked envelope protein of keratinocytes, is an esophageal signature protein (**Figure RL 14a**). It was reported that it had lost expression in esophageal SCC compared to the matched normal tissue samples (**Figure RL 14b**). As is shown in **Figure RL 14c**, SPRR1A had lost expression in esophageal SCC. In this case, this proved the tumor purity of the pan-SCC cohort to some degree. We have modified the result to make this clearer.

Figure RL 14 SPRR1A expression in normal tissues, esophageal cancer vs matched normal tissues, and in pan-SCC cohort. **a** SPRR1A expression in six normal tissues, including esophagus, liver, kidney, testis, cerebral cortex, and lymph node. **b** RT-PCR analysis showed the differential expression of SPRR1A and other proteins in eight pairs of esophageal cancer tissue samples and the matched normal tissue samples. (PMID: 14647409 DOI: 10.1038/sj.onc.1207218.) **c** SPRR1A expression in pan-SCC cohort.

Q4.

Fig. 2d; SFig. 7: Not sure what NES means.

Response: Thank the reviewer for the comments, and this is our negligence. NES stands for normalized enrichment score, and we have added this explanation in corresponding figure legends.

Q5.

Lines 179 and 325: Besides using their own proteomics data for Cox analysis of DEPs between SCCs and ACs, 9 RNA databases from cancer tissues were used to assume the role of proteins in prognosis. Did the authors evaluate if transcripts are somehow correlated with proteomics data? Otherwise, the assumptions may not be true.

Response: Thank the reviewer for this critical comment. Due to the availability of transcriptomics, we validated the proteins using RNA datasets. We did not evaluate the correlation between transcripts and proteomics, as limited corresponding SCC proteomics data was available. Some studies showed positive and significant correlations with the proteomics and mRNA transcripts (PMID: 31585088, 30962452, 31395880). **Figure RL 15** is belonged to a work focusing on proteogenomic characterization of HBV-related Hepatocellular carcinoma (PMID: 31585088). As is shown in **Figure RL 15**, mRNA and protein were positively correlated for most (98.6%) mRNA-protein pairs across the 159 samples, and 90.3% showed significant positive correlation (multiple-test adjusted $p < 0.01$) with a median Spearman's correlation coefficient of 0.54 in 6,203 mRNA-protein pairs.

In addition to the positive correlation between expression of proteins and transcripts, previously published proteomic researches also explored the prognostic value of proteins using survival data of the transcriptome databases, including The Cancer Genome Atlas (TCGA) and Queensland Centre for Medical Genomics (QCMG). One representative study (PMID: 31484774) showed poor prognostic proteins participating in ECM process, such as S100A6 and FN1, which were also captured by our analysis.

In the future studies, we will validate these proteins in SCC proteomics datasets. We talked about the limitation in the revised discussion part.

Redacted

Figure RL 15 The overall correlation between mRNA and protein data (PMID: 31585088).

Q6.

Fig. 5a: The authors should amend Subtype 2 name to FaSq.

Response: Thank the reviewer for bringing this to our attention. In the revised **Fig. 5**, we have modified this mistake.

Q7.

Fig. 5d: Not mentioned in the main text.

Response: Thank the reviewer for the comments. We annotated **Fig. 5d** in line 326-327 (revised manuscript), but no explanation was provided in the submitted manuscript. In the revised version, we further explain **Fig. 5d**.

Q8.

Fig. 6b: How can the authors explain the high levels of protein pRB in HPV-infected tumors? Since the silencing of pRb by E7 viral protein produces a rise in p16, the abundance of pRB is not in agreement with what is reported in the literature or in this study. Also, SOX2 is not frequently associated with HPV infection and it would be appropriate to describe the rationale for including this protein in the analysis.

Response: Thank the reviewer for the comments. To answer your question, we firstly compared the RB expression in anogenital vs non-anogenital SCCs. As shown in **Figure RL 16a**, RB expression is higher in non-anogenital SCCs than anogenital SCCs (Wilcoxon-rank sum test, $p < 0.001$), which is consistent with previous studies. Moreover, RB showed a significant expression difference among all five anogenital SCCs (Kruskal-Wallis test, $p < 0.001$) and cervical SCC showed the lowest expression (**Figure RL 16b**). Spearman correlations between CDKN2A (p16) and RB expressions were then calculated in all anogenital SCCs individually. Interestingly, a negative correlation trend was found in cervical SCC (Spearman correlation; $R = -0.255$, $p = 0.265$; **Figure RL 16c**). So, this cervical SCC data is consistent with previous work.

TP63 and SOX2 were considered as associated with squamous differentiation. Indeed, SOX2 is not frequently associated with HPV infection. We have removed SOX2 from **Fig.6b**. Thank the reviewer again for this meticulous suggestion.

Figure RL 16 The protein expression and correlation of RB and CDKN2A in SCCs. **a** RB expression in anogenital SCCs versus non-anogenital SCCs, Wilcoxon-rank sum test. **b** RB expression in five anogenital SCCs, Kruskal-Wallis test. **c** Scatterplots showed the correlation between RB (x axis) and CDKN2A (y axis) in cervical SCCs. Spearman correlation.

Q9.

Fig. 6e: The error bars are too large and it is difficult to believe that there is a real difference between HPV+ and HPV- cases for these proteins, even with an adjusted p value. Maybe including the protein abundances for the non-anogenital HPV-negative SCC types would make the hypothesis of Fig.6f stronger.

Response: Thank the reviewer for the comments. We added the non-anogenital SCCs in the comparison as you suggested (**Figure RL 17a** and **17b**).

There was no significant difference between Group 1-3 (HPV16 infected cases) and non-anogenital SCC of EZR (Wilcoxon-rank sum test, $p = 0.0937$), PAWR (Wilcoxon-rank sum test, $p = 0.4682$), and DUSP3 (Wilcoxon-rank sum test, $p = 0.1425$) belonging to negative regulation of T cell receptor signaling pathway (**Figure RL 17a**). As the immune response is ubiquitous in the tissue microenvironment, maybe it is the reason no significant difference between Group 1-3 and non-anogenital SCC.

Inositol phosphate catabolic process proteins showed differential expression patterns compared to non-anogenital SCCs (**Figure RL 17b**). Group 1-3 of INPP1 (Wilcoxon-rank sum test, $p < 0.0001$) and IMPA2 (Wilcoxon-rank sum test, $p < 0.0001$) showed a high-level expression, compared with

non-anogenital SCCs. Group 1-3 of NUDT3 showed a lower expression than non-anogenital SCCs (Wilcoxon-rank sum test, $p < 0.0001$), as NUDT3 mediates phosphate degradation (PMID: 34788624).

Interestingly, Inositol phosphates were reported promoting HIV-1 assembly and maturation to facilitate viral spread in human CD4+ T cells (PMID: 33476323). Multiple isomers of inositol phosphate were found in Epstein-Barr-virus-transformed (T5-1) B-lymphocytes and may be related with cell transformation or proliferation (PMID: 1660712). So, we think the Inositol phosphate catabolic process participates in HPV related tumorigenesis. However, due to the small sample size, a large-scale study will be needed to explore this further.

Figure RL 17 Boxplots showed protein expression of molecules in (a) Negative regulation of T cell receptor signaling pathway (EZR, PAWR, and DUSP3) and (b) Inositol phosphate catabolic process (INPP1, IMPA2, and NUDT3). Wilcoxon-rank sum test.

Q10.

Line 459: I didn't get the point of why the authors evaluated the 3 markers by IHC. If a signature of 19 proteins was accurately able to discriminate SCC tumor based on their origin (but the way, this information is not stated in the main text), why the 3 proteins were selected for validation? I understand the validation phase is an important step in large-scale experiments, but analyzing the 3 proteins alone does not make sense in the context of the classifier and did not make the proteomic data stronger.

Response: Thank the reviewer for the comments. We ordered all 19 antibodies to validate the proteomic data, and we successfully bought 16 antibodies. However, 13 antibodies were not getting good staining due to the poor antibody specificity. In this case, we only presented three markers in the study. Now we are ordering clinical grade antibodies and we will validate the other markers in future research when antibodies are available. An explanation had been added in the discussion part.

Discussion

Q11.

It is not necessary to extensively re-state the key findings in the discussion.

Response: Thank the reviewer for the valuable comments, and we appreciate it. In the revised manuscript, we rewrite the discussion part. On the one hand, reviewers' comments about the discussion part were considered. On the other hand, we discussed the key findings associated with published works instead of just restating these findings.

Methods

Q12.

Line 580: References should be provided for WHO classification and TNM system.

Response: Thank the reviewer for the suggestion. We have found corresponding chapters in the WHO classification and traced the original references. References for SCC classification were added in the revised manuscript (line 597-600, Page 28). TNM systems were referred to the AJCC cancer staging system 8th edition.

Q13.

Fundamental information is missing in the proteomic workflow and it is difficult to judge the reproducibility and quality of the methods employed. Please provide additional information.

Response: Thank the reviewer for the recommendation. We have revised the materials and methods part according to your suggestions. Detailed responses, please see below.

Sample preparation

a. Why did the authors add an acetone precipitation step in the FASP protocol? FASP did not take care of contaminant removal? The authors should provide the appropriate references.

Response: Thank the reviewer for the comments. As you mentioned, the FASP did not take care of the contaminant removal very well. In this case, we developed a novel method for FFPE proteomic sample preparation. To decrosslinking and lysing the FFPE samples, we applied a high concentration of detergent (4% SDS) and reducing agent (1mM DTT), which exceeded the normal range in FASP protocol (the number of identified proteins from FFPE samples only using FASP protocol were significantly lower than fresh tissue). Therefore, we added an acetone precipitation step to purify the proteins from the decrosslinking-lysis buffer. Moreover, we used the FASP protocol after the acetone precipitation to dissolve the protein pellets efficiently. The results demonstrated that our protocol was highly repeatable for FFPE proteome profiling with in-depth coverage.

b. How was alkylation and quenching performed?

Response: Thank the reviewer for the comments. We apologize for not describing the alkylation and quenching progress clearly. In our protocol, the homogenized samples were boiled in lysis buffer for decrosslinking and lysis. Acetone precipitation was applied to purify the proteins from the decrosslinking-lysis buffer. Then 8M Urea buffer was used to dissolve the protein pellets. The supernatant was loaded onto an ultrafiltration filter column (10 kD, 500 µl) and centrifuged at 12000 rpm for 15 min. The reduction and alkylation progresses were then carried out, 100 µl of reduction buffer (10 mM DTT, 25 mM NH₄HCO₃) were loaded on ultrafiltration filter column, incubated for 1 hour at 56°C, and centrifuged at 12000 rpm for 15 min. Then 100 µl of alkylation buffer (55 mM IAA, 25 mM NH₄HCO₃) were loaded on ultrafiltration filter column, incubated for 45 min in dark at room temperature. The filter was then washed three times by adding 100 µl ammonium bicarbonate (ABC, 50 mM) to the column, followed by centrifugation. The proteins were digested by trypsin. The resulting peptides were loaded on LC-MS.

c. The authors declare that a trypsin-to-protein ratio of 1:50 was used for digestion, but there is no information on protein quantification. How were protein levels determined? The authors also state that “Target on-column load was 200ng total peptide per injection”, indicating that peptides were also quantified. Please clarify.

Response: Thanks for the comments. Acetone precipitation was applied to purify the proteins, and 8M Urea buffer was used to dissolve the protein pellets. We measured the protein concentration of the solution using spectrophotometer (NanoDrop, Thermo, USA). The amounts of protein were adjusted to 400 µg. We also measured the peptide concentration of samples before loading to the LC-MS. A total of 200ng peptides was loaded per injection.

d. It is appropriate to present specifications of the trypsin used for digestion.

Response: Thank the reviewer and the trypsin used in this study is Sequencing Grade Modified Trypsin (Promega V5111).

e. What is MS water? How were peptides acidified to stop digestion?

Response: Sorry for this inaccurate description. MS water is referring to LC-MS grade water. The peptide was acidified in 0.1%FA solution buffer.

LC-MS/MS

f. Some fundamental aspects of the MS runs are missing, like m/z range, mode of data acquisition (DDA? DIA?), resolution, etc.

Response: Thank the reviewer for the comments.

- The timsTOF Pro was operated in PASEF mode (**PMID: 30385480**).
- The resolution parameter was set to 50,000 for MS1 and MS2.
- Mass spectra for MS1 and MS2 scans were recorded between 100 and 1700 m/z.
- Ion mobility resolution was set to 0.60–1.60 V·s/cm over a ramp time of 100 ms.
- Data-dependent acquisition (DDA) was performed using 10 PASEF MS/MS scans per cycle with a near 100% duty cycle.
- An active exclusion time of 0.4 min was applied to precursors that reached 20,000 intensity units.

g. Although the identifier is provided in the text, the repository where raw files are deposited is not informed and data could not be accessed.

Response: Sorry for this inconvenience and that is our negligence. In the revised version, we have added the assess link and password.

The accession number for the MS proteomics data reported in this paper is IPX0002831000 (<https://www.iprox.cn/page/PSV023.html?url=164468948261031Uy>, password: xAgI).

MS search

h. Details for the Uniprot library should be provided, including download date, number of residues considered. Were SwissProt and TrEMBL entries considered?

Response: Thank the reviewer for letting us know. We have added the detailed information of Uniprot library. The Uniprot library was downloaded on 2019_07. SwissProt was chosen, including 20,431 human entries.

i. A fragment ion tolerance of 0.05Da was used. Why did the authors use such a restrictive cut-off? I am afraid that important protein identification was lost.

Response: Thank the reviewer for the comments. The resolution of timsTOF pro is 50,000. The average MS2 is smaller than 3 ppm after evaluation. As Peak Online MS2 can only use 'Da' as unit, we set 0.05 Da as the fragment ion tolerance. As is reported in the work "Proteogenomic Analysis of Human Colon Cancer Reveals New Therapeutic Opportunities" in 2019 (**PMID: 31031003**), the product ion tolerance for MS/MS was 0.05 Da. Therefore, we think 0.05Da is appropriate.

j. How did the authors handle contaminants?

Response: Thank the reviewer for this comment.

- a) For FFPE tissue preparation, tumor samples were dissected and collected, and adjacent non-tumor sections were discarded. Therefore, our protocol could avoid sample contamination in the first step.
- b) For data acquisition, 200ng total peptide per injection was loaded, and a blank wash run followed each sample run to ensure no cross contamination.
- c) For MS search, we regularly searched MS runs for contaminants, which is mainly containing keratin (Human, Mouse, Bovin), and lab-derived contaminants (BSA, trypsin, etc.), as the mass spectrometry maintenance routine. Few SCC samples were also tested in routine work, and only human keratin and trypsin were detected. Our project includes human samples and the SCC is rich in keratin, so we did not test all samples.

k. I could not find any information about variable and fixed modifications, or the retention time

window considered.

Response: Sorry for this inconvenience and thank the reviewer for the comments.

- Fixed Modifications: Carbamidomethylation
- Variable Modifications: Oxidation (M)
- Retention Time Shift Tolerance (min): 4.0
- Retention Time Range: $0.0000 \leq \text{Retention Time} \leq 10000.0000$

Data analysis

1. Replacing missing values engenders intense debate in the scientific community. Are there any reasons why missing values were replaced? Maybe keep the original data would be appropriate and statistics would take care. Have other approaches been tested for data imputation to assure that replacing by one-tenth of the minimum intensity is the most suitable strategy?

Response: Thank the reviewer for the comments. We first applied the ID transfer (also known as match between runs, MBR) algorithm (**PMID: 24942700**) for the missing values in this study. A dynamic regression function based on commonly identified peptides in samples was built. According to correlation value, the function chooses linear or quadratic function for regression to calculate retention time (RT) of corresponding hidden peptides and check the existence of the extracted ion chromatogram (XIC) based on the m/z and calculated RT. The function evaluated the peak area values of those existing XICs. These peak area values are considered as parts of corresponding proteins. ID transfer has been proved to be an effective technique to fill the missing values, which was widely used in other proteomic studies (**PMID: 31495571**). With ID transfer, the missing values could be significantly reduced. As for the rest missing values after applying for ID transfer, to avoid artificially increasing the false discovery rate, we did not apply other algorithms but a lowest of detection (LOD) strategy for data imputation. We replaced missing values with a certain small number (1/10 of the minimum) to ensure the accuracy of subsequent analysis results. The detailed procedure is as follows. Firstly, we deleted the proteins which were not detected in 2/3 of the samples in each certain SCC. Then, we use the single value (1/10 of the minimum) to replace these missing values. This strategy has been proved to have a robust

performance in proteomic data and applied in the previously published studies, such as the proteomic landscape of diffuse-type gastric cancer project (**PMID: 29520031**) and the early-stage hepatocellular carcinoma project (**PMID: 30814741**).

Q14.

I was wondering whether the separation of SCCs and ACs in PCA before batch effect correction just reflects their distinct biological characteristics. How can the authors be sure that the separation was a batch effect?

Response: Thank the reviewer for the comments. To confirm the existence of the batch effect, we firstly compared the similarity between the commercial 293T cells of the pan-SCC and pan-AC cohort. As shown in **Supplementary Fig. 4a**, the 293T samples clustered together within pan-SCC or pan-AC cohort but separated from each other between pan-SCC and pan-AC. From here, we think an actual batch effect existed. Then, we remove the batch effect between pan-SCC and pan-AC using the same method as the 293 samples (**Supplementary Fig. 4b**). In this case, we believe the analyses will reflect the differences between SCCs and ACs more accurately.

Q15.

Not clear if IHC was performed in the same cohort as proteomics.

Response: Thank the reviewer for bringing this to our attention. Yes, the IHC was performed using the cases in the proteomics, as we constructed a tissue microarray using the same cohort. We have clarified that the tissue microarrays were constructed using the pan-SCC cohort.

Q16.

For HPV grouping, how the authors defined if HPV16 is the main type (group 2) or not (group 3) in multiple infections?

Response: Thank the reviewer for the comments. As the HPV16 infection was tested by RT-PCR in this study, we defined the main type as the one using the minimum cycling threshold. We have revised and added the explanation in line 871.

Reviewer #3 (Remarks to the Author): Expert in SCCs

Understanding the molecular pathways driving histologically similar cancers across anatomic sites may provide new treatment paradigms that historically have been site-specific. Differences in mutational patterns and gene expression profiles between squamous cell carcinomas (SCC) and adenocarcinomas (AC) arising across anatomic sites have been well described using common resources such as TCGA. However, the proteomics landscape of SCC across anatomic sites has not been previously investigated in large numbers of tumors. The current manuscript describes proteomic patterns in 333 treatment-naïve squamous cell carcinoma (SCC) tissues obtained from 17 organ sites and 69 treatment-naïve adenocarcinoma (AC) tissues obtained from 7 organ sites from a single university-based hospital in Shanghai, China. Proteomic characterization of tissues was conducted using a mass spectrometry-based approach validated using tissue microarray (TMA) immunohistochemistry (IHC).

Major findings include the elucidation of pathways differentiating SCC from AC (keratinization, glucose metabolism and extracellular matrix), molecules within those pathways associated with disease prognosis, and proteomic clusters/immune subtypes that may represent potential druggable targets. The resulting data repository will serve as a unique and valuable shared resource for investigators to use in the future.

Methods and results are described in great detail, yet there are some key points that should be clarified and/or expanded upon.

Response: We appreciate the reviewers for the positive review and valuable comments. We have revised the manuscript according to the comments. The point-to-point responses were as follows.

Case selection and classification:

Q1.

The methods state that the cases were randomly selected. How was this accomplished, and what percentage of the total SCC cases treated in the 18-year range do the cases included in the current study represent? Exclusion criteria are presented, yet it's unclear how many patients were excluded for the reasons listed.

Response: Thank the reviewer for pointing out this question, and we apologize for not explaining it clearly. We screened 595 documented SCC patients for 17 organs (**Figure RL 18**). For five common SCCs with high incidence, including throat, nasopharynx, esophagus, lung, and cervix, we screened patients who underwent surgery at Zhongshan Hospital, Fudan University in 2015

(40 cases in the beginning). All these patients were with complete clinical information and follow-up. However, for the other 12 SCCs with a lower incidence, we screened patients from the 2019 to 2001 flashback (395 cases in the beginning).

Among the 262 excluded patients, 86 patients were with no/complete standard clinical information (not including TNM stage information etc. for 12 SCCs with a lower incidence), 74 patients with other malignancies, 40 cases with neoadjuvant treatment, and 62 patients failed to pass pathological evaluation. Supplementary Table 1a shows the surgery year distribution of all 333 SCC patients.

We also added the patient selection flow chart in **Supplementary Fig. 1 (Figure RL 18)**, and described the detailed information in the methods part as the same time.

Figure RL 18 The quality control and sample filtering standards of sample collection in this cohort. Q2.

While the overall number of tumors (n=333) is substantial, the numbers of samples available per anatomic site ranged from 10-22 for SCC and 8-12 for AC. Therefore, inferences drawn for

specific sites are limited by small sample size. This point should be added to the discussion.

Response: Thank the reviewer for bringing this to our attention and we really appreciate it. This is an important point that should be discussed. The small sample size may limit our findings for specific tumor types, and large-scale studies are needed to validate these findings further. In the revised manuscript, we have added this comment in discussion part.

Q3.

The classification of rare versus common tumors is unclear. The authors state that the WHO Classification of Tumors was used, yet there is no reference, and some cancers seem to be misclassified. For example, SCC of the vagina is very rare (i.e. incidence is less than 6 per 100,000), yet it is included here as a common cancer.

Response: Thank the reviewer for the comments, and sorry for this inconvenience.

- The classification of common or rare SCCs in this study was depended on the originated tissue, whether it is squamous epithelium or not.
- Vaginal cancer is a rare gynecologic cancer; however, it was revealed that majority of vaginal cancers reported are SCCs (4th WHO Classification of tumors of female reproductive organs, **Figure RL 19**). We have made an explanation in our revised manuscript for common and rare SCCs. Also, the references for common or rare SCC classification were inserted in the manuscript.

Redacted

Figure RL 19 Vaginal SCC epidemiology in 4th WHO Classification of tumors of female reproductive organs (P211).

Q4.

As the authors point out in the Background, metastatic SCC's (or primaries with elevated metastatic potential) are an important clinical challenge. However, only primary SCC's were included in this case series, and no information was provided on whether or not patients developed metastases during follow-up. This is an important design limitation that should be discussed.

Response: Thank the reviewer for the comments. We pointed out the difficulty in metastatic SCC diagnosis in the background as you mentioned. However, we didn't include metastatic SCCs in this work. The main reason is that we intend to compare the proteome of primary SCCs, and then apply the markers with differentially diagnostic values for metastatic SCCs in the future. We are following up with these patients to acquire the metastases information, and new metastatic SCC cohort is under collecting. In future studies, we'd like to include metastatic SCCs to further validate our findings. In this study, we only compared the characteristics of primary SCCs. This limitation was also discussed in the revised manuscript.

Patient follow-up and survival analysis:

Q5.

No information is provided on the average length of follow-up (and range), as well as whether or not patients were lost to follow-up, and if so, how they were handled in the analysis. It is also not clear that the proportional hazards assumption was assessed.

Response: Thank the reviewer for the comments.

The average length of follow-up is 32 months (3-160 months). For SCCs with a low incidence, 68 patients lost follow-up. These patients were also included in this work but not included in survival analysis.

The multivariate COX proportional hazard model was assessed according to your suggestions. **Table RL 8** show the multivariate COX proportional hazards model, including SCC origin, gender, age, stage, and histology. Upon multivariate analysis, both OS and DFS were associated with age (OS, $p < 0.0001$; DFS, $p < 0.0001$) and stage (OS, $p = 0.0083$; DFS, $p = 0.01$). SCC origin is not significant concerning to OS and DFS.

Table RL 8 Association between clinicopathological characteristics and OS/DFS by multivariate COX proportional hazards model

	OS			DFS			
	HR	95%CI	p value	Factor	HR	95%CI	p value
SCC origin	1	(95%CI:0.95-1.1)	0.67	SCC origin	1	(95%CI:0.95-1.1)	0.82
Gender	0.82	(95%CI:0.46-1.5)	0.5	Gender	0.81	(95%CI:0.45-1.5)	0.48
Age	1.1	(95%CI:1-1.1)	6.00E-05	Age	1.1	(95%CI:1-1.1)	4.40E-05
Stage	1.5	(95%CI:1.1-1.9)	0.0083	Stage	1.5	(95%CI:1.1-1.9)	0.01
Histology	1.7	(95%CI:0.94-3.1)	0.077	Histology	1.6	(95%CI:0.89-3)	0.11

Q6.

Why were patient age and gender (as well as other patient characteristics such as stage at diagnosis) not considered as potential covariates in the multivariable modeling, along with the three covariates stated (protein expression, organ and histology)?

Response: Thank the reviewer for the critical comments. We agree that factors such as gender, age, and tumor stage may affect patients' prognosis a lot. In the revised manuscript, we reanalyzed all survival analyses. For the pan-SCC cohort, we included age, gender, stage, histology, organ, and protein expression as covariates in the multivariate Cox proportional hazards model. This cohort has its complexity, as it consists of 333 patients originating from 17 organs and only ~20 cases per organ. After this strict calculation, we determined molecules with prognostic statistical significance, including RPL12 (Ribosome; (95% CI: 0.45-0.82); $p = 0.036$, BH adjusted), ATM (Cell cycle; (95% CI: 0.41-0.84); $p = 0.049$, BH adjusted) with good prognostic value, and SERPINE1 (P53 downstream pathway; (95% CI: 3.6-210); $p = 0.0135$, BH adjusted) and MMP19 (Extracellular matrix; (95% CI:1.1-1.2); $p = 0.0178$, BH adjusted) with poor prognostic value. Please see **Supplementary Fig.6** and **Fig.4** in the revised manuscript. In future studies, we plan to collect more SCC patients to validate these findings.

Results:

Q7.

In general, it was difficult to follow the results section given the sheer number of figures, figure panels and supplementary materials. The figures were very detailed, as were the supplementary

data (often patient-level data files). In many cases, it would have been helpful to create summary tables that allow the reader to directly compare percentages between groups and better ascertain the statistical significance of the observed results. For example, for Figure 1b- it would be helpful to show the information in tabular form so that percentages of samples across anatomic sites could be more directly compared with respect to tissue characteristics such as stromal score and keratinization; statistically significant testing could be used to determine which differences are most likely to be real and not due to chance. The raw data are included in supplementary Table 1, but a table showing the percentages across groups would be most helpful to view.

Response: Thank the reviewer for the recommendation and we appreciate it very much. According to your suggestions, we have added 4 tabular forms to exhibit the data more convenient to review.

- a) We have added a table (**Supplementary Table 1c**) for **Fig. 1b** to make the information clearer.
- b) **Supplementary Table 1d** was added to show association between clinicopathological characteristics and OS/DFS by multivariate COX proportional hazards model.
- c) **Supplementary Table 6b** was added to show HPV type-specific prevalence in 5 anogenital SCCs.
- d) **Supplementary Table 6c** was added to show how HPV status affect the p53 expression.
- e) A table was added in **Fig.6c** to show HPV16 related five Groups distribution in 5 anogenital SCCs.

Q8.

Regarding the HPV results, it is difficult to glean from Figure 6e whether the protein patterns depicted are specific to HPV 16. It would be helpful to present HPV type-specific prevalence by tumor type in tabular form, and then present the percentages of each of the five HPV groups defined in Fig 6c that express proteins corresponding to the different pathway groups of interest defined in 6d. Were HPV16 E6 and E7 proteins detected in any of the SCC samples?

Response: Thank the reviewer for giving this recommendation. We added two tables according to your suggestions. **Table RL 9** showed HPV-type specific prevalence in 5 anogenital SCCs, and **Table RL 10** referred to five groups distribution in 5 anogenital SCCs.

- To explore the expression level of proteins in **Fig. 6e**, we firstly added the non-anogenital SCCs in the comparison (**Figure RL 20a** and **20b**). There was no significant difference between Group1-3 (HPV16 infected cases) and non-anogenital SCC of EZR (Wilcoxon-rank sum test, $p = 0.0937$), PAWR (Wilcoxon-rank sum test, $p = 0.4682$), and DUSP3 (Wilcoxon-

rank sum test, $p = 0.1425$) belonging to negative regulation of T cell receptor signaling pathway (**Figure RL 20a**). Inositol phosphate catabolic process proteins showed differential expression patterns compared to non-anogenital SCCs (**Figure RL 20b**). Group 1-3 of INPP1 (Wilcoxon-rank sum test, $p < 0.0001$) and IMPA2 (Wilcoxon-rank sum test, $p < 0.0001$) showed a high-level expression, compared with non-anogenital SCCs. However, Group 1-3 of NUDT3 showed a lower expression than non-anogenital SCCs (Wilcoxon-rank sum test, $p < 0.0001$), as NUDT3 mediates phosphate degradation (**PMID: 34788624**).

- Then we further explored the protein expression in Inositol phosphate catabolic process as you suggested (**Figure RL 20c**). INPP1 showed a high expression in Group1 compared to other groups (**Figure RL 20c**). Interestingly, Inositol phosphates were reported promoting HIV-1 assembly and maturation to facilitate viral spread in human CD4+ T cells (**PMID: 33476323**). Multiple isomers of inositol phosphate were found in Epstein-Barr-virus- transformed (T5-1) B-lymphocytes and may be related with cell transformation or proliferation (**PMID: 1660712**). We hypothesize Inositol phosphate catabolic process probably related to HPV related tumorigenesis. However, due to the small sample size, a large-scale study will be needed to explore this further.

To answer the question "whether HPV16 E6 and E7 proteins were detected in any of the SCC samples", we did two things.

- On the one hand, we ordered the antibodies for HPV16 E6 (Abcam, Clone: ab70) and E7 (ProMab, Clone: 6F3) and immunohistochemistry (IHC) was performed on all pan-SCC samples. After evaluating IHC staining, we believe that the antibodies with high background and poor specificity, as HPV negative samples were with positive staining (dilution 1:10000 for both antibodies).
- On the other hand, we did an HPV database searching by searching the library of Uniprot HPV (<https://www.uniprot.org/uniprot/?query=hpv&fil=reviewed%3Ayes&sort=score>) with MS Fragger (Nesvilab). Detailed searching parameters are shown in **Figure RL 21**. As shown in **Figure RL 22**, we didn't detect HPV 16 E6 or E7 in our data. HPV26 E1 was detected in cervical SCC as the most prevalent HPV protein. We have no information about HPV26 infection status, as the HPV detection kit only includes HPV16, 18, 31, 33, 35, 39, 45, 51, 52, 56, 58, 59, 66, 68, and 82. HPV16 E4 was detected in one Anal SCC with HPV16 infection.

Table RL 9 HPV type-specific prevalence in 5 anogenital SCCs.

	Penis (22 cases)	Perineum (20 cases)	Anus (10cases)	Cervix (21 cases)	Vagina (21 cases)
HPV16	27.27%	60.00%	80.00%	52.38%	52.38%
HPV18	4.55%	0	0	9.52%	9.52%
HPV39	0	5.00%	0	0	14.29%
HPV52	9.09%	0	0	9.52%	0
HPV58	4.55%	0	10.00%	9.52%	9.52%
Others	4.55%	5.00%	10.00%	14.29%	9.52%
Negative	50.00%	30.00%	0	4.76%	4.76%

Table RL 10 Five Groups distribution in 5 anogenital SCCs.

	Penis (22 cases)	Perineum (20 cases)	Anus (10cases)	Cervix (21 cases)	Vagina (21 cases)
Group 1	22.73%	50.00%	60.00%	28.57%	23.81%
Group 2	4.55%	10.00%	20.00%	23.81%	28.57%
Group 3	4.55%	0	20.00%	28.57%	33.33%
Group 4	18.18%	10.00%	0	14.29%	9.52%
Group 5	50.00%	30.00%	0	4.76%	4.76%

Figure RL 20 Plots showed protein expression of molecules in (a) Negative regulation of T cell receptor signaling pathway (DUSP3, EZR, and PIPT1) and Inositol phosphate catabolic process (INPP1, IMPA2, and NUDT3). The protein expression comparison of INPP1, IMPA2, and NUDT3 was compared among anogenital SCC Group 1-4, Group 4, Group5, and non-anogenital SCCs (b); was also compared among 5 Groups in each anogenital SCCs (c).

Redacted

Figure RL 21 HPV database searching parameters by MSFragger.

Figure RL 22 HPV database searching results.

Q9.

The survival analysis described in Fig 2e is intriguing, given that these proteomic features may be useful prognostic markers. It appears as if fewer proteins were predictive of survival in the current PanSCC dataset compared to a majority of the TCGA datasets included in Fig 2e. Could this be a function of sample size? It would be helpful if the Hazard Ratios and 95% confidence intervals were provided to better interpret the magnitude and precision of these estimates.

Response: Thank the reviewer for this valuable comment. We think that fewer proteins were predictive of survival in the pan-SCC cohort, probably due to the cohort's complexity and the survival analysis method we used is really strict.

- The pan-SCC cohort included a total of 333 patients from 17 different SCCs, while each of the TCGA dataset is just one tumor type.
- The survival analysis method for pan-SCC we used in this study is the multivariate Cox proportional hazards model. We included age, gender, stage, histology, organ, and protein expression as covariates in the revised analysis as you suggested in Q6 and did multiple testing correction (BH adjusted) per reviewer 1's comments. In this case, the numbers of proteins with prognostic values becomes less. The survival analysis method for TCGA cohorts is Kaplan-Meier curve with log-rank test.

After this strict analysis, we got two molecules with prognostic statistical significance in pan-SCC cohort, including SERPINE1 (P53 downstream pathway; (95%CI:3.6-210); $p = 0.0135$, BH adjusted), RPL12 (Ribosome; (95%CI:0.45-0.82); $p = 0.036$, BH adjusted). In this case, we believe these two proteins should be critical for pan-cancer prognosis. In the revised manuscript, we showed SERPINE1 and RPL12 in **Supplementary Fig. 6b**. The detailed survival information is available in **Supplementary Table 2d**.

REVIEWERS' COMMENTS

Reviewer #1 (Remarks to the Author):

This paper revision addresses the concerns and questions. We appreciate the significant effort and additional analysis that was performed in support of this revision.

Several issues still should be addressed:

- Batch effect analysis: Line 741, "interaction" should be "intersection".

- Please clarify in the text line 97 "68 patients lost follow-up". Does this mean that these patients did not have any information? The supplemental table suggests these patients are missing information. This is different from lost to follow-up which assumes information about the patient until a specific time point (at which the patient outcome is censored). As there are censored patients in the KM plots, I believe these patients have no outcome information. If so, "68 patients with no outcome information" or similar would be more appropriate.

- Line 462-464: Can you verify that the accuracy was from the training set (the manuscript states this)? Although it is stated that the data was split into 75/25 there is no accuracy reported for the validation set? Figure 7b shows 100% sensitivity/specificity (as reported from the training set) but the counts of tumors (Tumors, No) suggests that results may be based on validation set. Accuracy on the training set is not as informative as it is expected to be optimistically biased, but it is stated this way in the manuscript. However, if the intent is to report on the validation set accuracy this was not described.

Reviewer #2 (Remarks to the Author):

The authors have provided an extensive and convincing response to the reviewers' critiques. My suggestions have been addressed in the new version. I have only 3 minor comments:

1. Please provide the final concentration in the sample for DTT and IAA.
2. The authors could standardize the nomenclature of ammonium bicarbonate (NH_4HCO_3 or ABC).
3. Why was N-terminal acetylation not considered as a variable modification in MS analysis?

Reviewer #3 (Remarks to the Author):

The authors are to be commended for their thorough responses to the critiques. The manuscript revisions have greatly improved the quality of the work. Given this enhanced rigor and the importance of the topic, I suggest this work be published.

REVIEWERS' COMMENTS

Reviewer #1 (Remarks to the Author):

This paper revision addresses the concerns and questions. We appreciate the significant effort and additional analysis that was performed in support of this revision.

Several issues still should be addressed:

- Batch effect analysis: Line 741, "interaction" should be "intersection".

Response:

Thank the reviewer very much for this comment. We have changed the "interaction" to "intersection" (Line 718 of the manuscript with tracked changes accepted).

- Please clarify in the text line 97 "68 patients lost follow-up". Does this mean that these patients did not have any information? The supplemental table suggests these patients are missing information. This is different from lost to follow-up which assumes information about the patient until a specific time point (at which the patient outcome is censored). As there are censored patients in the KM plots, I believe these patients have no outcome information. If so, "68 patients with no outcome information" or similar would be more appropriate.

Response:

Thank the reviewer for this valuable comment. We agree that "68 patients with no outcome information" is more appropriate. We have changed "68 patients lost follow-up" to "68 patients with no outcome information" in line 94 (Results part) and line 592 (Methods part).

- Line 462-464: Can you verify that the accuracy was from the training set (the manuscript states this)? Although it is stated that the data was split into 75/25 there is no accuracy reported for the validation set? Figure 7b shows 100% sensitivity/specificity (as reported from the training set) but the counts of tumors (Tumors, No) suggests that results may be based on validation set. Accuracy on the training set is not as informative as it is expected to be optimistically biased, but it is stated this way in the manuscript. However, if the intent is to report on the validation set accuracy this was not described.

Response:

Thank the reviewer for this critical comment. We made a mistake that **Fig. 7b** is based on the validation set. In the revised manuscript, we made the following modifications.

On the one hand, we added the sensitivity/specificity of the training set in **Supplementary Fig. 11a**. The sensitivity and specificity of the training set were all 100%, which is as same as the validation set.

On the other hand, we modified the result descriptions as follows. "In our training set of 249 patients, the diagnostic SCC type was accurately predicted (both sensitivity and specificity were 100%) in all patients based on 10-fold cross-validation (Supplementary Fig. 11a). When applied to the validation set of 84 samples, the model achieved 100% for both sensitivity and specificity (Fig. 7b)."

Thank the reviewer again for pointing out this mistake.

Reviewer #2 (Remarks to the Author):

The authors have provided an extensive and convincing response to the reviewers' critiques. My suggestions have been addressed in the new version. I have only 3 minor comments:

1. Please provide the final concentration in the sample for DTT and IAA.

Response:

Thank the reviewer for this comment.

In the sample lysis procedure, the final concentration for DTT is 0.1 M. The Tris-HCL and DTT concentrations are both 0.1 M instead of 1 mM. We modified the concentration in the revised manuscript.

In the reduction and alkylation procedure, the final concentration in the sample for DTT and IAA are 10 mM and 55 mM, respectively.

2. The authors could standardize the nomenclature of ammonium bicarbonate (NH₄HCO₃ or ABC).

Response:

Thank the reviewer for this comment. We have standardized the “ammonium bicarbonate” to “NH₄HCO₃”.

3. Why was N-terminal acetylation not considered as a variable modification in MS analysis?

Response:

Thank the reviewer for this comment. Oxidation (M) and N-acetylation were set as the variable modifications. It is our neglect that forgets to write down the N-acetylation.

Reviewer #3 (Remarks to the Author):

The authors are to be commended for their thorough responses to the critiques. The manuscript revisions have greatly improved the quality of the work. Given this enhanced rigor and the importance of the topic, I suggest this work be published.

Thank the reviewer very much for the positive comments and suggestions.